# Sputum Microbiome and Chronic Obstructive Pulmonary Disease in a Rural Ugandan Cohort of Well-Controlled HIV Infection

Alex Kayongo,[a,b] Theda Ulrike Patricia Bartolomaeus,[c,d,e,f] Till Birkner,[d,e,f] Lajos Markó,[c,d,e,f] Ulrike Löber,[c,d,e,f] Edgar Kigozi,[b] Carolyne Atugonza,[b] Richard Munana,[j] Denis Mawanda,[a] Rogers Sekibira,[a] Esther Uwimaana,[a,b] Patricia Alupo,[a] Robert Kalyesubula,[h,i] Felix Knauf,[j] Trishul Siddharthan,[k] Bernard S. Bagaya,[b] David P. Kateete,[b] Moses L. Joloba,[b] Nelson K. Sewankambo,[i] Daudi Jjingo,[l,m] Bruce Kirenga,[a,i] William Checkley,[n,o] Sofia K. Forslund[c,d,e,f,g,p]

[a]Makerere University Lung Institute, Makerere University College of Health Sciences, Kampala, Uganda
[b]Makerere University, College of Health Sciences, Department of Immunology and Molecular Biology, Kampala, Uganda
[c]Charité-Universitätsmedizin Berlin, Corporate Member of Freie Universität Berlin, Humboldt-Universität zu Berlin, and Berlin Institute of Health, Berlin, Germany
[d]Experimental and Clinical Research Center, A Cooperation of Charité - Universitätsmedizin Berlin and Max Delbrück Center for Molecular Medicine, Berlin, Germany
[e]Max Delbrück Center for Molecular Medicine in the Helmholtz Association, Berlin, Germany
[f]German Centre for Cardiovascular Research, Berlin, Germany
[g]Berlin Institute of Health (BIH), Berlin, Germany
[h]African Community Center for Social Sustainability (ACCESS), Department of Research, Nakaseke, Uganda
[i]Makerere University, College of Health Sciences, Department of Medicine, Kampala, Uganda
[j]Department of Nephrology and Medical Intensive Care, Charité-Universitätsmedizin Berlin, Berlin, Germany
[k]University of Miami, School of Medicine, Division of pulmonary and critical care medicine, Miami, Florida, USA
[l]Makerere University, College of Computing and Information Sciences, Department of Computer Science, Kampala, Uganda
[m]African Center of Excellence in Bioinformatics and Data Science, Infectious Diseases Institute, Kampala, Uganda
[n]Division of Pulmonary and Critical Care Medicine, Johns Hopkins University, Baltimore, Maryland, USA
[o]Center for Global Non-Communicable Disease Research and Training, School of Medicine, Johns Hopkins University, Baltimore, Maryland, USA
[p]European Molecular Biology Laboratory, Structural and Computational Biology Unit, Heidelberg, Germany

Alex Kayongo, Theda Ulrike Patricia Bartolomaeus, and Till Birkner contributed equally to writing this manuscript. Author order was determined by drawing straws.

**ABSTRACT** Sub-Saharan Africa has increased morbidity and mortality related to chronic obstructive pulmonary disease (COPD). COPD among people living with HIV (PLWH) has not been well studied in this region, where HIV/AIDS is endemic. Increasing evidence suggests that respiratory microbial composition plays a role in COPD severity. Therefore, we aimed to investigate microbiome patterns and associations among PLWH with COPD in Sub-Saharan Africa. We conducted a cross-sectional study of 200 adults stratified by HIV and COPD in rural Uganda. Induced sputum samples were collected as an easy-to-obtain proxy for the lower respiratory tract microbiota. We performed 16S rRNA gene sequencing and used PICRUSt2 (version 2.2.3) to infer the functional profiles of the microbial community. We used a statistical tool to detect changes in specific taxa that searches and adjusts for confounding factors such as antiretroviral therapy (ART), age, sex, and other participant characteristics. We could cluster the microbial community into three community types whose distribution was shown to be significantly impacted by HIV. Some genera, e.g., *Veillonella*, *Actinomyces*, *Atopobium*, and *Filifactor*, were significantly enriched in HIV-infected individuals, while the COPD status was significantly associated with *Gammaproteobacteria* and *Selenomonas* abundance. Furthermore, reduced bacterial richness and significant enrichment in *Campylobacter* were associated with HIV-COPD comorbidity. Functional prediction using PICRUSt2 revealed a significant depletion in glutamate degradation capacity pathways in HIV-positive patients. A comparison of our findings with an HIV cohort from the United Kingdom revealed significant differences in the sputum microbiome composition, irrespective of viral suppression.

Address correspondence to Sofia K. Forslund, Sofia.Forslund@mdc-berlin.de.

The authors declare a conflict of interest. All authors declare no conflict of interest except Dr. Knauf who reports grants from Oxalosis and Hyperoxaluria Foundation and Deutsche Forschungsgemeinschaft during the conduct of the study; personal fees from Allena Pharmaceuticals, OxtheraPharmaceuticals, Sanofi Pharmaceuticals, Fresenius Medical Care, Alnylam Pharmaceuticals and Advicenne outside the submitted work.

**IMPORTANCE** Even with ART available, HIV-infected individuals are at high risk of suffering comorbidities, as shown by the high prevalence of noninfectious lung diseases in the HIV population. Recent studies have suggested a role for the respiratory microbiota in driving chronic lung inflammation. The respiratory microbiota was significantly altered among PLWH, with disease persisting up to 3 years post-ART initiation and HIV suppression. The community structure and diversity of the sputum microbiota in COPD are associated with disease severity and clinical outcomes, both in stable COPD and during exacerbations. Therefore, a better understanding of the sputum microbiome among PLWH could improve COPD prognostic and risk stratification strategies. In this study, we observed that in a virologically suppressed HIV cohort in rural Uganda, we could show differences in sputum microbiota stratified by HIV and COPD, reduced bacterial richness, and significant enrichment in *Campylobacter* associated with HIV-COPD comorbidity.

**KEYWORDS** sputum, airway microbiome, COPD, HIV, HIV-associated COPD, AIDS, microbiome, airway, human immunodeficiency virus

Improved access to antiretroviral therapy (ART) among people living with HIV (PLWH) has resulted in a decrease in HIV-associated morbidity and mortality over the past 2 decades (1, 2). This is particularly true in low- and middle-income countries (LMICs), which bear an immense burden of HIV/AIDS (3). Reducing mortality has substantially increased life expectancy, which now approaches the general population (2). Consequently, increased attention has been paid to the emerging burden of noncommunicable diseases (NCDs) among survivors (4). For example, Sub-Saharan Africa has the highest density of PLWH and has experienced dramatic increases in chronic obstructive pulmonary disease (COPD) prevalence (5–7).

Recent studies have suggested a role played by airway microbiota in driving chronic lung inflammation (8–10). The respiratory microbiota was shown to be significantly altered among PLWH, with disease persisting for up to 3 years after treatment initiation and HIV suppression (11). This significant alteration in the respiratory microbial community has been reported among PLWH with advanced disease compared to an uninfected population (11). A comparative study analyzing the respiratory microbiota among PLWH well-controlled on ART and an HIV-negative population, as part of the Lung HIV-Microbiome Project (LHMP), showed no differences in bacterial diversity or bacterial composition (12). Additionally, CD4 cell count, an essential parameter in HIV management, was not linked to changes in the respiratory microbiota composition among virologically suppressed participants (12). However, the question arises of how comorbidity of PLWH and COPD, as is increasingly common in sub-Saharan Africa (5–7), affects the dynamics of the respiratory microbiome. Using sputum samples as an acceptable proxy for the respiratory microbiota (13), we wanted to elucidate the microbial composition and its projected functional profile among PLWH diagnosed with COPD in our cohort. Although sputum microbiota may not fully represent the respiratory microbiota in individuals diagnosed with COPD, it has been shown that its community structure and diversity are affected in different COPD states (14). A better understanding of the respiratory microbiome among PLWH could improve COPD prognostic and risk stratification strategies in HIV (15). We analyzed the sputum microbiota among COPD and HIV-infected individuals with this aim. Finally, we compared our results with sputum microbiome data from a UK-based HIV cohort to identify geographical characteristics of microbial composition (16).

## RESULTS

**Characteristics of the Ugandan cohort.** We conducted a cross-sectional study in rural Uganda between February 2018 and February 2021. We randomly recruited participants from two large independent cohorts (Lung Function in Nakaseke and Kampala study (LiNK) and HIV-infected Lung Function in Nakaseke study (HiLiNK)) in the same geographic location (rural Nakaseke district) (5, 17). COPD screening was performed in both cohorts. In the rural Nakaseke communities, 656 HIV-negative individuals were screened, while 722 HIV-infected individuals from four HIV treatment centers within the Nakaseke district were screened (Fig. 1). Two hundred participants (50 HIV-positive [HIV$^+$]/COPD-positive [COPD$^+$], 50

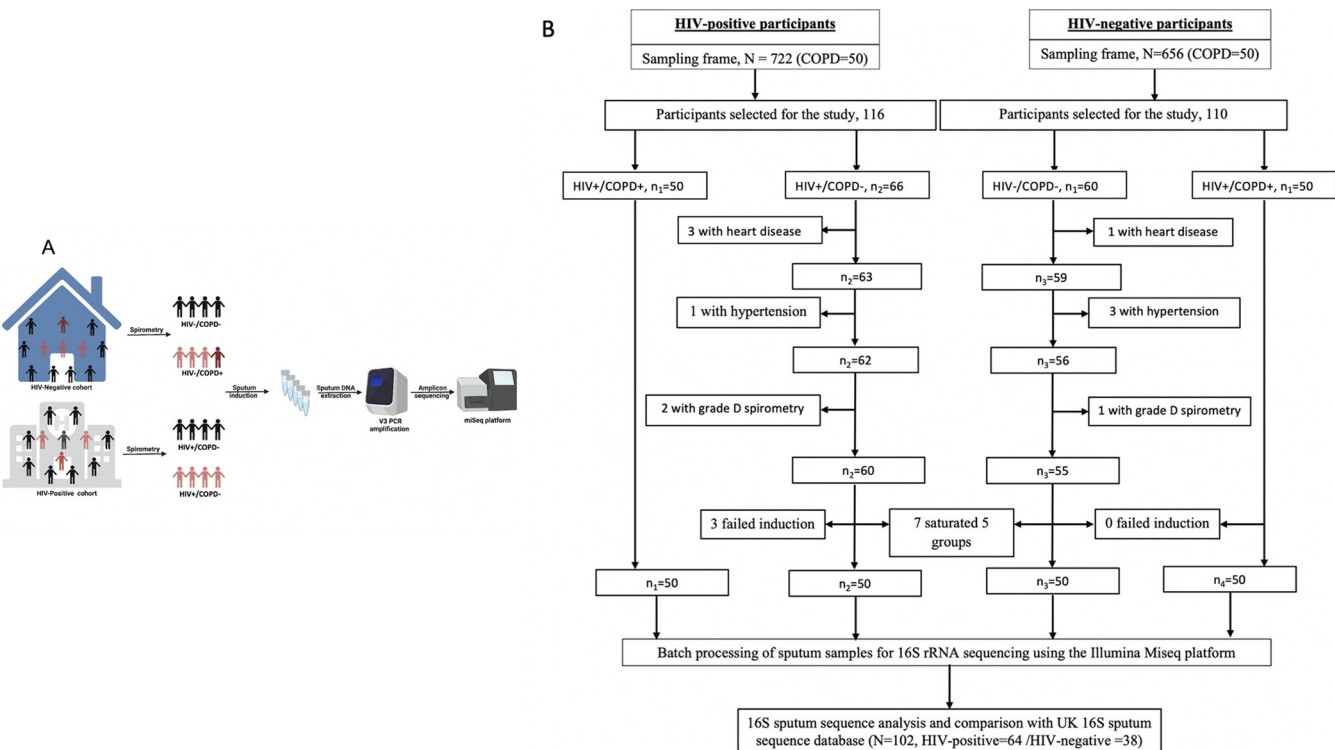

**FIG 1** Study design schema and flow diagram for participant screening and enrollment. The schema and flow diagram illustrate the two independent cohorts in the same geographic location from which participants were recruited. We randomly selected and enrolled 50 HIV-positive individuals diagnosed with COPD, 50 HIV-negative individuals with COPD, and 50 HIV-positive and 50 HIV-negative individuals without COPD. Once each group was saturated at 50 participants, we excluded extra participants.

HIV$^+$/COPD-negative [COPD$^-$], 50 HIV-negative [HIV$^-$/]/COPD$^+$, and 50 HIV$^-$/COPD$^-$) were recruited and provided induced sputum samples for downstream 16S rRNA sequencing.

Matching was performed based on three participant characteristics (age, sex, and smoking status). However, it was limited by a small sampling frame of $n = 50$ individuals in the COPD$^+$/HIV$^-$ group from the LiNK cohort, a small sampling frame of $n = 50$ individuals in the COPD$^+$/HIV$^+$ group from the HiLiNK cohort, and differences in sociodemographic and clinical characteristics between the LiNK and HiLiNK cohort as illustrated in Table 1. To control for these matching limitations, we determined the differences in the distribution of sociodemographic characteristics across the four groups (COPD$^+$/HIV$^+$, COPD$^-$/HIV$^+$, COPD$^+$/HIV$^-$, COPD$^-$/HIV$^-$) and used *post hoc* testing for the role of these covariates using a confounder-aware statistical tool (metadeconfoundR) in all downstream analyses (18). Fifty-nine percent of participants were male (59%), and 43% were aged >55 years. Among the HIV-negative group, participants with COPD were significantly older ($P < 0.001$), with lower body mass index (BMI) ($P = 0.001$), extensive use of respiratory medications (i.e., antibiotics, prednisone, and salbutamol) ($P < 0.001$), and were predominantly nonsmokers ($P = 0.017$) compared to COPD-negative individuals. Most COPD-positive participants had mild (49%) and moderate (37%) airflow limitation with no significant difference in HIV status (Table 1). Among HIV-infected participants, over 98% were on ART, 86% were virologically suppressed with a median viral load of <20 copies/mL and a median nadir CD4$^+$ T cell count of 330 cells/mm$^3$ (interquartile range [IQR], 167 to 544 cells/mm$^3$) and 383 cells/mm$^3$ (IQR, 222 to 520 cells/mm$^3$) among COPD-positive and -negative individuals, respectively. The median duration of HIV infection since diagnosis was 7.86 years (IQR, 4.96 to 14.87 years) among COPD-positive individuals and 10.24 years (IQR, 3.84 to 12.85 years) among COPD-negative individuals. Half (50%) of HIV-positive individuals with COPD and 10% without COPD reported Septrin (co-trimoxazole) use ($P < 0.001$). History of pulmonary tuberculosis was reported at 24% among HIV-infected individuals with COPD compared to 8% among those without COPD ($P = 0.026$) (Table 2). Generally, individuals with COPD reported frequent respiratory symptoms

**TABLE 1** Sociodemographic and clinical characteristics of the study participants

| Characteristic | HIV$^+$ COPD$^+$ (% [no.])$^a$; $n = 50$ | HIV$^+$ COPD$^-$ (% [no.])$^a$; $n = 50$ | P value ($x^2$) | HIV$^-$ COPD$^+$ (% [no.])$^a$; $n = 50$ | HIV$^-$ COPD$^-$ (% [no.])$^a$; $n = 50$ | P value ($x^2$) |
|---|---|---|---|---|---|---|
| Sex | | | 0.226 | | | 0.216 |
| Female | 38 (19) | 50 (25) | | 32 (16) | 44 (22) | |
| Male | 62 (31) | 50 (25) | | 68 (34) | 56 (28) | |
| | | | | | | |
| Age group (yrs) | | | 0.092 | | | 0.000 |
| 35–39 | 6 (3) | 8 (4) | | 0 (0) | 2 (1) | |
| 40–44 | 16 (8) | 20 (10) | | 8 (4) | 16 (8) | |
| 45–54 | 38 (19) | 42 (21) | | 18 (9) | 54 (27) | |
| 55–80 | 40 (20) | 30 (15) | | 74 (37) | 28 (14) | |
| Median age (IQR) | 54.7 (47.9, 61.5) | 51.0 (46.9, 56.8) | | 54.6 (48.4, 63.7) | 52.3 (47.3, 61.3) | |
| | | | | | | |
| BMI (kg/m$^2$) | | | 0.127 | | | 0.001 |
| <18.5 | 22 (11) | 2 (1) | | 36 (18) | 4 (2) | |
| 18.5–24.9 | 58 (29) | 74 (37) | | 40 (20) | 58 (29) | |
| 25–29.9 | 12 (6) | 14 (7) | | 20 (10) | 32 (16) | |
| >30 | 8 (4) | 10 (5) | | 4 (20) | 6 (3) | |
| Median BMI (IQR) | 21.3 (19.3, 24.8) | 22.2 (20.5, 26.6) | | 21.9 (19.1, 25.7) | 22.1 (20.0, 26.6) | |
| | | | | | | |
| Smoking history | | | 0.521 | | | 0.017 |
| Nonsmoker | 48 (24) | 70 (35) | | 46 (23) | 88 (44) | |
| Daily smoker | 6 (3) | 6 (3) | | 4 (2) | 2 (1) | |
| Occasional smoker | 44 (22) | 24 (12) | | 50 (25) | 10 (5) | |
| Unknown | 2 (1) | 0 (0) | | 0 (0) | 0 (0) | |
| | | | | | | |
| Pulmonary tuberculosis | | | 0.026 | | | 1.000 |
| Yes | 24 (12) | 8 (4) | | 2 (1) | 2 (1) | |
| No | 76 (38) | 92 (46) | | 98 (49) | 98 (49) | |
| | | | | | | |
| Biomass exposure | | | 0.232 | | | 0.662 |
| Wood | 86 (43) | 92 (46) | | 94 (47) | 94 (47) | |
| Other | 14 (7) | 8 (4) | | 6 (3) | 6 (3) | |
| | | | | | | |
| Respiratory medication | | | 0.233 | | | 0.0001 |
| Antibiotics >2 wks before study | 10 (5) | 4 (2) | | 32 (16) | 4 (2) | |
| Inhaled salbutamol use | 10 (5) | 0 (0) | | 22 (11) | 0 (0) | |
| Inhaled prednisone use | 4 (2) | 4 (2) | | 22 (11) | 0 (0) | |
| | | | | | | |
| Spirometry parameters | | | | | | |
| Post-BD FEV$_1$ (L), median (IQR) | 1.97 (1.55,2.34) | 2.53 (2.08, 2.96) | | 1.55 (1.29,2.15) | 2.48 (2.06,2.92) | |
| Post-BD FVC (L), median (IQR) | 2.96 (2.38,3.47) | 3.00 (2.50, 3.56) | | 2.98 (2.44,3.44) | 2.95 (2.45,3.48) | |
| Post-BD ratio, median (IQR) | 0.69 (0.62,0.80) | 0.79 (0.70, 0.84) | | 0.69 (0.62,0.78) | 0.79 (0.66,0.84) | |
| FEV$_1$ % predicted, median (IQR) | 89.5 (68.0,108.0) | 103 (87, 115.0) | | 88.5 (68.0,108) | 103 (82,116.0) | |
| | | | | | | |
| Airflow limitation classification | | | | | | 0.313 |
| Mild obstruction | 56 (28) | | | 42 (21) | | |
| Moderate Obstruction | 30 (15) | | | 44 (22) | | |
| Severe obstruction | 14 (7) | | | 14 (7) | | |
| Very severe obstruction | 0 (0) | | | 0 (0) | | |

$^a$Values indicate percentage and total number unless indicated otherwise.

compared to COPD-negative individuals. No participant reported the use of any recreational drugs (Table 3). Our cohort results were compared to the sputum microbiome data from a UK-based HIV cohort (16). In this cohort, sputum samples were collected from 64 HIV-infected individuals with a median CD4 count of 676 cells/μL and 38 HIV-negative individuals. Above 80% of HIV-infected individuals were virologically suppressed with viral loads below 40 copies/mL of blood. No significant differences were reported between HIV-infected and -negative groups regarding age, sex, educational level, body mass index (BMI), and comorbidities. In the UK cohort, current tobacco smoking and recent use of recreational drugs were significantly higher in HIV-infected individuals. The spirometric patterns were normal for most participants, with only 10 HIV-infected and two HIV-negative participants

**TABLE 2** HIV-specific clinical characteristics stratified by COPD status among participants

| Characteristic | COPD$^+$ (% [no.])$^a$; n = 50 | COPD$^-$ (% [no.])$^a$; n = 50 | $x^2$ P value |
|---|---|---|---|
| Duration since HIV diagnosis (yrs) | | | 0.916 |
| $<1$ | 2 (1) | 2 (1) | |
| 1–3.9 | 10 (5) | 20 (10) | |
| 4–9.9 | 30 (15) | 20 (10) | |
| 10+ | 30 (15) | 42 (21) | |
| Unknown | 8 (4) | 16 (8) | |
| Median duration since HIV diagnosis (IQR) | 7.86 (4.92, 14.87) | 10.24 (3.84, 12.85) | |
| Antiretroviral therapy (ART) | | | |
| Currently on ART | 98 (49) | 100 (50) | 0.999 |
| Not ART | 2 (1) | 0 (0) | |
| ART Duration (yrs) | | | 0.634 |
| $<1$ | 12 (6) | 6 (3) | |
| 1–3.9 | 16 (8) | 18 (9) | |
| 4–9.9 | 42 (21) | 34 (17) | |
| 10+ | 30 (15) | 42 (21) | |
| Median ART duration for those on ART, yrs (IQR) | 7 (4, 10) | 8 (4, 12) | |
| Drug use | | | |
| AZT | 32 (16) | 34 (17) | 0.376 |
| NVP | 26 (13) | 34 (17) | 0.123 |
| EFV | 50 (25) | 38 (19) | 0.679 |
| TDF | 54 (27) | 40 (20) | 0.568 |
| 3TC | 92 (46) | 78 (39) | 0.360 |
| ABC | 4 (2) | 2 (1) | 0.670 |
| Septrin | 50 (25) | 10 (5) | 0.000 |
| INH | 10 (5) | 10 (5) | 0.823 |
| Viral load | | | 0.338 |
| >20 copies/mL | 14 (7) | 8 (4) | |
| ≤20 copies/mL | 86 (43) | 92 (46) | |
| Median viral load/copies/mL | $<20$ | $<20$ | |
| Nadir CD4$^+$ T cells (cells/mm³) | | | 0.469 |
| $<200$ | 24 (12) | 18 (9) | |
| 200–499 | 38 (19) | 40 (20) | |
| ≥500 | 22 (11) | 24 (12) | |
| Unknown | 16 (8) | 18 (9) | |
| Median CD4$^+$ T cells (cells/mm³) IQR | 330 (167, 544) | 383 (222, 520) | |
| History of pulmonary tuberculosis | | | 0.029 |
| Yes | 24 (12) | 8 (4) | |
| History of recurrent bacterial pneumonia | | | |
| Yes | 0 (0) | 0 (0) | 1.000 |
| History of *Pneumocystis jirovecii* pneumonia | | | |
| Yes | 0 (0) | 0 (0) | 1.000 |

$^a$Values indicate percentage and total number unless indicated otherwise.

reported with COPD (defined as forced expiratory volume in 1 second [FEV$_1$]/Forced vital capacity [FVC] $< 0.7$). All participants were free of symptoms of acute respiratory illness at the time of recruitment (16).

**Taxonomic profile of sputum microbiome in well-controlled HIV minimally differed by COPD and HIV.** To determine induced sputum microbial composition by HIV and COPD status in the Ugandan cohort, we performed 16S rRNA gene sequencing, extracted genomic DNA, and PCR-amplified it using the 341F-785R primer set to amplify the V3-V4 hypervariable region of the 16S rRNA gene. Amplicon sequencing was performed using the MiSeq platform following standard protocol. A total of 8,318,227 sequence raw reads were generated and processed using LotuS (1.62) (19). The number of raw reads retrieved from the COPD$^-$/

**TABLE 3** Respiratory symptoms and other conditions among participants stratified by COPD/HIV status

| Characteristic | HIV$^+$ COPD$^+$ (% [no.]); $n = 50$ | HIV$^+$ COPD$^-$ (% [no.]); $n = 50$ | P value ($x^2$ test) | HIV$^-$ COPD$^+$ (% [no.]); $n = 50$ | HIV$^-$ COPD$^-$ (% [no.]); $n = 50$ | P value ($x^2$ test) |
|---|---|---|---|---|---|---|
| Cough over the past 12 mo | | | 0.002 | | | 0.000 |
| Yes | 22 (11) | 2 (1) | | 36 (18) | 4 (2) | |
| No | 78 (39) | 98 (49) | | 64 (32) | 96 (48) | |
| | | | | | | |
| Shortness of breath over the past 12 mo | | | 0.000 | | | 0.000 |
| Most days a wk | 0 (0) | 0 (0) | | 2 (1) | 0 (0) | |
| Several days a wk | 0 (0) | 2 (1) | | 8 (4) | 0 (0) | |
| Only with infection | 44 (22) | 0 (0) | | 44 (22) | 6 (3) | |
| Not at all | 46 (28) | 98 (49) | | 46 (23) | 94 (47) | |
| | | | | | | |
| Night awakening over the past 12 mo | | | 0.000 | | | 0.000 |
| Most days a wk | 0 (0) | 0 (0) | | 2 (1) | 0 (0) | |
| Several days a wk | 0 (0) | 4 (2) | | 6 (3) | 0 (0) | |
| Only with infection | 34 (17) | 0 (0) | | 42 (21) | 4 (2) | |
| Not at all | 66 (33) | 96 (48) | | 50 (25) | 96 (48) | |
| | | | | | | |
| Phlegm over the past 12 mo | | | 0.012 | | | 0.000 |
| Yes | 20 (10) | 4 (2) | | 32 (16) | 2 (1) | |
| No | 78 (39) | 96 (48) | | 66 (33) | 98 (49) | |
| Unknown | 2 (1) | 0 (0) | | 2 (1) | 0 (0) | |
| | | | | | | |
| Hospitalization in the past 12 mo | | | 0.988 | | | 0.014 |
| Yes | 2 (1) | 2 (1) | | 16 (8) | 2 (1) | |
| No | 98 (49) | 98 (49) | | 84 (42) | 98 (49) | |
| | | | | | | |
| Missed work last yr (respiratory illness) | | | 0.000 | | | 0.014 |
| Yes | 22 (11) | 0 (0) | | 38 (19) | 2 (1) | |
| No | 76 (38) | 100 (50) | | 62 (31) | 98 (49) | |
| No response | 2 (1) | 0 (0) | | 0 (0) | 0 (0) | |
| | | | | | | |
| Infancy hospitalizations (respiratory illness) | | | 0.988 | | | 0.014 |
| Yes | 2 (1) | 2 (1) | | 8 (4) | 2 (1) | |
| No | 98 (49) | 98 (49) | | 92 (46) | 98 (49) | |
| | | | | | | |
| Medical history | | | | | | |
| Heart disease history | 0 (0) | 0 (0) | | 0 (0) | 0 (0) | 0.315 |
| Diabetes mellitus history | 0 (0) | 0 (0) | | 8 (4) | 6 (3) | 0.565 |
| Reflux disease history | 0 (0) | 0 (0) | | 0 (0) | 0 (0) | |
| Stroke history | 0 (0) | 0 (0) | | 0 (0) | 0 (0) | |
| | | | | | | |
| Use of recreational drugs | | | | | | |
| Yes | 0 (0) | 0 (0) | | 0 (0) | 0 (0) | |

HIV$^-$ group was significantly higher than those from the COPD$^+$/HIV$^-$ and COPD$^+$/HIV$^+$ groups ($q = 0.042$, false-discovery rate [FDR] corrected) (see Fig. S1 in the supplemental material). Poisson binomial model-based read filtering was applied (20). Operational taxonomic unit (OTU) clustering (UPARSE) (21) was based on sequence similarity of 97%, while SILVA version 138 (22) was used for taxonomic profiling. Following taxonomic classification, samples were predominantly enriched for organisms belonging to the phyla *Firmicutes*, *Bacteroidota*, and *Proteobacteria* across the cohort groups (Fig. 2A). Genus level comparison showed that all cohort groups were predominated by *Streptococcus*, *Prevotella*, *Neisseria*, and *Veillonella* (Fig. 2B). Table 4 shows mean and IQR for all taxa present in the different study groups (COPD$^-$/HIV$^-$, COPD$^+$/HIV$^-$, COPD$^-$/HIV$^+$, and COPD$^+$/HIV$^+$) on the phylum and genus level. Taxa significantly associated with the different disease groups based on the metadeconfoundR output are annotated as "yes" in the rightmost column.

To determine whether distinct microbial community structures exist within our cohort, unsupervised modeling of genus abundance frequencies using Dirichlet multinomial mixtures (DMM) was applied to the 16S rRNA data sets. Using a Laplace approximation, DMM indicated that the data set presents three distinct microbial community structures (community

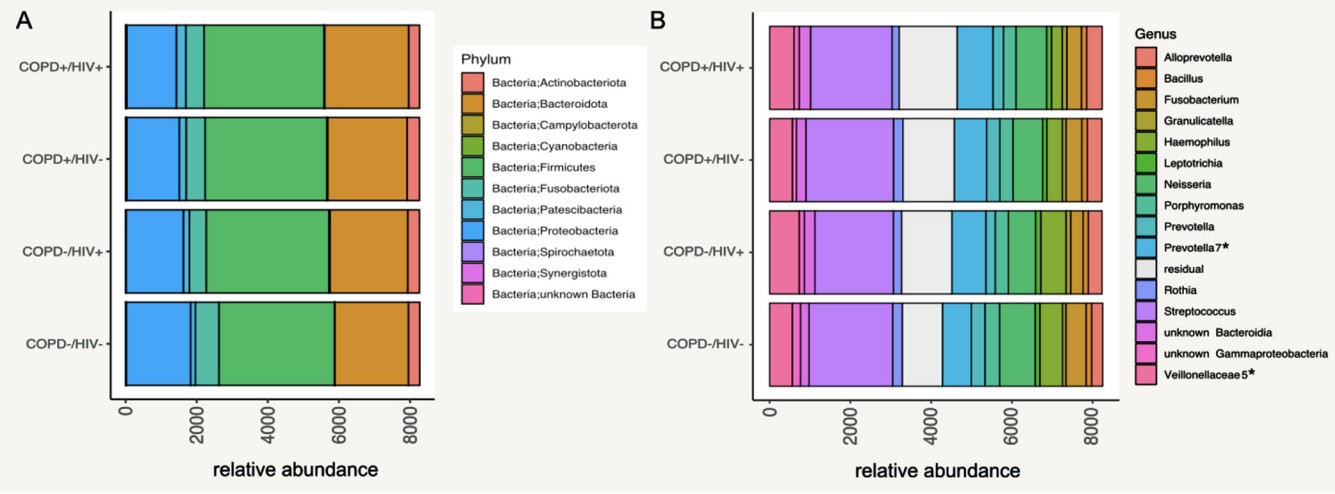

**FIG 2** Taxonomic profiles of the microbiome in all induced sputum samples. (A) Abundance of the microbiome in induced sputum samples at phylum level stratified by COPD and HIV status. (B) Abundances of the microbiome at genus level stratified by COPD and HIV status. *, Multiple groups of the same genus marked with numbers reported by Lotus due to lacking coverage in available phylogenetic databases. Here, OTUs were assigned to the same genus but had <97% identity and were not binned manually.

types 1 to 3). Community type 1, primarily dominated by *Streptococcus* followed by *Neisseria*, *Haemophilus*, and *Prevotella*, characterized 72 samples. Community type 2 is dominated by a mix of bacterial genera, including *Neisseria*, *Streptococcus*, and *Haemophilus*, followed by *Veillonella*, *Fusobacteria*, *Porphyromonas*, and *Prevotella* classified in 67 samples. Community type 3, dominated by *Prevotella*, *Streptococcus*, and *Veillonella*, followed by *Bacteroidia* and *Alloprevotella*, characterized 61 samples (Fig. 3A). Accordingly, these community types show overlapping sets of driver taxa in different proportions and accompanying rarer taxa. Univariate analysis of microbial richness and evenness diversity indices showed a significant reduction in Shannon, Simpson, inverse Simpson, and Pielou's indices and microbial evenness in community type 1 compared to the others (Fig. 3C). In contrast, community type 2 showed a significant increase compared to the other two in Shannon, Simpson, and inverse Simpson's indices and overall microbial richness (Fig. 3C). Stratification by disease status showed a slight skew in the community type distribution (Fig. 3B), with community type 3 being the rarest and slightly less so in the two HIV-positive subgroups. However, significance was achieved only by comparing COPD$^-$/HIV$^+$ subjects with COPD$^-$/HIV$^-$ ones (39% versus 17%, $q = 0.07$, FDR corrected).

**HIV-COPD comorbidity was associated with reduced bacterial richness.** To assess the impact of the studied diseases on the microbial composition, we calculated alpha diversity measures and compared the subcohorts using pairwise Wilcoxon test corrected for multiple testing. We observed a significantly higher microbial richness for the COPD$^+$/HIV$^-$ group than the COPD$^+$/HIV$^+$ ($q = 0.006$) and COPD$^-$/HIV$^-$ ($q = 0.003$) groups. Additionally, we could detect significantly higher Chao index in the COPD$^+$/HIV$^-$ group than in the other subgroups (COPD$^+$/HIV$^+$, $q = 0.001$; COPD$^-$/HIV$^+$, $q = 0.048$; and COPD$^-$/HIV$^-$, q = 0.003) (Fig. 4A). By determining intersample Bray-Curtis dissimilarities, we summarized the high-dimensional data into a reduced dimensional space to assess microbiome composition more comprehensively. Bray-Curtis dissimilarity between the different groups was visualized using principal coordinates analysis (Fig. 4B). Multivariate tests for differential composition using the Bray-Curtis distance between the sputum microbiota across the different disease states (COPD$^+$/HIV$^+$, COPD$^-$/HIV$^+$, COPD$^+$/HIV$^-$, and COPD$^-$/HIV$^-$) showed no significant differences between groups ($P > 0.05$) (Fig. 4B).

**Analysis of univariate signals in the sputum microbiome.** To elucidate whether group effects could be detected in the relative abundances of specific bacterial taxa, we conducted a univariate biomarker analysis of all taxa present in at least 1% of samples to assess their association with the collected participant clinical and sociodemographic characteristics such as age, sex, BMI, oxygen saturation, lung function parameters (pre- and

**TABLE 4** Mean relative abundance of phyla and genera across study groups

| Phylum/genus | Abundance[a] HIV+ COPD+ | Abundance[a] HIV+ COPD− | Abundance[a] HIV− COPD+ | Abundance[a] HIV− COPD- | Significant association (metaconfoundR analysis) |
|---|---|---|---|---|---|
| Phylum | | | | | |
| *Actinobacteriota* | 347.87 (±353) | 331.22 (±419) | 309.12 (±260) | 321.62 (±348) | No |
| *Bacteroidota* | 2,232.12 (±1,317.5) | 2,181.14 (±1,634) | 2,359.61 (±1,415.5) | 2,064.53 (±1,347.5) | Yes |
| *Campylobacterota* | 21.02 (±14.5) | 30.06 (±27) | 19.22 (±19) | 14.19 (±12.5) | No |
| *Cyanobacteria* | 3.42 (±0) | 4 (±1) | 0.18 (±0) | 0.77 (±0.5) | No |
| *Firmicutes* | 3,420.13 (±1,551.75) | 3,447.94 (±1,969) | 3,368.33 (±1,335.5) | 3,248.4 (±1,650) | No |
| *Fusobacteriota* | 534.17 (±376) | 471.29 (±370) | 513.14 (±394) | 658.13 (±777.5) | Yes |
| *Patescibacteria* | 192.25 (±145.25) | 171.67 (±165) | 266.65 (±215.5) | 134.36 (±107) | No |
| *Proteobacteria* | 1,475.33 (±1,249.5) | 1,606.33 (±1,943) | 1,393.31 (±1,232.5) | 1,810.09 (±1,722.5) | No |
| *Spirochaetota* | 29.98 (±23.25) | 14.43 (±17) | 31.27 (±20) | 15.96 (±15.5) | No |
| *Synergistota* | 3.48 (±3) | 1.94 (±3) | 2.63 (±2) | 2.6 (±2) | No |
| Unknown bacteria | 1.1 (±1) | 0.71 (±1) | 0.92 (±1) | 0.87 (±1) | No |
| | | | | | |
| Genus | | | | | |
| *Actinobacteriota*: *Actinomyces* | 58.98 (±52.25) | 58.55 (±63) | 87.49 (±89.5) | 47.04 (±35) | Yes |
| *Actinobacteriota*: unknown *Actinomycetaceae* | 1.42 (±2) | 1.47 (±1) | 1.37 (±2) | 1.19 (±2) | No |
| *Actinobacteriota*: unknown *Actinomycetales* | 8.79 (±8.25) | 18.04 (±8) | 14.71 (±16) | 9.45 (±6.5) | No |
| *Actinobacteriota*: *Corynebacterium* | 13.15 (±12.25) | 21 (±19) | 10.51 (±12) | 14.85 (±9.5) | No |
| *Actinobacteriota*: *Tropheryma* | 5.52 (±0) | 0.02 (±0) | 0 (±0) | 1.09 (±0) | No |
| *Actinobacteriota*: *Rothia* | 231 (±268.75) | 205.04 (±255) | 175.02 (±196.5) | 236.47 (±265) | No |
| *Actinobacteriota*: *Propionibacterium* | 2.29 (±2.25) | 8.57 (±2) | 1.47 (±2) | 0.57 (±1) | No |
| *Actinobacteriota*: *Pseudopropionibacterium* | 6.13 (±7.25) | 4.27 (±3) | 3.75 (±5) | 8.09 (±7) | Yes |
| *Actinobacteriota*: *Atopobium* | 12.83 (±6) | 11.24 (±10) | 16.25 (±13) | 4.26 (±4) | Yes |
| *Bacteroidota* F0058 | 1.33 (±1) | 3.57 (±2) | 0.84 (±1) | 2.19 (±2) | No |
| *Bacteroidota*: unknown *Paludibacteraceae* | 1.5 (±1) | 3.86 (±2) | 0.94 (±1) | 2.34 (±2) | No |
| *Bacteroidota*: *Porphyromonas* | 323.4 (±289.5) | 329.45 (±502) | 309.69 (±450) | 363.77 (±411) | No |
| *Bacteroidota*: *Alloprevotella* | 361.19 (±362.75) | 340.24 (±407) | 385.06 (±412.5) | 275.6 (±337.5) | No |
| *Bacteroidota*: *Prevotella* | 327.38 (±228.75) | 227.33 (±192) | 254.98 (±201.5) | 337.15 (±292) | No |
| *Bacteroidota*: *Prevotella* 7[a] | 802.62 (±742) | 845.9 (±1222) | 893.14 (±907.5) | 715.91 (±806) | No |
| *Bacteroidota*: unknown *Prevotellaceae* | 78.73 (±58.25) | 85.71 (±99) | 116.31 (±89) | 72.38 (±66) | No |
| *Bacteroidota*: *Rikenellaceae* RC9 | 1.83 (±2) | 1.06 (±1) | 1 (±1) | 2.02 (±1) | No |
| *Bacteroidota*: *Tannerella* | 11.08 (±12) | 9.24 (±10) | 8.31 (±9) | 8.28 (±10) | No |
| *Bacteroidota*: unknown *Bacteroidales* | 31.81 (±29.75) | 25.31 (±28) | 37.55 (±40) | 27.62 (±23.5) | No |
| *Bacteroidota*: *Capnocytophaga* | 32.9 (±19.5) | 33.98 (±30) | 23.47 (±23.5) | 24.19 (±25.5) | No |
| *Bacteroidota*: *Bergeyella* | 8.04 (±4.25) | 7.71 (±4) | 24.14 (±9) | 6.77 (±4) | No |
| *Bacteroidota*: *Lentimicrobium* | 6.92 (±6.25) | 2.22 (±1) | 4.08 (±1) | 1.72 (±1) | Yes |
| *Bacteroidota*: unknown *Bacteroidia* | 232.1 (±207.75) | 261.76 (±319) | 279.25 (±230) | 210.47 (±252.5) | No |
| *Bacteroidota*: unknown *Bacteroidota* | 11.4 (±10) | 8.51 (±5) | 8.96 (±9) | 7.77 (±7) | No |
| *Campylobacterota*: *Campylobacter* | 21.15 (±12.25) | 30.31 (±23) | 19.47 (±16.5) | 13.64 (±12) | Yes |
| *Cyanobacteria*: unknown | 1.71 (±0) | 2.39 (±1) | 0.08 (±0) | 0.43 (±0) | No |
| *Firmicutes*: *Acholeplasma* | 2.04 (±0) | 0.14 (±0) | 2.39 (±0) | 0.19 (±0) | No |
| *Firmicutes*: *Bacillus* | 135.48 (±155) | 128 (±129) | 124.08 (±107) | 131.87 (±119.5) | No |
| *Firmicutes*: *Solobacterium* | 5.23 (±5) | 8 (±7) | 9.25 (±9) | 3.98 (±3.5) | Yes |
| *Firmicutes*: unknown *Erysipelotrichaceae* | 6.19 (±6) | 8.59 (±9) | 10.92 (±10) | 4.6 (±3.5) | Yes |
| *Firmicutes*: *Abiotrophia* | 2.21 (±3) | 2.45 (±2) | 3.33 (±4) | 2.89 (±4) | No |
| *Firmicutes*: *Granulicatella* | 85.23 (±73.25) | 109.41 (±85) | 106.76 (±108.5) | 78.26 (±66) | No |
| *Firmicutes*: *Limosilactobacillus* | 2.48 (±1) | 0.57 (±0) | 0.57 (±0) | 0.32 (±0) | No |
| *Firmicutes*: 392 | 6.94 (±0.25) | 4.76 (±1) | 10.33 (±1) | 1.79 (±0) | No |
| *Firmicutes*: *Streptococcus* | 2,174.46 (±2,108) | 1,945.69 (±1,857) | 2,023.69 (±1,532) | 2,078.6 (±2,044.5) | No |
| *Firmicutes*: unknown *Lactobacillales* | 1.25 (±1) | 0.67 (±1) | 0.98 (±1) | 0.87 (±1) | No |
| *Firmicutes*: *Gemella* | 95.44 (±89.5) | 89.24 (±135) | 82.43 (±75.5) | 91.49 (±69.5) | No |
| *Firmicutes*: unknown *Staphylococcales* | 5.12 (±4) | 0.96 (±0) | 2.88 (±4) | 2.09 (±3) | Yes |
| *Firmicutes*: unknown *Bacilli* | 3.54 (±3) | 2.45 (±3) | 3.22 (±4) | 1.43 (±2) | No |
| *Firmicutes*: unknown *Clostridia* UCG | 12.08 (±11.5) | 16.94 (±13) | 16.27 (±13) | 11.4 (±13) | No |
| *Firmicutes*: *Defluviitaleaceae* UCG | 1.85 (±2) | 0.59 (±1) | 1.8 (±2) | 0.77 (±1) | No |
| *Firmicutes*: *Butyrivibrio* | 2.52 (±2) | 2.8 (±3) | 2.25 (±3) | 0.94 (±1) | Yes |
| *Firmicutes*: *Catonella* | 8 (±8.25) | 17.27 (±13) | 8 (±8) | 6.02 (±4) | Yes |
| *Firmicutes*: *Johnsonella* | 1.62 (±2) | 2.04 (±2) | 1.45 (±2) | 1.51 (±2) | No |
| *Firmicutes*: *Lachnoanaerobaculum* | 8.38 (±6.25) | 9.53 (±8) | 11.86 (±13.5) | 8.72 (±5) | No |
| *Firmicutes*: *Oribacterium* | 9.77 (±8.25) | 21 (±18) | 13.16 (±11.5) | 9.4 (±8.5) | No |
| *Firmicutes*: *Stomatobaculum* | 14.4 (±11.25) | 26.78 (±31) | 16.18 (±17.5) | 16.26 (±9.5) | Yes |
| *Firmicutes*: unknown *Lachnospiraceae* | 4.04 (±4) | 6.29 (±5) | 6.39 (±7.5) | 3.83 (±3) | No |
| *Firmicutes*: *Peptococcus* | 0.81 (±1) | 0.71 (±1) | 1.73 (±2) | 0.85 (±1) | No |
| *Firmicutes*: *Eubacterium* 2[a] | 0.75 (±0.25) | 1.37 (±0) | 0.76 (±1) | 0.7 (±1) | No |
| *Firmicutes*: *Eubacterium* | 19.79 (±19.75) | 13.88 (±17) | 13.04 (±14) | 9.15 (±10) | No |
| *Firmicutes*: *Eubacterium* 1[a] | 1.6 (±2) | 0.88 (±1) | 1.39 (±1) | 0.98 (±1) | No |
| *Firmicutes*: *Amnipila* | 0.92 (±1) | 0.82 (±1) | 1.51 (±1) | 0.6 (±0) | No |
| *Firmicutes*: *Parvimonas* | 5.19 (±8) | 2.12 (±2) | 4.35 (±5) | 4.23 (±6) | No |
| *Firmicutes*: *Filifactor* | 5.15 (±5.25) | 1.08 (±1) | 2.31 (±2.5) | 2.13 (±2.5) | Yes |
| *Firmicutes*: *Peptoanaerobacter* | 1.46 (±1) | 1.29 (±1) | 1.12 (±1) | 0.38 (±0) | No |
| *Firmicutes*: *Peptostreptococcus* | 17.38 (±19.75) | 20.69 (±13) | 23.06 (±23.5) | 23.66 (±15.5) | No |
| *Firmicutes*: unknown *Clostridia* | 28.67 (±27.5) | 45.71 (±37) | 31.84 (±25) | 26.91 (±24.5) | Yes |
| *Firmicutes*: unknown *Negativicutes* | 5.77 (±6) | 2.65 (±3) | 4.16 (±5) | 6.17 (±5.5) | Yes |
| *Firmicutes*: *Selenomonas* | 30.37 (±25) | 36.67 (±28) | 27.57 (±20) | 23 (±12.5) | Yes |

**TABLE 4** (Continued)

| Phylum/genus | Abundance[a] HIV[+] COPD[+] | Abundance[a] HIV[+] COPD[−] | Abundance[a] HIV[−] COPD[+] | Abundance[a] HIV[−] COPD- | Significant association (metadeconfoundR analysis) |
|---|---|---|---|---|---|
| *Firmicutes*: unknown *Selenomonadaceae* | 3.81 (±3) | 3.82 (±6) | 7.61 (±3) | 1.11 (±1) | No |
| *Firmicutes*: *Selenomonadales* | 38.1 (±30.75) | 39.73 (±37) | 49.04 (±44.5) | 37.53 (±26) | No |
| *Firmicutes*: *Anaeroglobus* | 2.63 (±1.25) | 0.94 (±1) | 1.29 (±2) | 1.77 (±1) | No |
| *Firmicutes*: *Dialister* | 6.27 (±6.25) | 5.92 (±6) | 7.55 (±8.5) | 11.85 (±6) | No |
| *Firmicutes*: unknown *Veillonellaceae* | 68.52 (±62.25) | 88.55 (±112) | 98.75 (±88.5) | 45.45 (±42.5) | Yes |
| *Firmicutes*: *Veillonella* | 567.19 (±431.25) | 736.12 (±971) | 603.08 (±539) | 563.96 (±545.5) | No |
| *Firmicutes*: unknown *Firmicutes* | 20.5 (±16) | 21.22 (±15) | 18.75 (±15) | 21.77 (±15.5) | No |
| *Fusobacteriota*: *Fusobacterium* | 394.1 (±369.25) | 310.71 (±266) | 368.16 (±351.5) | 505.53 (±502) | No |
| *Fusobacteriota*: *Leptotrichia* | 104.62 (±118.75) | 125.59 (±98) | 118.53 (±97.5) | 108.62 (±116.5) | No |
| *Fusobacteriota*: *Streptobacillus* | 11.04 (±3.5) | 11.49 (±4) | 10.35 (±14.5) | 3.49 (±3) | No |
| *Fusobacteriota*: unknown *Leptotrichiaceae* | 16.85 (±9.25) | 22.29 (±2) | 8.33 (±4) | 37.32 (±10.5) | Yes |
| *Fusobacteriota*: unknown *Fusobacteriales* | 1.9 (±2) | 5.61 (±5) | 2.45 (±2) | 2.77 (±2) | No |
| *Patescibacteria*: SR1 | 31.38 (±8.25) | 15.65 (±9) | 51.51 (±39.5) | 12.26 (±10.5) | No |
| *Patescibacteria*: SR1 1[a] | 4.1 (±3) | 4.33 (±4) | 10.59 (±13) | 6.19 (±5) | No |
| *Patescibacteria*: unknown *Gracilibacteria* | 1.37 (±0) | 0.96 (±0) | 8.31 (±1) | 0.32 (±0) | No |
| *Patescibacteria*: unknown "*Candidatus Saccharibacteria bacterium*" | 34.06 (±3) | 4.73 (±3) | 3.39 (±2) | 1.51 (±1.5) | No |
| *Patescibacteria*: "*Candidatus Saccharimonas*" | 13.94 (±10.25) | 20.27 (±19) | 24.63 (±24) | 12.15 (±13) | No |
| *Patescibacteria*: TM7x | 51.29 (±43) | 64.65 (±76) | 93.88 (±96.5) | 62.15 (±54) | No |
| *Patescibacteria*: "*Candidatus Saccharibacteria bacterium*" | 1.44 (±1) | 2.84 (±0) | 0.82 (±0.5) | 0.49 (±0.5) | No |
| *Patescibacteria*: unknown *Saccharimonadales* | 21.71 (±13.25) | 25.57 (±22) | 18.8 (±22.5) | 12.34 (±17) | No |
| *Patescibacteria*: unknown *Saccharimonadia* | 29.96 (±23.5) | 26.86 (±28) | 55.53 (±62) | 25.3 (±21.5) | No |
| *Proteobacteria*: unknown *Mitochondria* | 0.94 (±0) | 2.51 (±0) | 0.06 (±0) | 0.13 (±0) | No |
| *Proteobacteria*: *Achromobacter* | 0.13 (±0) | 7.18 (±0) | 10.53 (±0) | 0.02 (±0) | No |
| *Proteobacteria*: *Bordetella* | 0.04 (±0) | 8.61 (±0) | 0.02 (±0) | 0 (±0) | No |
| *Proteobacteria*: unknown *Alcaligenaceae* | 0.02 (±0) | 0.02 (±0) | 11.55 (±0) | 0 (±0) | No |
| *Proteobacteria*: *Lautropia* | 9.88 (±5.75) | 14.49 (±11) | 10.06 (±7.5) | 9.72 (±12) | No |
| *Proteobacteria*: *Neisseria* | 737.08 (±883) | 673.69 (±1070) | 760.59 (±1039) | 887.32 (±1088) | No |
| *Proteobacteria*: *Simonsiella* | 1.08 (±0) | 1.04 (±1) | 1.47 (±2) | 1.47 (±1) | No |
| *Proteobacteria*: unknown *Neisseriaceae* | 2.08 (±2) | 3.08 (±2) | 1.59 (±1) | 1.94 (±2) | No |
| *Proteobacteria*: unknown *Burkholderiales* | 0.94 (±0) | 2.69 (±0) | 0.71 (±0) | 4.09 (±0) | No |
| *Proteobacteria*: *Cardiobacterium* | 1.85 (±2) | 2.08 (±2) | 1.49 (±1.5) | 1.21 (±2) | No |
| *Proteobacteria*: *Escherichia* | 0.02 (±0) | 1.33 (±0) | 45.22 (±0) | 0 (±0) | No |
| *Proteobacteria*: *Klebsiella* | 1.29 (±0) | 0.08 (±0) | 7.92 (±0) | 1.81 (±0) | No |
| *Proteobacteria*: *Actinobacillus* | 28.96 (±8.75) | 51.18 (±23) | 30.29 (±22.5) | 35.87 (±21) | No |
| *Proteobacteria*: *Aggregatibacter* | 68.92 (±77.5) | 43.47 (±66) | 53.33 (±61) | 67.17 (±82) | No |
| *Proteobacteria*: *Haemophilus* | 387.75 (±258.75) | 630.43 (±407) | 272.84 (±287) | 566.53 (±313) | No |
| *Proteobacteria*: *Moraxella* | 128.71 (±2.25) | 34.43 (±4) | 46.41 (±16.5) | 24.85 (±3) | No |
| *Proteobacteria*: *Pseudomonas* | 0.19 (±0) | 0.02 (±0) | 3.75 (±0) | 0 (±0) | No |
| *Proteobacteria*: unknown *Gammaproteobacteria* | 98.37 (±108.75) | 123.1 (±186) | 128.16 (±159.5) | 200 (±178) | Yes |
| *Proteobacteria*: unknown *Proteobacteria* | 5.06 (±5) | 2.82 (±4) | 5.63 (±4) | 4.13 (±3) | No |
| *Spirochaetota*: *Treponema* | 25.58 (±21.5) | 12.57 (±14) | 28.9 (±22) | 13.94 (±11) | No |
| *Spirochaetota*: unknown *Spirochaetales* | 2.48 (±1) | 1.16 (±0) | 3.51 (±0) | 1.26 (±0) | No |
| *Synergistota*: *Fretibacterium* | 2.73 (±2) | 0.98 (±1) | 2.43 (±2.5) | 2 (±2) | No |
| Unknown bacteria | 1.31 (±2) | 0.57 (±1) | 0.78 (±1) | 0.81 (±1) | No |

[a]The mean abundances have been displayed in this table across all categories. The median abundances were zero for all low abundance phyla and genera.

postbronchodilator $FEV_1$, FVC, and $FEV_1$/FVC ratio), as well as the use of respiratory medications, antibiotics, and ART, while testing for confounding effects between these variables. We used here a confounder-aware biomarker search algorithm developed concurrently (metadeconfoundR package) (18) (see Fig. S2 in the supplemental material). Associations between microbial features and metadata were assessed with each potential confounder in turn for whether it retains significance in a nested rank-transformed mixed model test together with that confounder. This enabled us to determine an unbiased, confounder-aware correlation between bacterial genera and COPD/HIV status. Firstly, we could show that HIV status and its associated antiretroviral treatment, as well as the duration of such ART, display a similar pattern in influencing the sputum microbiome. Overall, HIV status has a more profound impact on the bacterial composition at phylum and genus level than COPD status (Fig. 5). We detect a significant increase in the abundance of genera, such as *Clostridia* ($q$ value = 0.02, Cliff's delta = 0.27), *Veillonellaceae* unclassified genus ($q$ value = 0.05, Cliff's delta = 0.24), *Butyrivibrio* ($q$ value = 0.05, Cliff's delta = 0.21), *Solobacterium* ($q$ value = 0.006, Cliff's delta = 0.32), and *Erysipelotrichaceae* unclassified genus ($q$ value = 0.006, Cliff's delta = 0.31) belonging to the *Firmicutes* phylum. We also detect an increase in *Atopobium* ($q$ value = 0.003, Cliff's delta = 0.34) and *Actinomyces* ($q$ value = 0.05, Cliff's delta = 0.22), with both genera belonging to the phylum *Actinobacteria*. Furthermore,

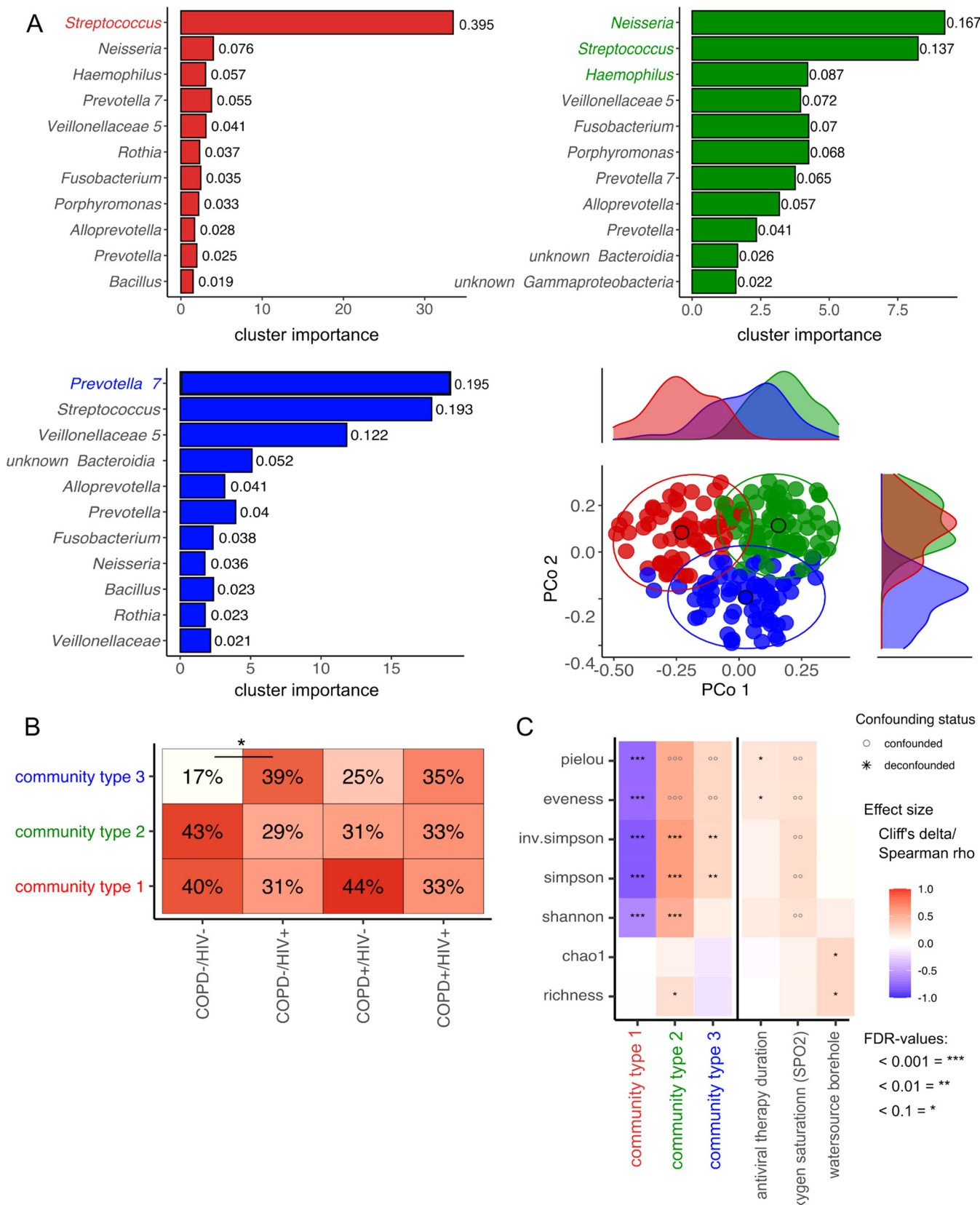

**FIG 3** Top driver microbial communities in all induced sputum samples. (A) Clustering (community typing) of all samples based on the Dirichlet multinomial model. The model was fitted on the genus relative abundance count matrix to classify genus abundance based on probability. The best fit for the tested data shows three Dirichlet multinomial groups (k), further named community types 1, 2, and 3. The x axis displays the cluster importance for each of the bacteria, and the y axis is

we see a significant decrease in *Filifactor* (*q* value = 0.007, Cliff's delta = −0.27) and *Lentimicrobium* (*q* value = 0.07, Cliff's delta = −0.19) as well as *Pseudopropionibacterium* (*q* value = 0.05, Cliff's delta = −0.22) associated with the HIV status. The COPD status is associated with a significantly increased abundance of *Selenomonas* (*q* value = 0.09, Cliff's delta = 0.26) belonging to the *Firmicutes* phylum and a significant decrease in *Gammaproteobacteria* (*q* value = 0.09, Cliff's delta = −0.26) belonging to the *Proteobacteria*. We also observed significant enrichment of *Campylobacter*. However, this signal is driven by the "dual disease status" (HIV$^+$/COPD$^+$ group). Interestingly, HIV$^+$/COPD$^+$ comorbidity (dual disease status) leads to a significant increase in *Campylobacter* (*q* value = 0.04, Cliff's delta = 0.31) and a decrease in *Staphylococcales* (*q* value = 0.01, Cliff's delta = −0.34) as well as *Negativicutes* (*q* value = 0.03, Cliff's delta = −0.32) (Fig. 5).

We determined other significant associations between the sputum microbiome, sociodemographic, and clinical characteristics. Most interesting are the associations with the variable "hospitalized as infant" and the abundance of bacteria from the phylum *Proteobacteria*, such as *Pseudomonas*, as well as an increase in *Corynebacterium* under Salbutamol use. Other host variables like BMI, pulmonary tuberculosis, treatment with co-trimoxazole, prednisone use, and cigarette smoking were not significantly associated with changes in bacterial composition. They, hence, were dropped by our model (Fig. S2). Finally, linear regression between lung function scores (FEV1, FVC, pre- and post-FEV1/FVC ratio) and alpha diversity score (Shannon index) revealed no significant correlation (see Fig. S3 in the supplemental material).

To determine the projected function profiles of the sputum microbiota using 16S rRNA data, we used PICRUST2 (version 2.2.3). PICRUST2 is a tool to infer the functional profiles of bacterial communities based on their taxonomic composition. Among the significantly associated KEGG and GMM modules, only the glutamate degradation module (MF0015) was negatively associated with HIV status and its associated antiviral therapy (see Fig. S4 in the supplemental material). COPD status was not associated with any changes in modules. However, we could detect a significant increase in modules associated with signaling machinery of two-component systems (TCSs), drug resistance, and smoking. Even more peculiarly, an individual's water source (dug well, borehole, or public tab) is significantly associated with abundant gene modules for propionate production and cellular transport systems (Fig. S4).

**Sputum microbiome composition differs by geographic location, irrespective of viral suppression.** Finally, we compared our results with sputum microbiome data from a UK-based HIV cohort (16). The recruitment criteria and the sequencing used in the UK cohort were similar to the ones used in our cohort. However, the two cohorts differed in sample handling and DNA extraction methods. We first determined the influence of combining the UK and Ugandan samples on the cluster approach by applying the Dirichlet multinomial package (version 1.36.0). The combined data set fitted the following four community types: communities 1, 2, 3, and 4. The top drivers of community 1 included a mixture of *Streptococcus*, *Bacillus*, and *Rothia*. Community 2 was predominated by *Streptococcus*; community 3 by *Neisseria* and *Streptococcus*; and community 4 had a mixture of *Streptococcus*, *Prevotella*, *Fusobacterium*, *Haemophilus*, and *Neisseria* (see Fig. S5A in the supplemental material). Community 1 dominated the UK cohort, whereas 2, 3, and 4 were predominant in

**FIG 3** Legend (Continued)

sorted by relative bacterial abundance as shown by labels. Top driver of community type 1 is *Streptococcus*; community type 2 is characterized by a mixed community between *Neisseria*, *Streptococcus*, and *Haemophilus*; and the top driver of community type 3 is *Prevotella*. In the right lower corner is the projection of genus-level sputum microbiome composition of all samples studied. Principal coordinates analysis (PCoA) plots of Bray-Curtis dissimilarity of samples colored according to their predicted sputum community type (significance based on PERMANOVA; *n* = 199 samples in 459 total). Marginal density plots depict sample group distribution alongside PCo1 and Pco2, respectively. The ellipses represent the 95% confidence interval for each cluster. (B) sputum community type distribution (vertical axis) stratified for disease status in all samples tested. Testing for significant differences between groups was done using pairwise chi-square test. (C) Influence of community type, medication, and other collected metadata variables on the alpha diversity indices of the sputum microbiome. Heatmap shows alpha diversity indices significantly [MWU (for categorical factors) and Spearman (for continuous traits) *FDR < 0.1, **FDR < 0.01, ***FDR < 0.001] associated with community type or participant characteristics. Heatmap cells show effect size (Cliff's Delta for categorical factors, Spearman's Rho for continuous traits). A multi-confounder test (nested linear model test, post hoc test) was performed, showing no asterisks or circles if it was not significant (NS) in the naive test step. For the remaining naively significant associations, only those that passed the deconfounding step as strictly deconfounded (SD), laxly deconfounded (LD), or without further covariates (NC) are black stars. At the same time, each confounded signal is a gray circle.

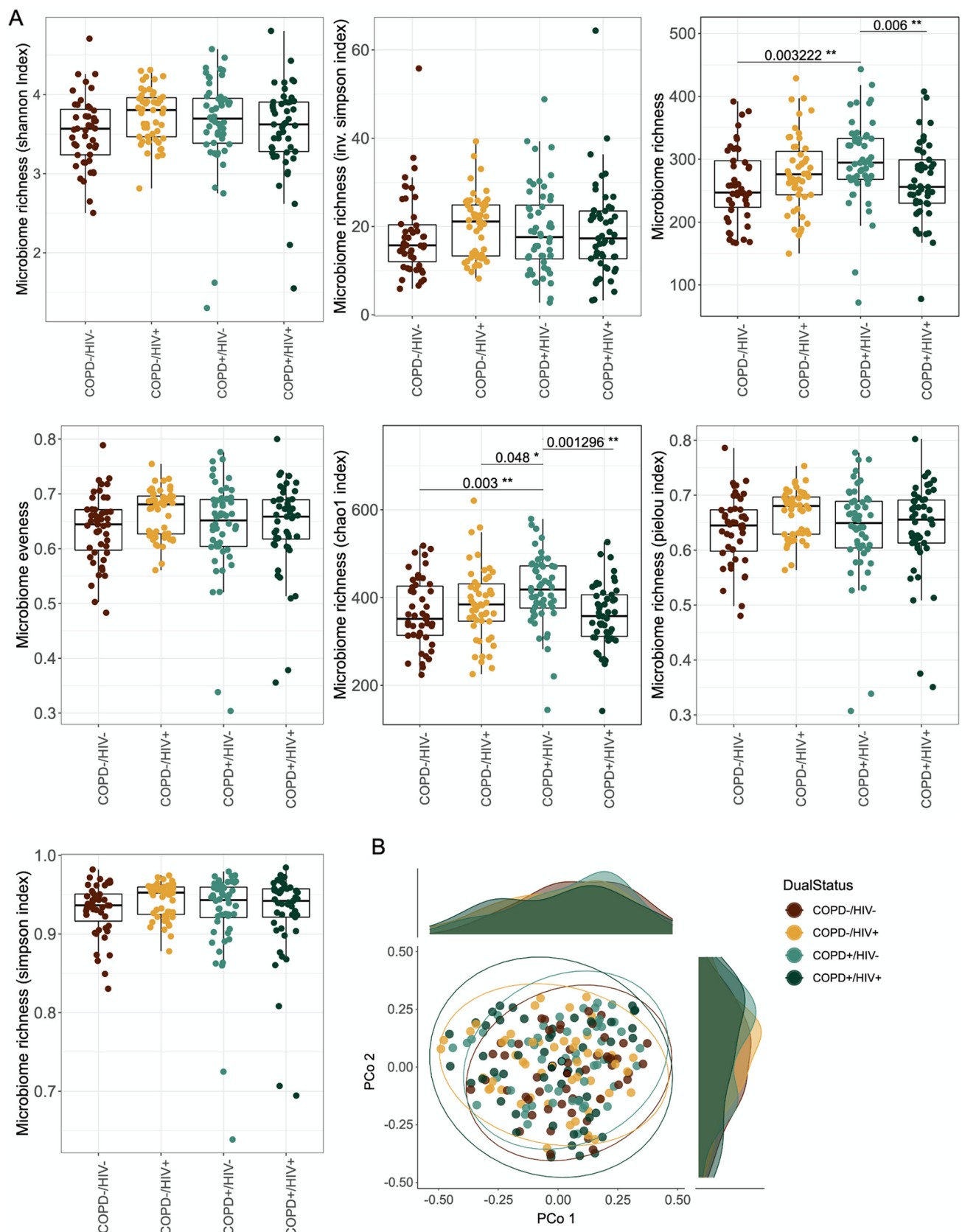

**FIG 4** Impact of disease status on sputum microbiota community structure. (A) Box plots of alpha diversity and evenness indices of the sputum microbiota. Significance based on pairwise Wilcoxon test, corrected for multiple testing using FDR. (B) Projection of genus level sputum microbiota composition of all

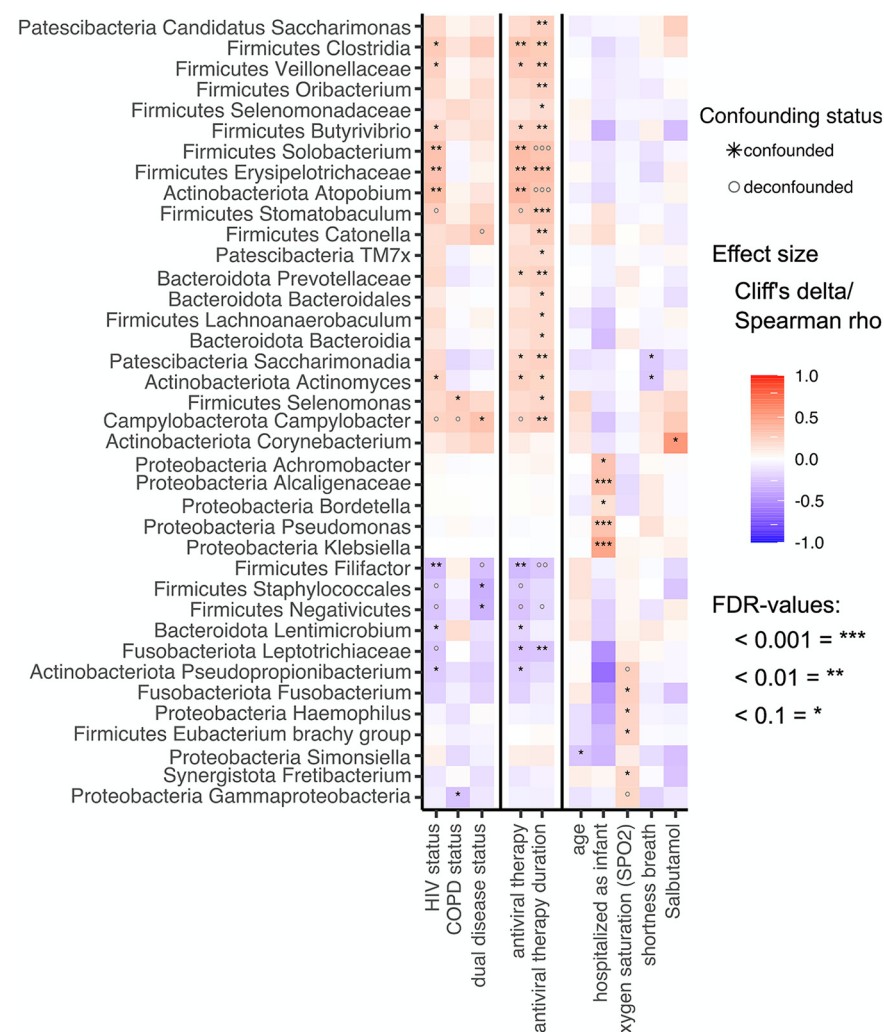

**FIG 5** Impact of disease status, medication, and other collected metadata variables on the taxonomic composition of the sputum microbiome. Heatmap shows all genus-level taxa significantly (MWU [for categorical factors] and Spearman [for continuous features]; *, FDR < 0.1; **, FDR < 0.01; ***, FDR < 0.001) different in abundance (binned rarefied 16S gene counts) depending on disease status (HIV/COPD) alongside participant characteristics. Heatmap cells show effect size (Cliff's delta for categorical factors and Spearman's rho for continuous features). Multiconfounder testing (nested linear model testing, *post hoc* test) was applied, showing no stars or circles if not significant (NS) in the naive test step. In the remaining naive-significant associations, only those passing the deconfounding step as strictly deconfounded (SD), laxly deconfounded (LD), or no other covariates (NC) are black stars. At the same time, any confounded signal is a gray circle. *, Multiple groups of the same genus marked with numbers in parentheses reported by Lotus due to lacking coverage in available phylogenetic databases. Here, OTUs were assigned to the same genus but had <97% identity and were not binned manually.

the Ugandan cohort (Fig. S5B and C). Communities 2, 3, and 4 showed similar patterns compared to the reported community types in the Ugandan cohort (Fig. 3). Principal-component analysis revealed a significant impact of cohort location and community on both PCo1 and PCo2 dimensions (see Fig. S6A in the supplemental material). We could detect a significantly higher microbial richness in the Ugandan cohort compared to that in the UK cohort ($P < 0.001$) (Fig. S6B). Additionally, some phyla were significantly influenced by the cohort ($q < 0.001$) (Fig. S6C). The authors of the UK cohort reported no difference in alpha diversity between HIV-infected and -uninfected participants, which is in line with our findings in the

**FIG 4** Legend (Continued)
samples studied colored according to their disease status. Principal coordinates analysis (PCoA) plots of Bray-Curtis dissimilarity of the sputum samples. Marginal density plots depict sample group distribution alongside PCo1 and PCo2, respectively. The ellipses represent the 95% confidence interval for each cluster.

Ugandan cohort (16). However, looking into beta diversity, the authors describe a shift between HIV-infected and HIV-uninfected participants (16).

## DISCUSSION

A few studies have investigated the impact of HIV-COPD comorbidity on the sputum microbiota composition. In this study, we profiled induced sputum microbiota from individuals in rural Uganda stratified by HIV and COPD using 16S rRNA sequencing. Univariate analysis revealed a significant difference for 18 genera, with four associations confounded by other tested metavariables. While the disease subcohorts showed only subtle differences in sputum microbiome composition in our present study, we observed a significantly higher microbiome richness for the COPD$^+$/HIV$^-$ group than the COPD$^+$/HIV$^+$ and COPD$^-$/HIV$^-$ groups. Additionally, the Chao index was significantly higher in the COPD$^+$/HIV$^-$ group than in the other subgroups. Such loss of diversity was previously reported in patients who have HIV. Somewhat unexpectedly, however, sputum richness and Chao1 index were elevated in COPD patients. It seems that these two diseases affect the microbiome differently. If underlying comorbidity of COPD/HIV is present, synergistic effects might occur, forming an interesting approach for future hypotheses.

Significant differences in the microbiota composition emerge with HIV replication and disease progression (11). Individuals with advanced HIV disease develop a microbiome profile that is markedly different from uninfected individuals (11). Therefore, in advanced HIV disease, immunosuppression results in greater variability in microbial composition because the immune system cannot restrict bacterial colonization of the respiratory tract (11). Indeed, examination of compositional differences based on alpha and beta diversity scores in HIV-infected individuals initiated on ART showed a significant decrease in microbial community dispersion, implying that viral suppression promotes microbiota composition stability (11). With the initiation of highly active antiretroviral therapy (HAART), the composition of the respiratory microbiome is not fully restored during a gradual transition from advanced HIV infection to virologic suppression. Residual microbiome changes persist for up to 3 years after ART initiation (11, 23). Such persistent changes contribute to reduced bacterial richness compared to healthy controls and impact the distribution of bacterial communities (11, 23).

In our analysis, we used an unsupervised cluster approach to define sputum microbiome communities describing variability in our cohort. The communities broadly separate into three clusters. Each sample consistently contains *Streptococcus*, *Veillonella*, *Fusobacterium*, and *Prevotella*, though in varying proportions and with other associated taxa accompanying them. We find these structures in healthy and diseased individuals; overall, the disease status does not substantially determine the community type. Comparing disease subcohorts, there is a trend toward community state 3, which generally is the least common, to be slightly less rare among participants living with HIV. However, for this data set, the significance of this trend is reached only by comparing HIV-discordant COPD-negative participants. Accordingly, while this sputum microbiome composition may represent a more dysbiotic state associated perhaps with immunosuppression, we cannot as yet conclude it, only raise it as a possibility for further testing.

Community type three is dominated by *Prevotella*, *Streptococcus*, and *Veillonellaceae*. This association between *Prevotella*, *Veillonella*, and HIV has been reported previously (24). We did not demonstrate a significant skew in community type along COPD morbidity, but community type 1, dominated by streptococci, *Neisseria*, and *Haemophilus*, was slightly more prominent in COPD$^+$ participants. This increase in streptococci, *Haemophilus*, and *Neisseria* was previously reported in COPD-positive individuals (25). Here, COPD mortality risk could be predicted using microbial-specific signatures, such as the presence of *Staphylococcus*, absence of *Veillonella*, and lower alpha diversity (26). COPD$^+$/HIV$^+$ patients showed a more even distribution between the three community types and no further bias toward any community type. How these dynamics change in HIV-associated COPD remains to be elucidated.

Several authors have reported an increased abundance of *Prevotella*, *Veillonella*, and *Streptococcus* in the lower airways of HIV-infected individuals (11, 27). A study investigating

the sputum microbiome found an association between *Staphylococcus* spp. and COPD (28). Here, COPD mortality risk could be predicted using microbial-specific signatures, such as the presence of *Staphylococcus*, absence of *Veillonella*, and lower alpha diversity scores (28). *Porphyromonas*, a prominent component of oral microbial communities, has also been reported in patients with COPD (29). Periodontal therapy improves lung function in COPD patients (29). COPD patients also showed a change in the prevalence of *Porphyromonas* associated with severe COPD exacerbation, which might indicate a possible role of *Porphyromonas* in COPD severity (30).

As expected, analysis of sputum microbe differential abundance reflected collinearity in the cohort of HIV seropositivity with ART treatment and its duration. Thus, we cannot at present disentangle the impacts of these factors. However, previously reported genera like *Veillonella*, *Actinomyces*, *Atopobium*, and *Filifactor* were significantly enriched in HIV-positive individuals (23, 27, 31). These genera were previously associated with proinflammatory cytokine production (27, 32), which may be an aspect of airway dysbiosis in HIV$^+$ subjects, possibly further interacting with other risk factors of COPD. Another interesting finding in our cohort is that COPD status is significantly associated with *Gammaproteobacteria* and *Selenomonas* abundance. *Selenomonas* was previously reported to be increased in COPD patients (33).

The interplay of the anaerobic metabolism of *Selenomonas* (34) and the underlying COPD infection could lead to anaerobic conditions in the airways favoring its growth. It has been discussed that *Selenomonas* interacts positively with anaerobes, such as *Campylobacter* and *Veillonellaceae*, forming a so-called anaerobic consortium (34). The association between COPD and *Campylobacter* is confounded by the COPD/HIV comorbidity. Indeed, patients with COPD/HIV comorbidity show significant further enrichment with *Campylobacter*, previously associated with inflammatory states. Previous research indicated *Campylobacter* elevations in PLWH, raising the possibility that further COPD comorbidity may allow increased bloom (26). One reason could be the anaerobic environment caused by COPD, which could favor the growth of *Campylobacter*. Alternatively, *Campylobacter* microaspiration might inoculate the airways and drive the establishment of a more stable ecological niche in COPD (35). Due to the robustness of sputum immune defense mechanisms, this genus rarely causes sputum infection in immunocompetent individuals. The persistence of *Campylobacter* may be favored by an impaired immune defense in the context of COPD/HIV multimorbidity, with a potential impact on further sequelae of either condition.

Furthermore, we found a depletion of *Staphylococcales* and *Negativicutes*, predominantly derived from the oral flora (26) under COPD/HIV comorbidity, again indicating an interaction between these conditions concerning host-microbiome homeostasis. Our finding of decreased glutamate degradation capacity in the HIV sputum microbiota may further elucidate aspects of pathology in context. Amino acid availability is central to the immune system's metabolism and function, especially during infection. As a condition becomes chronic, these alterations become more complex as various other areas of metabolism become impaired, and amino acids may antagonize each other's effects. Glutaminolysis has been postulated as a mechanism by which the tricarboxylic acid (TCA) cycle is replenished during viral infection (36). This decrease might indicate that the sputum microbiome in HIV patients reduces its ability to generate energy via TCA as an appropriate response to changes in the microenvironment, which might subsequently lead to dysbiosis, facilitating COPD pathogenesis.

Further direct functional assessment is needed to validate and explore this finding. Even with ART available, HIV patients are at high risk of suffering comorbidities, as shown by the high prevalence of noninfectious lung diseases in the HIV population. It is, therefore, important to better understand the complex changes in the sputum microbiota in patients with COPD/HIV comorbidity to find potential prevention and intervention targets. The presented study cohort is well-standardized and characterized. However, the cross-sectional design, which limits inference of causality, as well as the use of induced sputum samples, causing possible contamination from the oral cavity, leads to limitations. The use of short-read 16S amplicon 16S rRNA gene Illumina sequencing is limiting the resolution of taxonomic

classification to genus-level taxonomy, and the inferred function profiles using taxonomic projection likewise are limited in terms of interpretability.

Comparison of sputum microbiome between the Uganda and UK cohorts showed no significant difference in beta diversity based on HIV status. However, despite viral suppression in both studies, the difference in sputum microbiome composition was significantly driven by geographic location. The microbiome community types that we described were significantly influenced by cohort location. Community 1 (*Streptococcus*, *Bacillus*, and *Rothia*) was dominant in the UK cohort. In contrast, community 2 (predominantly *Streptococcus*), community 3 (*Neisseria* and *Streptococcus*), and community 4 (*Streptococcus*, *Prevotella*, *Fusobacterium*, *Haemophilus*, and *Neisseria*) were most common in the Ugandan cohort. Unfortunately, it is not possible within the scope of this manuscript to determine the underlying reasons for the pronounced differences in the sputum microbiota of both cohorts. First, the cohorts differ in terms of viral load and nadir CD4 level. Secondly, it is known from other publications that differences in DNA extraction methods and preprocessing of the data have an enormous impact on the bacterial community structure (37). Although we have tried to minimize the technical variation between the two data sets by processing the raw data simultaneously with the same pipeline, the sample handling and DNA extraction protocols between the two cohorts differ, and we cannot say with certainty what is causing the enormous differences between the data sets. Keeping this in mind, the geographic location is just a proxy for the two studies and their varying methodology. Although we have a very well-standardized and -characterized study population here, this study also has limitations. It is a cross-sectional study, which limits the assessment of causality.

We aimed to frequency match controls to the COPD$^+$/HIV$^+$ group based on three characteristics (i.e., age, sex, and smoking status). Unfortunately, this was not possible in the available source cohorts, resulting in at least moderate bias between groups. We used an algorithm to account for confounding factors (metadeconfoundR package) (18) (see Fig. S2 in the supplemental material), and had to rely, in this case, on the *post hoc* tests performed by the algorithm. This *post hoc* testing was performed for all potential confounders (using nested linear model comparisons). Associations between microbial traits and metadata were tested in turn with each potential confounder to see if they remained significant in a nested rank-transformed mixed model test along with the respective confounder. This way, we could detect an unbiased, unconfounded correlation between bacterial genera and HIV/COPD status.

Despite the use of induced sputum, given the passage of the specimen from the lower to the upper respiratory tract and oral cavity, it is challenging to eliminate contamination from saliva. This limitation is well known, although induced sputum is a good proxy for the lower respiratory tract (38). We used 16S rRNA gene sequencing, the limitations of which include a low resolution to genus-level taxonomy, lack of functional information, and sensitivity to methodological differences. Nonetheless, this method has been used in a wide range of different respiratory diseases. It has revealed a complex and dynamic community of microorganisms (27), and by using PICRUSt2, we could infer the functional capacity of microbiota data. Finally, our results apply to well-treated PLWH. Therefore, a clean separation of HIV and ART effects cannot be made.

In conclusion, our findings suggest that in the setting of well-controlled HIV in rural Uganda, subtle compositional differences exist in sputum microbiota stratified by HIV and COPD. We detected reduced bacterial richness and *Campylobacter* enrichment significantly associated with HIV-COPD comorbidity. Finally, in comparison with a similar, UK-based cohort, we saw sputum microbiome composition differences caused by geographic location irrespective of viral suppression; however, this finding should be verified in further studies.

## MATERIALS AND METHODS

**Study design.** We conducted a cross-sectional study in rural Uganda between February 2018 and February 2021. We randomly recruited participants from two large independent cohorts (LiNK and HiLiNK) in the same geographic location (rural Nakaseke district) (5, 17). COPD screening was performed in both cohorts. In the rural Nakaseke communities, 656 HIV-negative individuals were screened, while 722 HIV-

infected individuals from four HIV treatment centers within the Nakaseke district were screened. Figure 1 illustrates participant screening and enrollment from the two cohorts. We randomly selected 50 HIV-positive individuals diagnosed with COPD, 50 HIV-positive individuals without COPD, 50 HIV-negative individuals with COPD, and 50 HIV-negative individuals without COPD (total $n$ = 200). Participants were eligible for inclusion if they resided within the Nakaseke district, were $\geq$35 years of age, had confirmed HIV serostatus and spirometry-based COPD status at enrollment, were capable of understanding study procedures, underwent successful sputum induction, and did not have contraindications for spirometry and sputum induction procedure. We excluded participants with grade C/D spirometry curves, history of antibiotic use within 2 weeks before enrollment, asthma, or other respiratory diseases, as well as pregnant women.

**Ethical considerations.** We obtained written informed consent from all study participants. Ethical approval was granted by the Mulago Hospital Research and Ethics Committee (MHREC) (no.1996) and Uganda National Council for Science and Technology (UNCST) (no. HS2375) in Kampala, Uganda.

**Spirometry.** Participants were screened for COPD using spirometry (17). Spirometry was performed by trained research assistants using an Easy On-PC spirometer (ndd Medical Technologies, Zurich, Switzerland) before and 15 to 20 min after administration of 400 $\mu$g of salbutamol using a spacer. The quality of spirometry test results was assessed according to the American Thoracic and European respiratory societies (ATS/ERS) guidelines (39). In this study, grade A and B were considered the highest quality spirometry curves obtained following standard ATS/ERS guidelines. It has been shown that the average $FEV_1$ and FVC are minimally influenced by grades A and B (40). On the contrary, grade C/D were considered low-quality curves and excluded.

**HIV testing and HIV-related characteristics.** At recruitment, participants were retested for HIV according to ART clinic guidelines in Uganda. HIV-related medical history, including nadir $CD4^+$ T cell count, viral load, Septrin use, ART drug use, and history of opportunistic infections or pulmonary tuberculosis, was recorded from patient medical records, including sociodemographic and other clinical characteristics.

**Sputum induction and sample collection.** Before the sputum induction procedure, the following three-step cleansing routine using sterile water was followed: rinsing the mouth (gargling and discarding) three times, clearing the back of the throat followed by discarding, and blowing the nose to minimize contamination with saliva or postnasal drip. Upon deep coughing and expectoration, each sputum sample was assessed for mucoid consistency, and Gram's stain procedure was performed for quality assessment. Sputum samples with less than 10 squamous epithelial cells per low-power field ($\times$10) microscopy (indicative of a lower airway sample) passed the quality control check (40). Otherwise, the sample was rejected, and the induction procedure was repeated. During the procedure, each participant was instructed to inhale and exhale 3% nebulized hypertonic saline through the mouth with the nose clipped. Three 5-min nebulizing periods were used during the collection, with a rest period of 2 min. During each rest, spirometry was repeated to determine the percentage fall in $FEV_1$. Fifteen (15) minutes post-nebulization (41), 2 to 4 mL of expectorates were collected in sterile sputum containers. For all successful inductions, after a quality check, a sputum sample was collected in a sputum DNA collection, preservation, and isolation kit (Norgen Biotek Corporation, Canada; catalogue no. RU46100) as per manufacturer's instructions for downstream processing. Three participants failed induction due to tiny volume samples, which failed to pass the quality check, and were hence rejected.

**16S rRNA sequencing.** DNA was extracted from samples using a sputum DNA collection, preservation, and isolation kit (Norgen Biotek Corporation, Canada; catalogue no. RU46100). DNA purity and concentration were determined using both a spectrophotometer and a Qubit fluorescent-based kit. V3-V4 hypervariable region of the 16S rRNA gene was PCR amplified using 16S rRNA-specific 341F (5′ NNNNNNNNNNCCTACG GGNGGCWGCAG) and 785R (5′ NNNNNNNNNNGACTACHVGGGTATCTAAKCC) primers. All samples were tagged with unique 10 nucleotide sequences ("barcodes") incorporated into the forward primer. The PCRs included about 1 to 10 ng of DNA extract (total volume, 1 $\mu$L), 15 pmol of each forward primer and reverse primer in 20 $\mu$L of 1$\times$ MyTaq buffer containing 1.5 units MyTaq DNA polymerase (Bioline GmbH, Luckenwalde, Germany), and 2 $\mu$L of BioStabII PCR Enhancer (Sigma-Aldrich Co). PCRs were carried out for 30 cycles. DNA concentration of amplicons of interest was assessed by gel electrophoresis. About 20 ng amplicon DNA of each sample was pooled for up to 48 samples carrying different barcodes. The amplicon pools were purified with one volume of Agencourt AMPure XP beads (Beckman Coulter, Inc., IN, USA) to remove primer dimer and other small mispriming products, followed by an additional purification on MinElute columns (Qiagen GmbH, Hilden, Germany). About 100 ng of each purified amplicon pool DNA was used to construct Illumina libraries using the Ovation Rapid DR multiplex system 1-96 (NuGEN Technologies, Inc., CA, USA). Illumina libraries (Illumina, Inc., CA, USA) were pooled and size-selected by preparative gel electrophoresis. Bacterial 16S rRNA gene amplicons were sequenced targeting the V3-V4 (300-bp read length, paired-end protocol) region using the Illumina Miseq platform.

**Sequence processing and OTU classification.** The raw sequences were processed to remove potential human contamination (see Table S1 in the supplemental material). The human genome (https://www.ncbi.nlm.nih.gov/assembly/GCF_000001405.39) was masked with the ProGenomes2 microbial genome database (https://academic.oup.com/nar/advance-article/doi/10.1093/nar/gkz1002/5606617/). Raw reads were mapped to the human genome, discarded upon 95% identity, masked, and then filtered. Finally, we validated the human reads found by filtering out potential "human" contamination and aligned these against the NCBI nucleotide (nt) database, resulting in only top human hits. After removing human contamination, the remaining raw reads were processed using LotuS (1.62) (19). Poisson binomial model-based read filtering was applied (20). OTU clustering (UPARSE) (21) was based on sequence similarity of 97%, while SILVA (22) was incrementally used as a database for a taxonomic assignment using a Lambda taxonomic similarity search. The taxonomic classification (genus thresholded at 95% identity) was parsed using a custom Perl script, such that unassigned taxonomic levels were assigned to the last known

taxonomic level and sequentially numbered. Normalization and computation of alpha diversity measures were performed using the rarefaction tool kit (RTK 0.93.1) with default settings (42).

**Sample processing quality control.** All participants underwent sputum induction following a standard protocol to reduce sample variability. To avoid sample contamination, we used a single kit (Norgen Biotek sputum DNA isolation and preservation kit) that eliminated multiple steps of sputum processing and allowed for transportation at room temperature and inactivating microbial growth in the sample, hence preserving microbial composition. A stringent quality control check during sample collection was followed to reduce saliva and postnasal drip contamination. We also included negative controls (sputum kit with sterile water and buffer) during sample collection, DNA extraction, PCR amplification, and sequencing. Negative controls were negative for V3-V4 amplicons at PCR, and no sequences were generated.

**Outcomes.** Outcome variables included the following: (i) abundance of identified microbial operational taxonomic units (OTUs), (ii) derived abundance of microbial taxa broadly or narrowly associated with airway microbial communities, (iii) derived (projected) functional profiles of the airway microbiota, and (iv) derived ecosystem-level metrics (alpha- and beta-diversity measurements, clusterings of samples) stratified by HIV and COPD status.

**Exposure variables.** We included participant clinical and sociodemographic characteristics like age, sex, body mass index, use of wood for everyday cooking, smoking history (defined as never, occasional, or daily smoking), COPD (defined as postbronchodilator $FEV_1$/FVC ratio less than the fifth percentile of the National Health and Nutrition Examination Survey (NHANES) III African American reference population) (43), lung function profiles (defined as postbronchodilator $FEV_1$, FVC, and $FEV_1$/FVC ratio), airflow limitation classification, HIV serostatus, nadir $CD4^+$ T cell count (defined as the lowest CD4 count registered), viral load, use and duration of ART, co-trimoxazole use, duration of HIV since diagnosis at the clinic, and history of previous tuberculosis.

**Biostatistical methods. (i) Sample size calculation.** The sample size was estimated based on taxonomic-based human lung microbiome project (HMP) data (44). A conservative approximation of our power to detect microbiome associations at the genus level between groups was tested using a $t$ test. Given $n = 50$ individuals per group, assuming an effect size of 0.8 (Cohen's $d$) and an alpha threshold of 0.05/30, reflecting a Bonferroni correction for the 30 main airway bacterial genera, power was approximately 77% to conclude significance for each true microbial biomarker. Clinically relevant microbial feature abundance effects that fall within the convention of a "large" effect size Cohen ($d \geq 0.8$) are frequently used as starting points for translational medicine.

**(ii) Analysis plan.** Operational taxonomical unit (OTU) counts were rarefied to the smallest retained sample size (i.e., 8,278 raw reads) to obtain relative abundances of microbiota in each sample, accounting for read depths. Univariate analysis was done using metadeconfoundR (version 0.2.9) (18, 37), and relative abundances were tested for univariate associations with clinical variables, requiring a Benjamini-Hochberg adjusted FDR of <0.1 and the absence of any clear confounders such as age, sex, and body mass index (see Fig. S7 in the supplemental material). Only major taxa and OTUs detected after rarefaction in at least 10% of samples were used. Since the data were not normally distributed, nonparametric tests were used for all association tests. The Wilcoxon or the Kruskal-Wallis analysis of variance was used for discrete predictors. For pairs of continuous variables, a nonparametric Spearman correlation test was used. Benjamini-Hochberg false discovery rate control (FDR) was applied in all multiple testing situations requiring controlling the family-wise error rate at 10%. Hierarchical clustering was used to establish grouping patterns of the different study samples, including an updated adaptation of the approach used to define "enterotypes" in the human gut using the "Dirichlet multinomial" R package (version 1.36.0) (45). The chi-square test implemented in base R was used to test for significant differences in the resulting community type distribution between samples grouped by disease status. Beta diversity was calculated as Bray-Curtis dissimilarities as implemented in the vegan R (version 2.5-7) package (46). To determine the impact of participant clinical and sociodemographic characteristics on the taxonomic composition of the sputum microbiome, permutational multivariate analysis of variance (PERMANOVA) was performed. Bray-Curtis distances were used for all analyses. PERMANOVA test was performed using the adonis test, and pairwise multilevel comparison was conducted using the "pairwiseAdonis" package in R (version 0.4).

To project functional profiles from the composition of the airway microbiota assessed using 16S rRNA sequence data, PICRUSt2 (phylogenetic investigation of communities by reconstruction of unobserved states) (version 2.2.3) was used. PICRUSt2 uses marker gene data and a database of reference genomes, as well as the airway microbiota quantified using 16S rRNA sequences (47). PICRUSt2 was applied to all samples from Uganda and the United Kingdom and the merged UK-Ugandan data set. In our results, we present the results of the PICRUSt2 analysis from the Ugandan data set.

**Comparative analysis with the UK HIV sputum microbiome cohort.** The Uganda cohort results were compared to sputum microbiome data from a similar cohort in the United Kingdom. The UK sputum samples were collected from 64 PLWH (median blood CD4 count, 676 cells/$\mu$L) and 38 HIV-negative participants (16). United Kingdom inclusion criteria were age over 18 years, consent to participate, and absence of symptoms of acute respiratory illness at study entry. Sputum samples were collected from participants who could expectorate. The DNA was extracted using the automated DiaSorin lxt extraction platform combined with the DiaSorin Arrow DNA extraction kit. A sequencing library was created by amplifying the V3-V4 regions of the bacterial 16S rRNA. Sequencing was performed using the Illumina MiSeq platform. Raw reads were processed together with the Ugandan samples as mentioned above. We reclustered the merged data set using the Dirichlet multinomial package and projected the genus-level sputum microbiome composition of all samples (United Kingdom and Uganda) using principal coordinates analysis (PCoA) of Bray-Curtis dissimilarity of samples based on their predicted community type, geographical origin, and HIV status.

**Data availability.** The data can be accessed using the following data-specific identifiers: accession number https://www.ncbi.nlm.nih.gov/sra/PRJNA726058.

The link to the git repo is as follows: https://github.com/Theda-sys/Sputum_HIV_COPD_Cohort.

## SUPPLEMENTAL MATERIAL

Supplemental material is available online only.

**SUPPLEMENTAL FILE 1**, PDF file, 1.7 MB.

## ACKNOWLEDGMENTS

Alex Kayongo has been funded by the project "Balamu - Else Kröner-Fresenius Center for NCD Care and Education in Uganda," funded by the Else Kröner-Fresenius Foundation, Germany (2016-HA44 and 2019-HA178). Trishul Siddharthan is supported by a Mentored Career Development Award through the National Heart, Lung, and Blood Institute of the NIH (K23HL146946).

This study was funded by the GlaxoSmithKline (GSK) Africa NCD Open Lab grant (project number 8636). The funder provided in-kind scientific and statistical support in the study design but had no role in data collection, analysis, or decision to publish. Authors retained control of the final content of the publication.

All authors declare no conflict of interest except Felix Knauf who reports grants from Oxalosis and Hyperoxaluria Foundation and Deutsche Forschungsgemeinschaft during the conduct of the study; personal fees from Allena Pharmaceuticals, Oxthera Pharmaceuticals, Sanofi Pharmaceuticals, Fresenius Medical Care, Alnylam Pharmaceuticals, and Advicenne outside the submitted work.

A.K. and T.S.: Conceptualization, Funding Acquisition, Investigation, Methodology, Project Administration, Original Draft Writing and Editing; R.M. and D.M.: Investigation; T.U.P.B., L.M., U.L., and T.B.: Data Curation and Formal Analysis; D.J. and B.S.B.: Conceptualization, Investigation, Design, Funding Acquisition and Drafting the Manuscript for Important Intellectual Content; R.K.: Investigation, Methodology and Project Administration; P.A., B.K., E.K., C.A., E.U., and R.S.: Investigation, Methodology and Project Administration; F.K.: Draft Review and Editing; M.L.J., N.K.S., W.C., and S.K.F.: Conceptualization, Design and Drafting the Manuscript for Important Intellectual Content.

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
