## [Reviewer comments · Microbiology Spectrum]

Microbiology Spectrum

Sputum microbiome and Chronic Obstructive Pulmonary Disease in a rural Ugandan cohort of well-controlled HIV infection

Alex Kayongo, Theda Bartolomaeus, Till Birkner, Lajos Markó, Ulrike Löber, Edgar Kigozi, Carolyne Atugonza, Richard Munana, Denis Mawanda, Rogers Sekibira, Esther Uwimaana, Patricia Alupo, Robert Kalyesubula, Trishul Siddharthan, Felix Knauf, Bernard Bagaya, David Kateete, Moses Joloba, Nelson Sewankambo, Daudi Jjingo, Bruce Kirenga, William Checkley, and Sofia Forslund

Corresponding Author(s): Sofia Forslund, Max Delbrück Center for Molecular Medicine

Review Timeline:

Submission Date:	November 3, 2021
Editorial Decision:	January 14, 2022
Revision Received:	May 17, 2022
Editorial Decision:	August 7, 2022
Revision Received:	November 7, 2022
Accepted:	January 23, 2023

Editor: Kevin Theis

Reviewer(s): Disclosure of reviewer identity is with reference to reviewer comments included in decision letter(s). The following individuals involved in review of your submission have agreed to reveal their identity: Monica Ticlla (Reviewer #2)

Transaction Report:

DOI: <https://doi.org/10.1128/spectrum.02139-21>

January 14, 2022

Dr. Alex Kayongo
Makerere University
Lung Institute
New Mulago Hospital Complex
Kampala, Uganda 256
Uganda

Re: Spectrum02139-21 (Sputum microbiome and COPD status in a rural cohort of Ugandan adults with HIV)

Dear Dr. Alex Kayongo:

Thank you for submitting your manuscript to Microbiology Spectrum. The manuscript has been reviewed by two experts in the field. Based on their comments, major revisions of the manuscript are required for publication. If you choose to revise and resubmit the manuscript, please provide (1) point-by-point responses to the issues raised by the reviewers as file type "Response to Reviewers," not in your cover letter, and (2) a PDF file that indicates the changes from the original submission (by highlighting or underlining the changes) as file type "Marked Up Manuscript - For Review Only". Please use this link to submit your revised manuscript - we strongly recommend that you submit your paper within the next 60 days or reach out to me. Detailed instructions on submitting your revised paper are below.

Link Not Available

Sincerely,

Kevin R. Theis

Journals Department
Reviewer comments:

Reviewer #1 (Comments for the Author):

Thank you for giving me the opportunity to review the paper by Kayongo et al titled "Sputum microbiome and COPD status in a rural cohort of Ugandan adults with HIV". In this paper, the authors compare four age, sex and smoking matched cohorts (n=50) of rural Ugandan individuals- People living with HIV (PLWH) and Chronic Obstructive Pulmonary Disease (COPD), PLWH, HIV-negative with COPD and HIV-negative without COPD, and also compared the microbiome results to a UK based PLWH microbiome cohort. Unfortunately, based on the data presented in its current form, it appears that the primary research question of comparing the microbiome of these four groups resulted in no significant differences being found. Whilst this may be biologically plausible due to the high proportion of individuals sampled who were virologically suppressed due to ART treatment (86% of PLWH) and also that 86% of COPD patients were classed as mild or moderate, it is also possible that this is due to the bioinformatics methods employed, notably what appears to be a lack of appropriate initial filtering and also rarefaction of the data. The remainder of the results presented, including the comparison with the UK based PLWH microbiome dataset and determination of antimicrobial resistance genes in the cohort, are not presented clearly and therefore the manuscripts

conclusions are hard to justify. Specifically, I have additional comments which could aid the authors in improving their manuscript.

Major Comments:

1. References 6&7 don't support the statement "Recent studies suggest that the airway microbiota drives chronic lung inflammation observed in COPD"; one reference is for a mouse model of IPF and the second for PLWH lung microbiome with no assessment of lung inflammation status. Please either find supporting references or rewrite this sentence.
2. In the final sentence of the introduction, I am unsure why the airway microbiome would improve strategies for COPD diagnosis as a proxy for biomarker-based diagnosis. COPD diagnosis is normally achieved by spirometry and clinical assessment of symptoms, supported where necessary by radiological/ CT findings. Please review what you mean by this sentence and how the data you present in this paper will have an impact on PLWH and COPD.
3. How easy was it to obtain sputum from non-COPD individuals? What proportion of each cohort failed to produce sputum of the desired quality? Was a clinically validated assessment of sputum quality used, and if so please reference it. The fact that it appears cohorts were defined after sputum collection is concerning- please ensure this limitation of only including a sub-cohort of subjectively graded sputum producing individuals in this study is discussed- how generalisable are the results of this microbiome study of thick and mucoid sputum producing individuals to the general COPD (and non COPD) population, considering not all COPD patients produce sputum and sputum production would be expected to be even less in non-COPD patients. It is not completely clear from Fig 1 at what point those individuals unable to produce sputum were excluded from the study population, please clarify.
4. No mention of the use of the UK PLWH microbiome cohort is made in the introduction nor methods of the main manuscript. Please add this detail to your manuscript instead of in the supplement. How did the clinical characteristics of the UK PLWH microbiome cohort compare to your PLWH cohort (some are mentioned in the discussion section)? Could these differences account for any differences in your microbiome results between these two patient groups? Were all Ugandan patients compared to the UK PLWH cohort or just the Ugandan PLWH cohort? Were the same sequencing approaches used in the two cohorts (e.g. same primers etc). If not, what impact could this have had on the results in Fig S7?
5. I have concerns over the bioinformatics in their current form:
 - a. Please review the bioinformatics methods used and how much detail is reported including considering providing a link to an online markdown script for reproducibility/ clarity. Please provide details of what versions of the taxonomic databases or computer packages were queried and used. Please provide reference for metadecomfoundR, vegan and pairwiseAdonis packages.
 - b. No negative controls data is presented, and it only becomes apparent that these were performed in the discussion, please amend.
 - c. Please clarify what filtering of the OTUs was carried out for potential contaminants. Did you remove reads identified as e.g. Eukaryota, Human and Cyanobacteria? Fig 2b shows an unknown chloroplast in the results which suggests no filtering of results was performed.
 - d. How many raw reads were initially obtained in total and for each sample? I am concerned a large proportion of your data could have been omitted due to inappropriate rarefaction/ quality filtering processes but this is impossible to determine currently
 - e. Is there any justification to investigating as many alpha diversity indices as you have?
 - f. Did any samples fail quality controls? "retained sample" implies some sample data was excluded.
 - g. It is now generally not recommended to rarefy microbiome data (see McMurdie & Holmes 2014 for details <https://journals.plos.org/ploscompbiol/article?id=10.1371/journal.pcbi.1003531>). Please consider what effect this has had on your results. You have also not stated the number of reads you rarefied to ("smallest retained sample"). Did you use rarefied data for alpha diversity calculations, again this is no longer recommended (<https://www.frontiersin.org/articles/10.3389/fmicb.2019.02407/full>). I am very concerned these data analysis steps will have impacted your results significantly.
6. Please provide further detail about how functional profiles (and why you carried it out) of the microbiome was determined in the main manuscripts introduction and methods. The method detail in the supplement is also very limited (1 sentence). Were all samples included in this analysis or just the 100 from HIV-positive patients? The discussion (which could also be expanded) seems to imply the data was from COPD patients. Are the UK-PLWH data also used for functional profiling?
7. Presentation of results:
 - a. Table 1 contains a lot of clinically relevant information, please consider revising for clarity and also performing statistical testing to identify where differences occur between groups. It may not be necessary to report all the spirometry parameters currently included (suggest limit it to FEV1 (L and %predicted), FVC and FEV1/FVC with the pre-BD spirometry reported in supplement if at all). Please revise the GOLD classification in line with current guideline (A,B,C,D). For some variables (eg Age, BMI, spirometry), it is best practice to report median and IQR not mean and 95%CI.
 - b. Please test and report statistical significance of the clinical characteristics in table 2
 - c. Please ensure labelling and order of parts of figures is correct compared to legends and is sequential to how the data is presented in the manuscript (eg Fig3C before Fig3B). Also, please ensure legends only give technical description of the figure instead of presenting results data- this is particularly noticeable in Fig 3.
 - d. Please be careful to avoid sweeping statements (e.g. "were significantly associated with") when presenting your data without stating the direction and magnitude of the associations.
 - e. The paragraph titled "Impact of participant characteristics on airway microbiome diversity" comprises of very vague statements leaving the reader to draw their own conclusions. Please be more specific than "several bacterial genera" in line 188, and presumably you mean alpha diversity (Shannon) not microbial changes in relation to Fig S3. Why was only Shannon shown in Fig S3? This whole paragraph needs removing or significantly revising.

f. You have a small section of results titled "Distinct airway bacterial genera are associated with COPD among PLHWA" which I feel could be expanded to provide further results of interest to readers by considering other factors which could impact the microbiome: Did you consider stratifying the microbiome based on virological suppression status or COPD severity, likewise use of cotrimoxazole (Septrin) or other prophylactic antibiotics? Were there different amounts of AMR genes (presented later in the manuscript) in the cohort based on stratification for cotrimoxazole use?

g. Are the axis labels correct in Fig S6? There are no Staphylococcus, Pseudomonas, Lactobacillus, Klebsiella and Bilophila showing compared to the manuscript text and HIV status and geo location are not clear labels- do you mean HIV+ and UK or HIV- and Uganda?

8. There is no evidence in your socio-demographics of recreational drug use in the Ugandan cohort yet you present this in the discussion as a potential reason for the differences in the microbiome between UK and Ugandan PLWH cohorts.

Minor Comments:

1. Recommend not abbreviating COPD in the title and add 'living' between 'adults' and 'with'.

2. In abstract, please consider rewording the sentence beginning "Airway microbiome is essential..." as the literature does not support the statement in its current form. We understand that a healthy (non-dysbiotic) airway microbiome is important marker for better long-term outcomes but there is not yet sufficient evidence about how "essential" the airway microbiome is for maintaining a healthy immune response.

3. Correct the abbreviation of rDNA for ribosomal RNA in abstract (and throughout manuscript).

4. Is the emergence of NCDs and increase in COPD mortality only in PLWH or in the general sub-Saharan population? Or are the emergence on NCDs and increase in COPD mortality greater in the PLWH population compared to the non-PLWH population? Please clarify in these sentences in the introduction: "This has resulted, however, in an emergence of non-communicable diseases (NCDs). For example, Sub-Saharan Africa, which has the highest density of PLWH, has experienced dramatic increases in mortality related to chronic obstructive pulmonary disease (COPD), a chronic respiratory illness characterized by progressive and irreversible decline in lung function."

5. Suggest changing "unlikely represents" in the sentence beginning "Although sputum microbiome..." to "may not fully represent".

6. In this sentence: "The number of raw reads retrieved from the HIV COPD- group was significantly higher compared to all HIV+, all COPD+ and both COPD/HIV+ groups (S1)", please remove the words "all" and "both" as these imply HIV+ would include an overlap of patients with and without COPD, and vice versa.

1. Were COPD patient samples collected when clinically stable (i.e. no exacerbation)? You have excluded antibiotic use 2 weeks prior to sampling but are all exacerbations in your COPD population treated or are some exacerbations untreated?

2. Please check all abbreviations have been defined on first use (eg MWU, IHK-IRR)

3. In the online supplement, the data should be split into methods and results- the UK PLWH comparison is combined. Please move the supplementary figures to the supplement.

4. Why were three C/D spiro samples excluded in Fig 1?

5. Why isn't the data in table 3 kept with the data in table 1?

6. Please avoid using the brand name for a drug (Septrin)

7. It is best practice to report P values themselves instead of using *, **, ***. Please also consider adding P values into the manuscript text rather than just stating "was significantly....".

8. Please review your group naming throughout the manuscript- you are using HIV- COPD- and control interchangeably in text/ figures, likewise for community type 1,2 and 3 vs Streptococcus, Prevotella and Mixed in Fig 3.

9. Fig S1 shows the comparison between HIV- COPD- and HIV+ as NS but the manuscript states this difference in read number was significant.

10. Why is abundance presented in Fig2a and relative abundance in Fig2b? Would a stacked bar graph in Fig 2b be clearer? Do these bars represent the average abundance/ relative abundance of the four patient cohorts or the total combined reads in each group?

11. In Fig3a legend, please specify what the ellipses represent.

12. I am interested to know how many (instead of %) patients from each of the four patient cohorts are represented in the three community types presented in Fig 3, rather than just HIV-/+ and COPD -/+.

13. I am unsure how Fig 5 shows well controlled HIV (instead of HIV+) is associated with the genera Atopobium, Actinomyces and Megasphaera (line176).

14. In the volcano plot legends (eg Fig 5, S4) please confirm whether "subject" should be plural throughout (eg "the subject suffering from both COPD and HIV...").

15. Add "could" between "counts" and "explain" on line 237

Reviewer #2 (Comments for the Author):

The manuscript by Kayongo et al., titled "Sputum microbiome and COPD status in a rural cohort of Ugandan adults with HIV", describe three community types whose distribution differed by HIV status but not by COPD status. Except for richness and Chao1, alpha-diversity metrics did not differ between the HIV+/COPD+ group and the control groups (HIV-/COPD-, HIV-/COPD+, HIV+/COPD-). Although factors such as sex, BMI, history of pulmonary tuberculosis, HIV status, and years of ART were associated with differences in overall microbial community composition, the sputum composition did not significantly differ in the compared groups. However, using an interesting yet-to-be-published statistical approach, named metadeconfoundR, the authors identified a decreased abundance of specific genera associated with HIV+COPD+ after accounting for confounding

effects of other variables. In general, this work is a valuable contribution to better understanding the COPD-associated airway microbiome in HIV patients from a setting with a high risk for HIV, such as Uganda. However, three aspects need clarification:

(1) The study design. The authors explained a case-control design where a case group (HIV+/COPD+) is matched to three control groups (HIV-/COPD-, HIV-/COPD+, HIV+/COPD-). The matching was performed based on the frequency distribution of values for three variables in the case group: age (4 categories), sex (2 categories), and smoking status (4 categories). Thus, it would give a total of 32 categories that need matching. It is unlikely to achieve this from a pool of "226 potential participants". I presume the pool of participants recruited between February 2018 and February 2020 was much larger, considering that they were part of population-based cohorts. In this sense, the description of the "Study design" is incomplete and needs to be slightly extended both in the Methods (or Supplementary material) and in Figure 1 (Flow diagram for participant screening and enrollment).

(2) The sputum induction and sample collection. The procedures for sputum induction are well described. The strength of the procedure is the three-step cleansing routine to minimize contamination with microbes from the upper respiratory tract. However, there are some aspects of the procedure that need clarification. Based on the description in the methods (Lines 308 to 321), I understand that after each 5-minute nebulizing period, an expectorate was collected. Only if the percentage fall in FEV1 was less than 20% (relative to the baseline before the induction procedure started), in addition to passing quality control check (basically evaluating consistency). This procedure introduces a technical variability, the time interval at which the sputum was collected. This variability is important because it has been shown that the composition of induced sputum varies depending on the time point during the duration of the procedure (Gershman et al., 1999). Do the authors have records of the time interval at which each sample was collected? When comparing the case and control groups, are there differences in the time interval? Also, are there any records of whether or not a sample was collected after a repeated induction procedure? This information will help identify technical biases that could contribute to the differences between groups.

(3) The statistical approach. The authors used an interesting/novel statistical procedure to find associations between characteristics of the sputum microbial community, relative abundances of individual taxa, and exposure variables; this approach emphasizes identifying associations after accounting for potential confounders. The approach is a bit cumbersome. The authors described it in a very technical way with no explanation of why this approach was preferred over others that take into account the compositional nature of the relative abundances (e.g. ALDEx2). To understand the approach and interpret the figures in the manuscript, I had to look at the two publications where the methodology was previously used (Bartolomaeus 2021, Forslund 2021). This is not ideal. The manuscript should be self-contained and provide the readers with an intuitive explanation of how the outcomes of the statistical methods must be interpreted. The manuscript would benefit from improving the description of the biostatistical methods and the legends of figures where the methodology is used (e.g. Figure 3C, Figure 5, S2, S4).

Additional comments follow. I used the number of the line and quoted some text.

Major:

Line 128 "... recruited from 226 potential participants". Based on the description of the study design (Line 281) and the legend for Figure 1 (Line 439), it is my impression that a larger pool of individuals was needed to be able to create the groups; matched by the frequency of age-, sex-, and smoking status values in the HIV+/COPD+ group. Please clarify, and update Figure 1 if needed to reflect the real pool of participants.

Line 129 "Fifty-nine percent of participants were male (59%), 43% were aged>55 years and 63% were non-smokers." Given that the groups shown in Table 1 were matched based on the frequency of the values of these variables in the HIV+/COPD+ group, it should be expected to observe similar frequencies in the other three groups. Can the authors explain why this is not the case?

Line 144, "... after demultiplexing and quality control filtering". The quality control filtering is described neither in the Methods nor in the Supplementary Material. I would briefly describe or add it to the Supplementary material.

Line 144, "OTU counts were rarefied to the size of the smallest retained sample." Please specify this size. This approach has the disadvantage of including low-quality samples that generally have fewer raw reads. An alternative approach could be to set the rarefaction threshold above the number of reads obtained in negative controls. Alternatively, set a threshold where a good coverage of the microbial community richness can be obtained. I would recommend plotting the rarefaction curves at different sampling sizes for each sample to assess how well the sputum communities were covered.

Line 147, "We accounted for these differences during further analysis using rarefaction toolkit for normalization." Rarefaction to an even sampling size does not remove the effect of differences in raw read counts. Raw read counts should be included in all the models to account for heterogeneous sequencing depth, especially if the HIV-/COPD- group has higher raw read counts.

Line 223, "COPD was also associated with a higher abundance of Staphylococcus and lower abundance of organisms belonging to the genera Pseudopropionibacterium, Porphyromonas, and Parvimonas." Here the paragraph does not discuss the lower abundance of Pseudopropionibacterium, Porphyromonas, and Parvimonas associated with COPD in the study's cohort, which is the opposite of what has been found in the previous studies mentioned in the same paragraph.

Line 253, "A stringent quality control check at the time of sample collection was followed to reduce on saliva and postnasal drip contamination." The methods section only mentions checking for consistency (mucoïd), were there other qualitative/quantitative features assessed (e.g. volume, color, presence of blood)?

Line 254, "We also included negative controls (sputum kit with sterile water and buffer) during sample collection." This is a strength of the work presented. However, it is not mentioned how the negative controls were used to identify contaminated samples or the presence of contaminants in the community profiles.

Line 315, "... and the induction procedure repeated." Is there a record of the number of attempts the induction procedure was performed? When repeated, was it performed just after the previous attempt? Would this introduce certain biases in the sampling of the sputum microbial community?

Line 374, "The code is available upon request". The code should be available as supplementary material.

Minor:

Line 57, the authors stated, "we show that among PLWH, airway enrichment with *Staphylococcus* spp as well as depletion of *Pseudopropionibacterium* and *Porphyromonas* spp are associated with COPD." However, based on the results presented, *Staphylococcus* was associated with COPD status (including HIV- participants) but not with COPD-HIV status. Thus the statement needs to be rephrased.

Line 132 "... 86% virologically suppressed with a median viral load of <20 copies/ml". This seems to conflict with what is shown in Table 2. Only 14% and 8% of HIV+ participants with COPD+ and COPD-, respectively, had a viral load of <20 copies/ml. Please clarify.

Line 146, "The number of raw reads retrieved from the HIV-COPD- group was significantly higher compared to all HIV+, all COPD+ and both COPD/HIV+ groups (S1)." Notice that the raw read count of the HIV-COPD- is not significantly higher than in the HIV+.

Line 167, "Microbial richness was significantly lower among COPD+/HIV+ group compared with other groups (Figure 4A)." Notice that mean microbial richness in COPD+/HIV+ does not seem to be lower than in COPD-/HIV- in Figure 4A. Please add the actual mean values to the legend in Figure 4A.

Line 213, "Nevertheless, 16S raw reads and microbial richness was reduced among PLWH with COPD despite HIV control". See previous comment.

Line 241, "Furthermore, in our study, antimicrobial resistance genes reflect a potential multidrug resistome reservoir among COPD individuals". Considering that *Staphylococcus* is enriched in COPD participants, could *Staphylococcus* be the primary driver of the antimicrobial resistance genes associated with COPD?

Line 324, "Bacterial 16S rRNA V3 region was amplified and ..." Notice that the supplementary material mentions the V3-V4 hypervariable region. Please clarify.

Line 445, the legend of Figure 2B: "Abundances of microbiome at genus level stratified by COPD and HIV." are the displayed genera representing the top 20 most abundant ones? *Streptococcus* is not displayed, but it should be expected to be among the most abundant genera.

Line 450, Figure 3A: for consistency, please display relative abundances either as percentages, fractions, or rarefied counts in all figures.

Line 476, Legend of Figure 4: The panel's figures should be shown in the order they appear in the text. It seems the letters of the sub-figures were swapped.

Line 481, Please display the amount of variance explained by PCo1 and PCo2.

Line 567, correct title. it should be ST1

Line 570, correct title. it should be ST2

Staff Comments:

Preparing Revision Guidelines

Please return the manuscript within 60 days; if you cannot complete the modification within this time period, please contact me. If you do not wish to modify the manuscript and prefer to submit it to another journal, please notify me of your decision immediately so that the manuscript may be formally withdrawn from consideration by Microbiology Spectrum.

Sputum microbiome and COPD status in a rural cohort of Ugandan adults with HIV

Alex Kayongo* 1-2 , Trishul Siddharthan 3,4, Theda Ulrike Patricia Bartolomaeus 5,6,7,8 , Till Birkner 5,6,7 , Lajos Markó 5,6,7,8,9 , Ulrike Löber 5,6,7,8 , Edgar Kigozi 2 , Carolyne Atugonza 2 , Richard Munana 10 , Denis Mawanda 1 , Rogers Sekibira 1 , Esther Uwimaana 1,2 , Patricia Alupo 1 , Robert Kalyesubula 10,11 , Felix Knauf 12 , Bernard S Bagaya 2 , David P Kateete 2 , Moses L Joloba 2 , Nelson K Sewankambo 11 , Daudi Jjingo 13,14 , Bruce Kirenga 1,12 , William Checkley 3,4 and Sofia K. Forslund 5,6,7,8,9,15

The manuscript by Kayongo et al., titled “**Sputum microbiome and COPD status in a rural cohort of Ugandan adults with HIV**”, describe three community types whose distribution differed by HIV status but not by COPD status. Except for richness and Chao1, alpha-diversity metrics did not differ between the HIV+/COPD+ group and the control groups (HIV-/COPD-, HIV-/COPD+, HIV+/COPD-). Although factors such as sex, BMI, history of pulmonary tuberculosis, HIV status, and years of ART were associated with differences in overall microbial community composition, the sputum composition did not significantly differ in the compared groups. However, using an interesting yet-to-be-published statistical approach, named metadeconfoundR, the authors identified a decreased abundance of specific genera associated with HIV+COPD+ after accounting for confounding effects of other variables. In general, this work is a valuable contribution to better understanding the COPD-associated airway microbiome in HIV patients from a setting with a high risk for HIV, such as Uganda. However, three aspects need clarification:

(1) The study design. The authors explained a case-control design where a case group (HIV+COPD+) is matched to three control groups (HIV-/COPD-, HIV-/COPD+, HIV+/COPD-). The matching was performed based on the frequency distribution of values for three variables in the case group: age (4 categories), sex (2 categories), and smoking status (4 categories). Thus. It would give a total of 32 categories that need matching. It is unlikely to achieve this from a pool of “226 potential participants”. I presume the pool of participants recruited between February 2018 and February 2020 was much larger, considering that they were part of population-based cohorts. In this sense, the description of the “Study design” is incomplete and needs to be slightly extended both in the Methods (or Supplementary material) and in Figure 1 (Flow diagram for participant screening and enrollment).

(2) The sputum induction and sample collection. The procedures for sputum induction are well described. The strength of the procedure is the three-step cleansing routine to minimize contamination with microbes from the upper respiratory tract. However, there are some aspects of the procedure that need clarification. Based on the description in the methods (Lines 308 to 321), I understand that after each 5-minute nebulizing period, an expectorate was collected. Only if the percentage fall in FEV1 was less than 20% (relative to the baseline before the induction procedure started), in addition to passing quality control check (basically evaluating consistency). This procedure introduces a technical variability, the time interval at which the sputum was collected. This variability is important because it has been shown that the composition of induced sputum varies depending on the time point during the duration of the procedure (Gershman et al., 1999). Do the authors have records of the time interval at which each sample was collected? When comparing the case and control groups, are there differences in the time interval? Also, are there any records of whether or not a sample was collected after a repeated induction procedure? This information will help identify technical biases that could contribute to the differences between groups.

(3) The statistical approach. The authors used an interesting/novel statistical procedure to find associations between characteristics of the sputum microbial community, relative abundances of individual taxa, and exposure variables; this approach emphasizes identifying associations after accounting for potential confounders. The approach is a bit cumbersome. The authors described it in a very technical way with no explanation of why this approach was preferred over others that take into account the compositional nature of the relative abundances (e.g ALDEx2). To understand the approach and interpret the figures in the manuscript, I had to look at the two publications where the methodology was previously used (Bartolomaeus 2021, Forslund 2021). This is not ideal. The manuscript should be self-contained and provide the readers with an intuitive explanation of how the outcomes of the statistical methods must be interpreted. The manuscript would benefit from improving the description of the biostatistical methods and the legends of figures where the methodology is used (e.g, Figure 3C, Figure 5, S2, S4).

Additional comments follow. I used the number of the line and quoted some text.

Major:

Line 128 "... recruited from 226 potential participants". Based on the description of the study design (Line 281) and the legend for Figure 1 (Line 439), it is my impression that a larger pool of individuals was needed to be able to create the groups; matched by the frequency of age-, sex-, and smoking status values in the HIV+/COPD+ group. Please clarify, and update Figure 1 if needed to reflect the real pool of participants.

Line 129 "Fifty-nine percent of participants were male (59%), 43% were aged>55 years and 63% were non-smokers." Given that the groups shown in Table 1 were matched based on the frequency of the values of these variables in the HIV+COPD+ group, it should be expected to observe similar frequencies in the other three groups. Can the authors explain why this is not the case?

Line 144, "... after demultiplexing and quality control filtering". The quality control filtering is described neither in the Methods nor in the Supplementary Material. I would briefly describe or add it to the Supplementary material.

Line 144, "OTU counts were rarefied to the size of the smallest retained sample." Please specify this size. This approach has the disadvantage of including low-quality samples that generally have fewer raw reads. An alternative approach could be to set the rarefaction threshold above the number of reads obtained in negative controls. Alternatively, set a threshold where a good coverage of the microbial community richness can be obtained. I would recommend plotting the rarefaction curves at different sampling sizes for each sample to assess how well the sputum communities were covered.

Line 147, "We accounted for these differences during further analysis using rarefaction toolkit for normalization."

Rarefaction to an even sampling size does not remove the effect of differences in raw read counts. Raw read counts should be included in all the models to account for heterogeneous sequencing depth, especially if the HIV-/COPD- group has higher raw read counts.

Line 223, “COPD was also associated with a higher abundance of Staphylococcus and lower abundance of organisms belonging to the genera Pseudopropionibacterium, Porphyromonas, and Parvimonas.” Here the paragraph does not discuss the lower abundance of Pseudopropionibacterium, Porphyromonas, and Parvimonas associated with COPD in the study’s cohort, which is the opposite of what has been found in the previous studies mentioned in the same paragraph.

Line 253, “A stringent quality control check at the time of sample collection was followed to reduce on saliva and postnasal drip contamination.” The methods section only mentions checking for consistency (mucoid), were there other qualitative/quantitative features assessed (e.g. volume, color, presence of blood)?

Line 254, “We also included negative controls (sputum kit with sterile water and buffer) during sample collection.” This is a strength of the work presented. However, it is not mentioned how the negative controls were used to identify contaminated samples or the presence of contaminants in the community profiles.

Line 315, “... and the induction procedure repeated.” Is there a record of the number of attempts the induction procedure was performed? When repeated, was it performed just after the previous attempt? Would this introduce certain biases in the sampling of the sputum microbial community?

Line 374, “The code is available upon request”. The code should be available as supplementary material.

Minor:

Line 57, the authors stated, “we show that among PLWH, airway enrichment with Staphylococcus spp as well as depletion of Pseudopropionibacterium and Porphyromonas spp are associated with COPD.” However, based on the results presented, Staphylococcus was associated with COPD status (including HIV- participants) but not with COPD-HIV status. Thus the statement needs to be rephrased.

Line 132 “... 86% virologically suppressed with a median viral load of <20 copies/ml”. This seems to conflict with what is shown in Table 2. Only 14% and 8% of HIV+ participants with COPD+ and COPD-, respectively, had a viral load of <20 copies/ml. Please clarify.

Line 146, “The number of raw reads retrieved from the HIV-COPD- group was significantly higher compared to all HIV+, all COPD+ and both COPD/HIV+ groups (S1).” Notice that the raw read count of the HIV-COPD- is not significantly higher than in the HIV+.

Line 167, “Microbial richness was significantly lower among COPD+/HIV+ group compared with other groups (Figure 4A).” Notice that mean microbial richness in COPD+/HIV+ does not seem to be lower than in COPD-/HIV- in Figure 4A. Please add the actual mean values to the legend in Figure 4A.

Line 213, “Nevertheless, 16S raw reads and microbial richness was reduced among PLWH with COPD despite HIV control”. See previous comment.

Line 241, “Furthermore, in our study, antimicrobial resistance genes reflect a potential multidrug resistome reservoir among COPD individuals”. Considering that Staphylococcus is enriched in COPD participants, could Staphylococcus be the primary driver of the antimicrobial resistance genes associated with COPD?

Line 324, “Bacterial 16S rRNA V3 region was amplified and ...” Notice that the supplementary material mentions the V3-V4 hypervariable region. Please clarify.

Line 445, the legend of Figure 2B: “Abundances of microbiome at genus level stratified by COPD and HIV.” are the displayed genera representing the top 20 most abundant ones? Streptococcus is not displayed, but It should be expected to be among the most abundant genera.

Line 450, Figure 3A: for consistency, please display relative abundances either as percentages, fractions, or rarefied counts in all figures.

Line 476, Legend of Figure 4: The panel's figures should be shown in the order they appear in the text. It seems the letters of the sub-figures were swapped.

Line 481, Please display the amount of variance explained by PCo1 and PCo2.

Line 567, correct title. it should be ST1

Line 570, correct title. it should be ST2

Response to reviewer's comments

We would like to thank the reviewers for their time and dedication to provide detailed feedback concerning our manuscript. We have taken ample time to critically consider and work on the reviewers' suggestions and address their concerns below in a point-by-point response to the comments raised.

Reviewer #1 (Comments for the Author):

Comment 1: *Thank you for giving me the opportunity to review the paper by Kayongo et al titled "Sputum microbiome and COPD status in a rural cohort of Ugandan adults with HIV". In this paper, the authors compare four age, sex and smoking matched cohorts (n=50) of rural Ugandan individuals- People living with HIV (PLWH) and Chronic Obstructive Pulmonary Disease (COPD), PLWH, HIV-negative with COPD and HIV-negative without COPD, and also compared the microbiome results to a UK based PLWH microbiome cohort. Unfortunately, based on the data presented in its current form, it appears that the primary research question of comparing the microbiome of these four groups resulted in no significant differences being found.*

Response: We thank the reviewers for pointing this out. We have carefully considered the design and clarify it in Figure 1 of the revised manuscript. In the current study, a comparison of two groups i.e HIV+/- and COPD+/- each with a control for orthogonal stratification were performed (i.e. all HIV+ samples are compared to all HIV- samples adjusting for COPD status and vice versa). We found significant differences (specifically COPD, HIV and combined HIV-COPD effects inferred from the interaction term), which we have now clearly described in the revised manuscript.

Comment 2: *Whilst this may be biologically plausible due to the high proportion of individuals sampled who were virologically suppressed due to ART treatment (86% of PLWH) and also that 86% of COPD patients were classed as mild or moderate, it is also possible that this is due to the bioinformatics methods employed, notably what appears to be a lack of appropriate initial filtering and also rarefaction of the data.*

Response: We thank the reviewers for this insight. We have performed the required filtering and decontamination of the data as now outlined in the revised materials and methods section. Specifically, we now perform additional filtering for spurious human read matching, and we have implemented a method to control for sample depth that does not depend on rarefaction.

"The raw sequences obtained were processed to remove potential human contamination. The human genome (https://www.ncbi.nlm.nih.gov/assembly/GCF_000001405.39/) was masked with the progenome 2 database <https://academic.oup.com/nar/advance-article/doi/10.1093/nar/gkz1002/5606617/>. Raw reads were mapped to the masked human genome and discarded upon 95% identity. Finally, we validated the human reads found by filtering

“human” contamination and aligned these against the NCBI-database, resulting in only human top hits.”

We thank the reviewer for this suggestion. We implemented additional functionality in metadeconfoundR (updating the package accordingly to include this as a user option) to include the raw read count as an additional covariate into all GLMs used in the analysis. Comparing this approach to our previous approach showed minimal differences in the resulting inferences, though we agree this may have greater impact in another dataset.

Impact of disease status, medication, and other collected metadata variables on taxonomic composition of the sputum microbiome. Heatmap shows all genus-level taxa significantly [MWU (for categorical factors) and Spearman (for continuous features) *FDR<0.1, **FDR<0.01, ***FDR<0.001] different in abundance (binned rarefied 16S gene counts) depending on disease status (HIV/COPD) alongside participant characteristics. Heatmap cells show effect size (Cliff's Delta for binary factors, Spearman's Rho for continuous features). Parallel post-hoc testing for all possible confounders was applied (using nested linear model comparisons and including total read count to account for heterogeneous sequencing depth), for each cell showing no stars or circles if the association was not significant (NS) in the initial naïve test step. In the remaining naïvely significant associations, only those additionally passing the deconfounding post-

hoc testing step as being strictly deconfounded (SD) or laxly deconfounded (LD), or having no other significant covariates (NC) are shown as black stars, while any confounded signal is shown as a grey circle.

Comment 3: *The remainder of the results presented, including the comparison with the UK based PLWH microbiome dataset and determination of antimicrobial resistance genes in the cohort, are not presented clearly and therefore the manuscripts conclusions are hard to justify.*

Response: We thank the reviewer for this comment, we have now included details of the UK dataset in the revised methods and results sections. Re-analysis of our data with PiCRUST2 following filtering and decontamination revealed no disease-associated antimicrobial genes, therefore, we removed this section.

Others Comments from reviewer 1:

Specifically, I have additional comments which could aid the authors in improving their manuscript.

Major Comments

Comment 1. *References 6&7 don't support the statement "Recent studies suggest that the airway microbiota drives chronic lung inflammation observed in COPD"; one reference is for a mouse model of IPF and the second for PLWH lung microbiome with no assessment of lung inflammation status. Please either find supporting references or rewrite this sentence.*

Response: We agree with the reviewer, and have included relevant additional references (8,9 and 10) in the revised manuscript as outlined below.

8. Rigauts C, Aizawa J, Taylor S, Rogers GB, Govaerts M, Cos P, et al. Rothia mucilaginosa is an anti-inflammatory bacterium in the respiratory tract of patients with chronic lung disease. European Respiratory Journal. 2021.

9. Segal LN, Clemente JC, Tsay J-CJ, Koralov SB, Keller BC, Wu BG, et al. Enrichment of the lung microbiome with oral taxa is associated with lung inflammation of a Th17 phenotype. Nature microbiology. 2016;1(5):1-11.

10. Wu BG, Sulaiman I, Tsay J-CJ, Perez L, Franca B, Li Y, et al. Episodic aspiration with oral commensals induces a MyD88-dependent, pulmonary T-helper cell type 17 response that mitigates susceptibility to Streptococcus pneumoniae. American Journal of Respiratory and Critical Care Medicine. 2021;203(9):1099-111.

Comment 2. *In the final sentence of the introduction, I am unsure why the airway microbiome would improve strategies for COPD diagnosis as a proxy for biomarker-based diagnosis. COPD diagnosis is normally achieved by spirometry and clinical assessment of symptoms, supported where necessary by radiological/ CT findings. Please review what you mean by this sentence and how the data you present in this paper will have an impact on PLWH and COPD.*

Response: We agree with the reviewer, and in the revised manuscript have edited this sentence to read as follows “A better understanding of the lung microbiome among PLWH could improve COPD prognostic and risk stratification strategies in HIV”.

Comment 3a. How easy was it to obtain sputum from non-COPD individuals?

Response: We did not find any difficulty obtaining induced sputum from non-COPD participants. Following nebulisation with 3% hypertonic saline, we successfully induced sputum samples from all non-COPD participants described.

Comment 3b: What proportion of each cohort failed to produce sputum of the desired quality?

Response:

Group	Failed quality sputum induction	Percentage
HIV+COPD+	0	0%
HIV+COPD-	3	5.7%
HIV-COPD+	0	0%
HIV-COPD-	0	0%

Comment 3c: Was a clinically validated assessment of sputum quality used, and if so please reference it.

Response: We thank the reviewer for this comment. Yes, a clinically validated sputum quality assessment was used. Upon deep coughing and expectoration, each sputum sample was assessed for mucoid consistency and Gram’s stain procedure performed for quality assessment. Sputum samples with less than 10 squamous epithelial cells and more than 25 polymorphonuclear cells per low-power field (x10) microscopy (indicative of a lower airway sample) passed a quality control check. Otherwise, the sample was rejected, and the induction procedure repeated. As noted above only three samples failed the quality control check. This information has now been included in the revised methods section.

Reference

Loens K, Van Heirstraeten L, Malhotra-Kumar S, Goossens H, Ieven M. Optimal sampling sites and methods for detection of pathogens possibly causing community-acquired lower respiratory tract infections. *Journal of clinical microbiology*. 2009;47(1):21-31.

Comment 3d: *The fact that it appears cohorts were defined after sputum collection is concerning- please ensure this limitation of only including a sub-cohort of subjectively graded sputum producing individuals in this study is discussed*

Response: We thank the reviewers for this observation. We have clarified the design in the revised Figure 1 and manuscript text, more clearly illustrating how the cohorts were defined during screening for COPD and HIV. Briefly, participants were recruited from two independent cohorts in the same geographic location. The first cohort screened for COPD among 656 HIV-negative individuals in rural Nakaseke communities while the second cohort screened for COPD among 722 HIV-infected individuals attending four HIV treatment centers within Nakaseke district (5, 16). Figure 1 illustrates participant screening and enrolment from the two cohorts. We randomly selected 50 HIV-positive individuals diagnosed with COPD, 50 HIV-positive individuals without COPD, 50 HIV-negative individuals with COPD, and 50 HIV-negative individuals without COPD (total N = 200). Participants were eligible for inclusion if they resided within Nakaseke district, were ≥35 years of age, had confirmed HIV serostatus and spirometry-based COPD status at enrollment, were capable of understanding the study procedures, underwent successful sputum induction and did not have contraindications for spirometry or sputum induction procedure.

Comment 3e: *- how generalisable are the results of this microbiome study of thick and mucoid sputum producing individuals to the general COPD (and non COPD) population, considering not all COPD patients produce sputum and sputum production would be expected to be even less in non-COPD patients.*

Response: We thank the reviewers for this comment. We performed sputum induction using 3% hypertonic saline for all study participants following standard protocol. We consider our results broadly generalizable to cohorts which use sputum induction as a sample collection method. Sputum induction ensures that a sputum sample is collected in the same way from all participants irrespective of whether they have COPD or not.

Comment 3f: *It is not completely clear from Fig 1 at what point those individuals unable to produce sputum were excluded from the study population, please clarify.*

Response: We have clarified this in the revised Figure 1. If a participant failed to produce induced sputum, we excluded that participant. Our final analytical dataset included all individuals who underwent successful sputum induction. It should be noted however that we registered high success rates in sputum induction with only 3 participants excluded based on failure to produce high quality sputum.

Comment 4. *No mention of the use of the UK PLWH microbiome cohort is made in the introduction or methods of the main manuscript. Please add this detail to your manuscript instead of in the supplement.*

Response: We agree with the reviewer, information about the UK microbiome cohort has been added to the results and methods sections in the revised manuscript.

How did the clinical characteristics of the UK PLWH microbiome cohort compare to your PLWH cohort (some are mentioned in the discussion section)?

Response: This information has been included in the revised results section as follows. To compare our results with a similar cohort, we considered the United Kingdom (UK)-based HIV sputum microbiome cohort (13). In this cohort, sputum samples were collected from 64 HIV-infected individuals with a median CD4 count of 676 cells/ μ L, comparably higher than in our cohort with a median nadir CD4+T cell count of 330 cells/ mm^3 (IQR 167-544 cells/ mm^3) and 383 cells/ mm^3 (IQR 222-520 cells/ mm^3) among COPD-positive and -negative individuals respectively. In addition, The UK cohort recruited 38 HIV-negative individuals. Above 80% of HIV-infected individuals in the UK cohort were virologically suppressed with viral loads below 40 copies per mL of blood which was comparable to our cohort with over 86% well controlled with viral load below detectable level. No significant differences were reported between HIV-infected and -negative groups regarding age, sex, educational level, body mass index (BMI), and co-morbidities. In the Ugandan cohort, however, we noted some differences. Among the HIV-negative group, participants with COPD were significantly older ($p < 0.001$), with lower body mass index (BMI) ($p = 0.001$), extensive use of respiratory medication (antibiotics, prednisone and salbutamol) ($p < 0.001$) and predominantly non-smoker ($p = 0.017$) compared to COPD negative individuals. In the UK cohort, current tobacco smoking, and recent use of recreational drugs were significantly higher in HIV-infected individuals. The spirometric patterns were normal for most participants with only 10 HIV-infected and 2 HIV-negative participants reported with COPD (defined as $\text{FEV}_1/\text{FVC} < 0.7$). All participants were free of symptoms of acute respiratory illness at the time of recruitment (16).

Could these differences account for any differences in your microbiome results between these two patient groups?

Response: Yes, we agree with the reviewer, these differences potentially accounted for the observed differences in the sputum microbiome between the UK and Ugandan cohort. Using the metadecomfoundR tool for tracing potential confounding influences in biomarker inference, we noted differences in the microbiome between the two cohorts which in some cases does, in others does not, reduce to these demographic and clinical differences. These findings are described in the revised results section. We have also discussed these findings in the revised discussion section.

Comment: *Were all Ugandan patients compared to the UK PLWH cohort or just the Ugandan PLWH cohort?*

Response: We compared all Ugandan patients to the UK PLWH cohort.

Comment: Were the same sequencing approaches used in the two cohorts (e.g. same primers etc). If not, what impact could this have had on the results in Fig S7?

Response: DNA extraction in the Ugandan and UK cohort was done with different extractions kits, and we have previously shown that DNA extraction method has a strong effect on inferred microbial community structure.

<https://academic.oup.com/circovasces/article/117/3/863/5831292?login=true>

Sequencing was done using the V3-V4 hypervariable regions of the 16S rRNA gene using the Illumina sequencing platform in both cohorts, additionally the computational (re-)processing of the samples was done together for the pooled dataset. The new figure (S6 now) shows the differences between the cohorts still present, which we note may reflect differences in sample extraction methodology. As seen in Figure S6C, we can see that there are clear differences between the relative abundance of gram-positives (*Firmicutes* and *Actinobacteria*) between the cohorts, which is in line with our expectations from the differences in extraction protocols (Bartolomaeus et al., 2020). Overall, this is interesting since it points out how careful one must be when comparing datasets, which is in agreement with the reviewer's comments. However, as the variable we test for (HIV status) is balanced in both datasets separately, we are able to conduct stratified analysis which, in principle, allows us to circumvent this bias for the specific purpose we undertook the data comparison. We thank you for bringing this issue to our attention. This has been included in our discussion section as factors potentially accounting for differences in overall microbiome results between UK and Ugandan cohorts.

Furthermore, we added a section to the revised manuscript describing the methods used in the UK cohort as follows

"We compared our results with results from a HIV UK cohort. The UK study sequenced sputum samples collected from 64 PLW-HIV (median blood CD4 count 676 cells/ μ L) and 38 HIV-negative participants. (13). UK inclusion criteria were age over 18 years, consent to participate, and absence of symptoms of acute respiratory illness at study entry. Sputum samples were collected from participants who could expectorate. The DNA was extracted using the automated DiaSorin® Ixt extraction platform combined with the DiaSorin® Arrow DNA extraction kit. A sequence library was created by amplification of V3-V4 regions of the bacterial 16S rRNA. Sequencing was performed using the Illumina MiSeq Platform. Raw reads were processed together with the Ugandan samples as mentioned above."

Figure: Impact of cohort (UK vs. Uganda) and clustered community type on taxonomic composition of the sputum microbiome. A)

Principal coordinates analysis (PCoA) plots of Bray–Curtis dissimilarity of samples colored according to their predicted community type, geographical origin, and HIV status (0 = HIV-negative, 1 = HIV-positive). Comparing the geographical origin ($n = 302$ samples in total, $n = 200$ from Uganda and $n = 102$ from UK) reveals significant impact on both PCo1 and PCo2 dimensions (PERMANOVA $p = 0.001$). **B)** Alpha diversity (Shannon index) the sputum microbiota stratified for geolocation. Significant differences between the samples from the Uganda cohort versus the UK cohort (pairwise Wilcox test corrected for multiple testing using FDR). **C)** Heatmap shows all phylum-level taxa significantly [MWU (for categorical factors) and Spearman (for continuous features) * $FDR < 0.1$, ** $FDR < 0.01$, *** $FDR < 0.001$] different in abundance (binned rarefied 16S gene counts) depending on cohort alongside HIV status. Heatmap cells show effect size (Cliff's Delta for categorical factors, Spearman's Rho for continuous features). Multi-confounder testing (nested linear model testing, post hoc test) was applied showing no stars or circles if not significant (NS) in naive test step. In the remaining naive-significant associations, only those passing the deconfounding step as strictly deconfounded (SD), laxly deconfounded (LD) or no other covariates (NC) are black star, while any confounded signal is grey circle.

Comment 5. *I have concerns over the bioinformatics in their current form:*

a. Please review the bioinformatics methods used and how much detail is reported including

Responses: **We agree with the reviewers, we have in the revised manuscript expanded the methods section to clearly show what we have done, as follows:**

“Operational taxonomical unit (OTU) counts were rarefied to the smallest retained sample size (i.e 8278 raw reads) to obtain relative abundances of microbiota in each sample, accounting for read depths. Univariate analysis was done using metadeconfoundR (v 0.2.9)(45), relative abundances were tested for univariate associations with clinical variables, requiring Benjamini-Hochberg adjusted FDR < 0.1 and the absence of any clear confounders such as age, sex and body mass index. Only major taxa and OTUs detected after rarefaction in at least 10% of samples were used. Since the data was not normally distributed, non-parametric tests were used for all association tests. The Wilcoxon or the Kruskal-Wallis analysis of variance were used for discrete predictors. For pairs of continuous variables, a non-parametric Spearman correlation test was used. Benjamini-Hochberg False Discovery Rate control (FDR) was applied in all multiple testing situations requiring controlling the family-wise error rate at 10%. Hierarchical clustering was used to establish grouping patterns of the different study samples, including an updated adaptation of the approach used to define “enterotypes” in the human gut using the ‘Dirichlet Multinomial’ R package (v 1.36.0)(46). The chi-square test implemented in base R was used to test for significant differences in the resulting community type distribution between samples grouped by disease status. Beta diversity was calculated as Bray-Curtis dissimilarities as implemented in the vegan R (v 2.5-7) package (47). To determine the impact of participant clinical and sociodemographic characteristics on taxonomic composition of the sputum microbiome, permutational multivariate analysis of variance (PERMANOVA) was performed. Bray-Curtis distances were used for all analyses. PERMANOVA test was performed using the adonis test and pairwise multilevel comparison was conducted using the 'pairwiseAdonis' package in R (v 0.4)(48). To determine the projected functional profiles of the sputum microbiota using 16S rRNA sequence data, PICRUST (phylogenetic investigation of communities by reconstruction of unobserved states)(version 2.2.3) was used to infer the functional profiles of the bacterial community. “

Comment: *considering providing a link to an online markdown script for reproducibility/ clarity.*

We thank the reviewer for the suggestion and prepared a GitHub repository(https://github.com/Theda-sys/Sputum_HIV_COPD_Cohort) containing both markdown scripts and their compiled version in html format.

Comment: *Please provide details of what versions of the taxonomic databases or computer packages were queried and used.*

We thank the reviewer for the suggestion and added all details as requested in the method section:

“ Sequence processing and OTU classification

The raw sequences obtained were processed to remove potential human contamination. The human genome (https://www.ncbi.nlm.nih.gov/assembly/GCF_000001405.39/) was masked with

the progenome 2 database <https://academic.oup.com/nar/advance-article/doi/10.1093/nar/gkz1002/5606617/>. Raw reads were mapped to the masked human genome and discarded upon 95% identity. Finally, we validated the human reads found by filtering “human” contamination and aligned these against the NCBI-database, resulting in only human top hits. The new human contamination removed raw reads were processed using LotuS (1.62) (16). Poisson binomial model based read filtering was applied (17). OTU clustering (UPARSE) (18) was based on a sequence similarity of 97%, while SILVA version 138 (19) was used for taxonomic profiling. The taxonomic classification (genus 95% identity) was parsed using a custom Perl script, such that unassigned taxonomic levels were assigned to the last known taxonomic level and sequentially numbered. Normalization and computation of alpha diversity measures were performed using the rarefaction tool kit (RTK 0.93.1) with default settings(41).

Under data sharing section, data can be accessed using the following data-specific identifiers: Accession number: PRJNA726058, Submission ID: SUB9549838 and data link: <https://www.ncbi.nlm.nih.gov/sra/PRJNA726058>.”

“Analysis plan:

Operational taxonomical unit (OTU) counts were rarefied to the smallest retained sample size (i.e 8278 raw reads) to obtain relative abundances of microbiota in each sample, accounting for read depths. Univariate analysis was done using metadeconfoundR (v 0.2.9)(45), relative abundances were tested for univariate associations with clinical variables, requiring Benjamini-Hochberg adjusted FDR < 0.1 and the absence of any clear confounders such as age, sex and body mass index. Only major taxa and OTUs detected after rarefaction in at least 10% of samples were used. Since the data was not normally distributed, non-parametric tests were used for all association tests. The Wilcoxon or the Kruskal-Wallis analysis of variance were used for discrete predictors. For pairs of continuous variables, a non-parametric Spearman correlation test was used. Benjamini-Hochberg False Discovery Rate control (FDR) was applied in all multiple testing situations requiring controlling the family-wise error rate at 10%. Hierarchical clustering was used to establish grouping patterns of the different study samples, including an updated adaptation of the approach used to define “enterotypes” in the human gut using the ‘Dirichlet Multinomial’ R package (v 1.36.0)(46). The chi-square test implemented in base R was used to test for significant differences in the resulting community type distribution between samples grouped by disease status. Beta diversity was calculated as Bray-Curtis dissimilarities as implemented in the vegan R (v 2.5-7) package (47). To determine the impact of participant clinical and sociodemographic characteristics on taxonomic composition of the sputum microbiome, permutational multivariate analysis of variance (PERMANOVA) was performed. Bray-Curtis distances were used for all analyses. PERMANOVA test was performed using the adonis test and pairwise multilevel comparison was conducted using the 'pairwiseAdonis' package in R (v 0.4)(48).”

Comment: *Please provide reference for metadeconfoundR, vegan and pairwiseAdonis packages.*

Response: Here are the references for the above packages

Oksanen J, Blanchet FG, Kindt R, Legendre P, Minchin P, O'hara R, et al. Community ecology package. R package version. 2013;2(0).

Martinez AP. pairwiseAdonis: pairwise multilevel comparison using adonis. R package version 0.3. 2020.

Forslund SK, Chakaroun R, Zimmermann-Kogadeeva M, Markó L, Aron-Wisnewsky J, Nielsen T, et al. Combinatorial, additive and dose-dependent drug-microbiome associations. Nature. 2021 Dec;600(7889):500-505. doi: 10.1038/s41586-021-04177-9. Epub 2021 Dec 8. PMID: 34880489.

Comment b. *No negative controls data is presented, and it only becomes apparent that these were performed in the discussion, please amend.*

Response: We thank the reviewers for this observation. We have included details in the methods section of the revised manuscript under “Quality control” as follows:

“We included negative controls (sputum kit with sterile water and buffer) during sample collection, DNA extraction, PCR amplification and sequencing. Negative controls were negative for V3-V4 amplicons at PCR and no sequences were generated after batch processing and sequencing with all other samples.”

Comment c. *Please clarify what filtering of the OTUs was carried out for potential contaminants. Did you remove reads identified as e.g. Eukaryota, Human and Cyanobacteria? Fig 2b shows an unknown chloroplast in the results which suggests no filtering of results was performed.*

Response: We thank the reviewer for bringing this to our attention.

The raw sequences obtained were processed to remove potential human contamination.

The potential human genome

(https://www.ncbi.nlm.nih.gov/assembly/GCF_000001405.39/) was masked using the

ProGenomes2 microbial genome database

<https://academic.oup.com/nar/advance-article/doi/10.1093/nar/gkz1002/5606617/>

Raw reads were mapped to the human genome and discarded upon 95% identity, masked and then filtered. Finally, we validated the human reads found by filtering out potential “human” contamination and aligned these against the NCBI nt database, resulting in only human top hits. After removal of human contamination, remaining raw reads were processed using LotuS (1.62)(16). Poisson binomial model based read filtering was applied(17). OTU clustering (UPARSE)(18) was based on a sequence similarity of 97%, while SILVA version 138 (19) was incrementally used as databases for taxonomic assignment using lambda taxonomic similarity search. The taxonomic classification (genus thresholded at 95% identity) was parsed using a custom Perl script, such that unassigned taxonomic levels were assigned to the last known taxonomic level and

sequentially numbered. Normalization and computation of alpha diversity measures were performed using the rarefaction tool kit (RTK 0.93.1) with default settings(41).

Comment d. How many raw reads were initially obtained in total and for each sample? I am concerned a large proportion of your data could have been omitted due to inappropriate rarefaction/ quality filtering processes but this is impossible to determine currently

We thank the Reviewer for the helpful suggestions and prepared supplementary Table 1 showing the raw read count before and after filtering for potential human contamination for each sample.

Comment e. Is there any justification to investigating as many alpha diversity indices as you have?

Response: We thank the reviewer for this comment. We wanted to get a better insight into the community structure by considering different levels of diversity. Specifically, Species evenness informed us how equally abundant species were in our sputum samples. The Simpson diversity index was used to calculate a measure of diversity taking into account the number of taxa as well as their abundance. The Shannon index summarized the diversity in the population while assuming all species were represented in a sample and were randomly sampled. CHAO1 index was appropriate for abundance data, assuming that the number of organisms identified for a taxa had a poisson distribution and therefore corrected for variance. To account for this multiple testing, we FDR-adjusted all reported p-values.

Comment: f. Did any samples fail quality controls? "retained sample" implies some sample data was excluded.

Response: No sample failed quality control. We included all 200 samples in the analysis. We excluded one sample from analysis due to missing metadata.

Comment g. It is now generally not recommended to rarefy microbiome data (see McMurdie & Holmes 2014 for details <https://journals.plos.org/ploscompbiol/article?id=10.1371/journal.pcbi.1003531>). Please consider what effect this has had on your results.

Response: We thank you for bringing this to our attention. We implemented additional functionality in the 'metadeconfoundR' R package for covariate-aware biomarker inference to include the raw read count in all models used for likelihood ratio testing. While we still relied on rarefied data to identify unadjusted associations between microbial features and patient group as well as potential confounders (as simple statistical association tests have no way to account for a third parameter, and as otherwise lower total read count samples would have been more likely to have taxa falling below detection threshold solely by virtue of that; similarly, for calculating standardized effect sizes such as the Cliff's delta and Spearman rho metrics used here, any other approach would have propagated any noise

or bias from sample total read count variation), we included the per-patient total read count into the linear models as follows: `glm (cbind (raw_reads_of_species_X, total_reads) ~ Group_label, family = "binom")`, an approach similar to that taken in DESeq. Comparing results from this approach to those obtained using our previous approach showed minimal differences in the resulting inferred significant and non-confounded associations. Accordingly, we conclude that accounting for total read count differences through rarefaction to size of the smallest sample, in this dataset specifically, resulted in minimal loss of sensitivity.

Impact of Disease Status, Medication, and other collected Metadata variables on the taxonomic profile of the sputum microbiota. Heatmap shows all genus-level taxa significantly [MWU (for categorical factors) and Spearman (for continuous features) *FDR<0.1, **FDR<0.01, ***FDR<0.001] different in abundance (binned rarefied 16S gene counts) depending on Disease Status (HIV/COPD) alongside participant characteristics. Heatmap cells show effect size (Cliff's Delta for categorical factors, Spearman's Rho for continuous features). Multi-confounder testing (nested generalized linear model testing, post hoc test), taking into account read depth in all models, was applied showing no stars or circles if either not significant (NS) in naive test or failing the pure pseudoreplication test. And, in the remaining naive-significant associations,

only those passing the deconfounding step as strictly deconfounded (SD), laxly deconfounded (LD) or no other covariates (NC) are black star, while any confounded signal is grey circle.

Using this method we indeed get a signal for COPD, however we decided to use the more conservative approach for this manuscript.

Comment: You have also not stated the number of reads you rarefied to ("smallest retained sample").

Response: The smallest retained sample had 8278 raw reads. We have included this in the results section.

“Operational taxonomical unit (OTU) counts were rarefied to the smallest retained sample size (i.e 8278 raw reads) to obtain relative abundances of microbiota in each sample, accounting for read depths.”

Comment: Did you use rarefied data for alpha diversity calculations, again this is no longer recommended (<https://www.frontiersin.org/articles/10.3389/fmicb.2019.02407/full>).

Response: Alpha diversity metrics were computed in the course of the process of rarefaction using the RTK tool <https://pubmed.ncbi.nlm.nih.gov/28398468/>) and are based on the raw data.

Comment: I am very concerned these data analysis steps will have impacted your results significantly.

Response: We thank the reviewer for raising these concerns. As stated above, we 1) do not use rarefied data to calculate the alpha diversity, but raw data and 2) for differential abundance tests, we verified results fundamentally agree between the analysis previously outlined (using rarefied data to account for differences in absolute read count) and the one now additionally implemented, accounting instead for such differences by its formal inclusion as a covariate in the linear models. Accordingly, we are confident that the results we report in this manuscript are not impacted by these particular concerns. We have outlined these additional tests in the revised manuscript.

6. Please provide further detail about how functional profiles of the microbiome were determined in the main manuscripts introduction and methods.

Response: Further details on these methods have now been included in the revised manuscript as follows.

“To project functional profiles from the composition of the airway microbiota assessed using 16S rRNA sequence data, PICRUSt2 (phylogenetic investigation of communities by reconstruction of unobserved states) (version 2.2.3) was used. PICRUSt2, does this using marker gene data and a database of reference genomes, as well as the airway microbiota quantified using 16S rRNA sequences. PICRUSt was applied to all samples from Uganda and UK as well as the merged UK-

Ugandan dataset. In our results, we present results of PICRUSt2 analysis from the Ugandan dataset.”

Comment: *The method detail in the supplement is also very limited (1 sentence). Were all samples included in this analysis or just the 100 from HIV-positive patients? The discussion (which could also be expanded) seems to imply the data was from COPD patients. Are the UK-PLWH data also used for functional profiling?*

Response: We have performed the PICRUSt2 analysis on all data from Uganda (HIV and COPD) as well as on the merged UK-PLWH data. However, the data we are presenting in the manuscript just include the 200 Ugandan samples.

Comment: *7. Presentation of results:*

a. Table 1 contains a lot of clinically relevant information, please consider revising for clarity and also performing statistical testing to identify where differences occur between groups.

Response: We thank the reviewers for pointing this out. We have revised the table and included p-values from a Chi-square test performed to determine whether statistically significant differences existed in sociodemographic and clinical characteristics between COPD+/- participants stratified by HIV status.

Comment: *It may not be necessary to report all the spirometry parameters currently included (suggest limiting it to FEV1 (L and %predicted), FVC and FEV1/FVC with the pre-BD spirometry reported in supplement if at all).*

Response: We agree with the reviewer. We have altered the revised manuscript accordingly.

Comment: Please revise the GOLD classification in line with current guideline (A,B,C,D).

Response: We thank the reviewer for this observation. We now use the term “airflow limitation classification” since we did not collect data on CAT and mMRC scores required for GOLD staging using A,B,C and D nomenclature. In our follow-up study, we are documenting CAT and mMRC scores.

Comment: *For some variables (eg Age, BMI, spirometry), it is best practice to report median and IQR not mean and 95%CI.*

Response: Agreed, this has been done.

Comment b. Please test and report statistical significance of the clinical characteristics in table 2

Response: Agreed, this has now been done and has been added to Table 2.

Comment c. Please ensure labeling and order of parts of figures is correct compared to legends and is sequential to how the data is presented in the manuscript (eg Fig3C before Fig3B).

Response: Agreed, this has now been done.

Comment: Also, please ensure legends only give technical description of the figure instead of presenting results data- this is particularly noticeable in Fig 3.

Agreed, we have streamlined figure legends to reflect the recommendation.

Comment: d. Please be careful to avoid sweeping statements (e.g. "were significantly associated with") when presenting your data without stating the direction and magnitude of the associations. Add in those cases such details (e.g. [MWU FDR < 0.1, Cliff's delta = -0.32]).

Response: Agreed, we now have added appropriate statistical detail in this regard wherever applicable.

Comment e. The paragraph titled "Impact of participant characteristics on airway microbiome diversity" comprises of very vague statements leaving the reader to draw their own conclusions. Please be more specific than "several bacterial genera" in line 188, and presumably you mean alpha diversity (Shannon) not microbial changes in relation to Fig S3. Why was only Shannon shown in Fig S3? This whole paragraph needs removing or significantly revising.

Response: We thank the reviewer for this observation, we have expanded and clarified this in the results sections accordingly. We have replaced all vague statements with more specific such. We performed correlation analysis between all alpha diversity scores (chao1, Simpson and Shannon) and lung function parameters. We found no significant correlation. We illustrate one of the alpha diversity indices (Shannon) here.

Comment f. You have a small section of results titled "Distinct airway bacterial genera are associated with COPD among PLWHA" which I feel could be expanded to provide further results of interest to readers by considering other factors which could impact the microbiome: Did you consider stratifying the microbiome based on virological suppression status or COPD severity, likewise use of cotrimoxazole (Septrin) or other prophylactic antibiotics?

Response: We thank the reviewer for this suggestion, and we agree it is essential to account for these factors. Rather than stratification, we accomplish this using the 'metadeconfoundR' R package, which In our analysis, we used MetadeconfoundR which does not use stratification, but tests for potential confounding factors among all supplied

metadata for each tested feature (e.g microbial genus) individually, using a mixed-effects linear model framework, as also described elsewhere (Forslund et al., Nature 2021). All requested variables were part of the current analysis as potential confounders, but many are not shown in the final figures because no association between them and the microbiome features evaluated reached statistical significance. We prepared an alternative plot forcing the inclusion of also such variables in the rendering, which is included here for clarity.

Impact of Disease Status, Medication, and other collected Metadata variables on the taxonomic profile of the sputum microbiota. Heatmap shows all genus-level taxa significantly [MWU (for categorical factors) and Spearman (for continuous features) *FDR<0.1, **FDR<0.01, ***FDR<0.001] different in abundance (binned rarefied 16S gene counts) depending on Disease Status (HIV/COPD) alongside participant characteristics. Heatmap cells show effect size (Cliff's Delta for categorical factors, Spearman's Rho for continuous features). Multi-confounder testing (nested linear model testing, post hoc test) was applied showing no stars or circles if either not significant (NS) in naive test or failing the pure pseudoreplication test. And, in the remaining naive-significant associations, only those passing the deconfounding step as strictly deconfounded (SD), laxly deconfounded (LD) or no other covariates (NC) are black star, while any confounded signal is grey circle. Using the function: keepMeta = c("co-trimoxazole", "Dapsone") enabled us to show results for the requested variables.

Comment: Were there different amounts of AMR genes (presented later in the manuscript) in the cohort based on stratification for cotrimoxazole use?

Response: We thank the reviewer for this suggestion. As per the immediately preceding concern and response, we used a statistical framework that evaluates whether any association is redundant and reducible to any other association, including co-trimoxazole use. Including the latter as a predictor in models does not significantly increase explanatory power for any microbiome measurement made here, whereas the variables shown does increase the predictive power containing only the co-trimoxazole status as predictor.

Impact of Disease Status, Medication, and other collected Metadata variables on functional profile of the sputum microbiota. Heatmap shows KEGG modules significantly [MWU (for categorical factors) and Spearman (for continuous features) *FDR<0.1, **FDR<0.01, ***FDR<0.001] different in abundance (KEGG

modules) depending on Disease Status (HIV/COPD) alongside participant characteristics. Heatmap cells show effect size (Cliff's Delta for categorical factors, Spearman's Rho for continuous features). Multi-confounder testing (nested linear model testing, post hoc test) was applied showing no stars or circles if either not significant (NS) in naive test or failing the pure pseudoreplication test. And, in the remaining naive-significant associations, only those passing the deconfounding step as strictly deconfounded (SD), laxly deconfounded (LD) or no other covariates (NC) are black star, while any confounded signal is grey circle. Using the function: keepMeta = c("co-trimoxazole", "Dapsone") enabled us to show results for the requested variables.

Comment g. *Are the axis labels correct in Fig S6? There are no Staphylococcus, Pseudomonas, Lactobacillus, Klebsiella and Bilophila showing compared to the manuscript text*

Response: We agree with the reviewer's observation. Bacterial genera Bilophila, Staphylococcus, Pseudomonas and Klebsiella were reported from the article by Rofael et al, 2020, referenced as article 13. In our results section, we compare potential differential abundance of these genera to the findings in the Ugandan cohort where instead, genera such as Atopobium, Stomatobaculum, Oribacterium, Butyrivibrio, Peptostreptococcus, Clostridium and Lentimicrobium were enriched in the cohort. Furthermore, from re-analysis, individuals with COPD-HIV comorbidity were enriched for gut Campylobacter and depleted of genera Staphylococcus and Negativicutes (Fig.5).

Comment. *There is no evidence in your socio-demographics of recreational drug use in the Ugandan cohort yet you present this in the discussion as a potential reason for the differences in the microbiome between UK and Ugandan PLWH cohorts.*

Response: There was no single participant in our cohort who reported used any recreational drug such as cocaine or marijuana. We describe this in the results section and table 3.

Minor Comments:

Comment 1. *Recommend not abbreviating COPD in the title and add 'living' between 'adults' and 'with'.*

Response: We have edited the revised title to read as follows: "Sputum microbiome and Chronic Obstructive Pulmonary Disease in a rural Ugandan cohort of well-controlled HIV infection."

Comment 2. *In abstract, please consider rewording the sentence beginning "Airway microbiome is essential..." as the literature does not support the statement in its current form. We understand that a healthy (non-dysbiotic) airway microbiome is an important marker for better long-term outcomes but there is not yet sufficient evidence about how "essential" the airway microbiome is for maintaining a healthy immune response.*

Response: We agree with the reviewer, we have edited this statement to in the revised manuscript instead read as follows

“A healthy airway microbiome is an important marker for better long-term outcomes and could be essential in maintaining a healthy airway immune response.”

***Comment:** 3. Correct the abbreviation of rDNA to rRNA for ribosomal RNA in abstract (and throughout manuscript).*

Response: Agreed, this has been rectified throughout the manuscript.

***Comment:** 4. Is the emergence of NCDs and increase in COPD mortality only in PLWH or in the general sub-Saharan population? Or are the emergence on NCDs and increase in COPD mortality greater in the PLWH population compared to the non-PLWH population?*

Responses: We thank the reviewers for this comment. We agree that the burden of NCDs and increase in COPD-mortality has increased both in the general population and among PLWH. In the current era of well-controlled HIV, PLWH now live longer, with a life expectancy close to that of the general population. We and others have reported a high prevalence and excess mortality of COPD among PLWH compared to the non-PLWH population.

References

Kayongo A, Wosu AC, Naz T, Nassali F, Kalyesubula R, Kirenga B, Wise RA, Siddharthan T, Checkley W. Chronic obstructive pulmonary disease prevalence and associated factors in a setting of well-controlled HIV, a cross-sectional study. *COPD: Journal of Chronic Obstructive Pulmonary Disease*. 2020 May 3;17(3):297-305.

North CM, Kakuhikire B, Vořechovská D, Hausammann-Kigozi S, McDonough AQ, Downey J, et al. Prevalence and correlates of chronic obstructive pulmonary disease and chronic respiratory symptoms in rural southwestern Uganda: a cross-sectional, population-based study. *Journal of global health*. 2019;9(1).

North CM, Allen JG, Okello S, Sentongo R, Kakuhikire B, Ryan ET, et al. HIV infection, pulmonary tuberculosis, and COPD in rural Uganda: a cross-sectional study. *Lung*. 2018;196(1):49-57.

Byanova K, Kunisaki KM, Vasquez J, Huang L. Chronic obstructive pulmonary disease in HIV. *Expert Review of Respiratory Medicine*. 2021 Jan 2;15(1):71-87.

Bigna JJ, Kenne AM, Asangbeh SL, Sibetcheu AT. Prevalence of chronic obstructive pulmonary disease in the global population with HIV: a systematic review and meta-analysis. *The Lancet Global Health*. 2018 Feb 1;6(2):e193-202.

We have edited the first paragraph to improve clarity in the revised manuscript. It now reads as follows

“Improved access to antiretroviral therapy (ART) among people living with HIV (PLWH) has resulted in a decrease in HIV-associated morbidity and mortality over the past two decades(1, 2).

This is particularly true in low- and middle-income countries (LMICs), which bear the largest burden of HIV/AIDS(3). The reduction in mortality has substantially increased life expectancy, which now approaches that of the general population(2). Consequently, there has been increased attention paid to the emerging burden of non-communicable diseases (NCDs) among survivors(4). For example, Sub-Saharan Africa, which has the highest density of PLWH, has experienced dramatic increases in COPD prevalence (5-7).”

Comment: Please clarify in these sentences in the introduction: "This has resulted, however, in an emergence of non-communicable diseases (NCDs).

Response: In the revised manuscript, this statement has been removed from the first paragraph.

For example, Sub-Saharan Africa, which has the highest density of PLWH, has experienced dramatic increases in mortality related to chronic obstructive pulmonary disease (COPD), a chronic respiratory illness characterized by progressive and irreversible decline in lung function."

Responses: This statement has been edited to reflect COPD prevalence more clearly

Comment 5. Suggest changing "unlikely represents" in the sentence beginning "Although sputum microbiome..." to "may not fully represent".

Response: Agreed, this has now been done.

Comment 6. In this sentence: "The number of raw reads retrieved from the HIV COPD- group was significantly higher compared to all HIV+, all COPD+ and both COPD/HIV+ groups (S1)", please remove the words "all" and "both" as these imply HIV+ would include an overlap of patients with and without COPD, and vice versa.

Response: The statement reads now as "The number of raw reads retrieved from the HIV-COPD- group was significantly higher than to those from the HIV+, COPD+ and COPD+HIV+ groups' ".

Comment 1. Were COPD patient samples collected when clinically stable (i.e. no exacerbation)?

Response: Correct, all COPD participants were clinically stable with no exacerbation.

Comment: You have excluded antibiotic use 2 weeks prior to sampling but are all exacerbations in your COPD population treated or are some exacerbations untreated?

Response: In this study, all COPD participants were stable with no exacerbations. Most patients with COPD exacerbations who seek treatment at the clinic are treated with

antibiotics. In this study however, we did not recruit patients with COPD exacerbation, our target was stable COPD.

Comment: 2. Please check all abbreviations have been defined on first use (eg MWU, IHK-IRR)

Response: This has been rectified throughout the manuscript.

Comment 3. In the online supplement, the data should be split into methods and results- the UK PLWH comparison is combined. Please move the supplementary figures to the supplement.

Response: Agreed, this has now been done in the revised manuscript.

Comments 4. Why were three C/D spiro samples excluded in Fig 1?

Response: In this study, Grade A and B were considered the highest quality spirometry curves obtained following standard ATS/ERS guidelines. Average FEV1 and FVC is minimally influenced by grades A and B. On the contrary, Grade C/D were considered low quality curves. FEV1 and FVC values are significantly influenced by grades C, D and lower, hence cannot be relied upon to diagnose airflow limitation in COPD. We have included this information in the revised methods section.

Reference

Hankinson JL, Eschenbacher B, Townsend M, Stocks J, Quanjer PH. Use of forced vital capacity and forced expiratory volume in 1 second quality criteria for determining a valid test. European Respiratory Journal. 2015 May 1;45(5):1283-92.

Comment 5. Why isn't the data in table 3 kept with the data in table 1?

Response: To declutter table 1 and reduce its length, we created a separate table 3 for respiratory symptoms and past medical history.

Comment 6. Please avoid using the brand name for a drug (Septrin)

Response: Agreed, this has been changed to read "co-trimoxazole" to reflect this.

Comment 7. It is best practice to report P values themselves instead of using *, **, ***. Please also consider adding P values into the manuscript text rather than just stating "was significantly....".

Response: Agreed, we have included FDR-adjusted p-values throughout the revised manuscript.

Comment 8. Please review your group naming throughout the manuscript- you are using HIV-COPD- and control interchangeably in text/ figures, likewise for community type 1,2 and 3 vs Streptococcus, Prevotella and Mixed in Fig 3.

Response: Agreed. This has been corrected in the revised manuscript.

Comment 9. Fig S1 shows the comparison between HIV- COPD- and HIV+ as NS but the manuscript states this difference in read number was significant.

Response: Agreed. This has been corrected in the revised manuscript. The number of raw reads retrieved from the HIV-COPD- group was significantly higher than the number assessed in COPD+ and COPD+HIV+ groups.

Comment 10. Why is abundance presented in Fig2a and relative abundance in Fig2b? Would a stacked bar graph in Fig 2b be clearer?

Response: We thank the reviewer for this suggestion. We have now used stacked bar graphs to visualize the relative abundances of phyla and genera per study group in the revised manuscript.

Comment Do these bars represent the average abundance/ relative abundance of the four patient cohorts or the total combined reads in each group?

Responses: They represent relative abundances of the microbiome phyla and genera in all induced sputum samples stratified by COPD and HIV.

Comment 11. In Fig3a legend, please specify what the ellipses represent.

Response: The ellipses represent 95% confidence intervals. We specify this in the revised legend.

Comments 12. I am interested to know how many (instead of %) patients from each of the four patient cohorts are represented in the three community types presented in Fig 3, rather than just HIV-/+ and COPD -/+.

Response: Percentage and actual number of participants with each community type in tested patient cohorts i.e %(n)

Community type	HIV-/COPD-	HIV-/COPD+	HIV+/COPD-	HIV+/COPD+
community type 1	25.68% (18)	21.62% (16)	31.08% (22)	21.62% (16)
community type 2	29.85% (20)	22.39% (15)	23.88% (16)	23.88% (16)
community type 3	13.79 % (12)	34.48% (19)	22.41% (12)	29.31% (18)

Comment 13. *I am unsure how Fig 5 shows well controlled HIV (instead of HIV+) is associated with the genera Atopobium, Actinomyces and Megasphaera (line176).*

Response: This has been clarified in the revised figure legend.

Comment 14. *In the volcano plot legends (e.g. Fig 5, S4) please confirm whether "subject" should be plural throughout (e.g. "the subject suffering from both COPD and HIV...".Add "could" between "counts" and "explain" on line 237.*

Response: We thank the reviewer for noting this, we have now gone over the revised manuscript and addressed this concern.

Reviewer #2 (Comments for the Author):

The manuscript by Kayongo et al., titled "Sputum microbiome and COPD status in a rural cohort of Ugandan adults with HIV", describe three community types whose distribution differed by HIV status but not by COPD status. Except for richness and Chao1, alpha-diversity metrics did not differ between the HIV+/COPD+ group and the control groups (HIV-/COPD-, HIV-/COPD+, HIV+/COPD-). Although factors such as sex, BMI, history of pulmonary tuberculosis, HIV status, and years of ART were associated with differences in overall microbial community composition, the sputum composition did not significantly differ in the compared groups. However, using an interesting yet-to-be-published statistical approach, named metadeconfoundR, the authors identified a decreased abundance of specific genera associated with HIV+COPD+ after accounting for confounding effects of other variables. In general, this work is a valuable contribution to better understanding the COPD-associated airway microbiome in HIV patients from a setting with a high risk for HIV, such as Uganda. However, three aspects need clarification:

Comment (1) *The study design. The authors explained a case-control design where a case group (HIV+COPD+) is matched to three control groups (HIV-/COPD-, HIV-/COPD+, HIV+/COPD-). The matching was performed based on the frequency distribution of values for three variables in the case group: age (4 categories), sex (2 categories), and smoking status (4*

categories). Thus. It would give a total of 32 categories that need matching. It is unlikely to achieve this from a pool of "226 potential participants". I presume the pool of participants recruited between February 2018 and February 2020 was much larger, considering that they were part of population-based cohorts. In this sense, the description of the "Study design" is incomplete and needs to be slightly extended both in the Methods (or Supplementary material) and in Figure 1 (Flow diagram for participant screening and enrollment).

Response: We thank the reviewer for this observation and suggestion. We have clarified the study design further in the revised Figure 1 and in the methods section.

Comment (2) *The sputum induction and sample collection. The procedures for sputum induction are well described. The strength of the procedure is the three-step cleansing routine to minimize contamination with microbes from the upper respiratory tract. However, there are some aspects of the procedure that need clarification. Based on the description in the methods (Lines 308 to 321), I understand that after each 5-minute nebulizing period, an expectorate was collected. Only if the percentage fall in FEV1 was less than 20% (relative to the baseline before the induction procedure started), in addition to passing a quality control check (basically evaluating consistency). This procedure introduces a technical variability, the time interval at which the sputum was collected. This variability is important because it has been shown that the composition of induced sputum varies depending on the time point during the duration of the procedure (Gershman et al., 1999). Do the authors have records of the time interval at which each sample was collected?*

Responses: We thank the reviewer for this comment. All sputum samples were collected within 15 minutes of sputum induction. This was standardized across all participants to minimize technical variability. Three participants who failed sputum induction were excluded from the analytical sample size.

When comparing the case and control groups, are there differences in the time interval?

There was no difference in times of sample collection between cases and controls. All sputum samples were collected within 15 minutes of sputum induction. This was standardized across all participants to minimize technical variability.

Also, are there any records of whether or not a sample was collected after a repeated induction procedure? This information will help identify technical biases that could contribute to the differences between groups.

Responses: All samples were collected after the first attempt. We had high success rates for the sputum induction procedure. Three participants who failed sputum induction were excluded from the analytical sample size.

Comment (3) *The statistical approach. The authors used an interesting/novel statistical procedure to find associations between characteristics of the sputum microbial community, relative abundances of individual taxa, and exposure variables; this approach emphasizes identifying associations after accounting for potential confounders. The approach is a bit cumbersome. The authors described it in a very technical way with no explanation of why this approach was preferred over others that take into account the compositional nature of the relative abundances (e.g ALDEx2).*

Response: We thank the reviewer for this comment and recognize the relevance of noting the compositional nature of microbiome data. The tool ('metadeconfoundR' R package) has been more formally published (Forslund et al., Nature 2021) since initial submission of the manuscript as well as used in several other studies by us and others (e.g. Thirion et al., Biol Psychiatry Glob Open Sci. 2022). There are several reasons we prefer this tool, the first being that it offers an automated and scalable capacity to address an arbitrary number of potential confounding factors, by systematic nested mixed effects modeling applied as a post-hoc filter. Moreover, the tool is not limited to relative abundances but can make use of the fact that reads are count data, which offer substantially higher statistical power as the error function can reflect this. While e.g. ALDEx2 could be used for this purpose also, implementing the full scope of intended confounder testing would have been more time consuming, and we have substantial benchmarking of metadeconfoundR performance on simulated data (Wirbel et al., Nat Met, submitted). In the revised manuscript, we now elaborate more on the tool and its application to make these rationales clearer.

Comment: *To understand the approach and interpret the figures in the manuscript, I had to look at the two publications where the methodology was previously used (Bartolomaeus 2021, Forslund 2021). This is not ideal. The manuscript should be self-contained and provide the readers with an intuitive explanation of how the outcomes of the statistical methods must be interpreted. The manuscript would benefit from improving the description of the biostatistical methods and the legends of figures where the methodology is used (e.g, Figure 3C, Figure 5, S2, S4).*

Response: We thank the reviewer for this suggestion and agree wholeheartedly. We have included a Supplementary Figure S7 of the revised manuscript to describe the method better, as well as clarified throughout method and especially figure legends.

“**Supplementary Figure 7:** Overview schematic of metadeconfoundR statistical methods: (left) naive association testing of individual features and covariates using rank-based tests. (middle) nested model post-hoc linear model likelihood ratio tests to determine relative redundancy between covariates associated to the same feature. (right) label assignment based for each feature, covariate combination based on initial naive tests as well as linear model testing results.”

Major:

Comment: Line 128 "... recruited from 226 potential participants". Based on the description of the study design (Line 281) and the legend for Figure 1 (Line 439), it is my impression that a larger pool of individuals was needed to be able to create the groups; matched by the frequency of age-, sex-, and smoking status values in the HIV+/COPD+ group. Please clarify, and update Figure 1 if needed to reflect the real pool of participants.

Response: We have clarified this in Figure 1. In the revised manuscript.

Comment: Line 129 "Fifty-nine percent of participants were male (59%), 43% were aged >55 years and 63% were non-smokers." Given that the groups shown in Table 1 were matched based on the frequency of the values of these variables in the HIV+/COPD+ group, it should be expected to observe similar frequencies in the other three groups. Can the authors explain why this is not the case?

Response: We thank the reviewer for this observation. We have corrected the terminology in the revised manuscript. We randomly selected participants from the two HIV- and HIV+ cohorts described in Figure 1.

Comment: Line 144, "... after demultiplexing and quality control filtering". The quality control filtering is described neither in the Methods nor in the Supplementary Material. I would briefly describe or add it to the Supplementary material.

Response: Agreed, we now provide this information in the methods section of the revised manuscript.

“The raw sequences obtained were processed to remove potential human contamination (supplementary table 1). The human genome (https://www.ncbi.nlm.nih.gov/assembly/GCF_000001405.39/) was masked with ProGenomes2 microbial genome database <https://academic.oup.com/nar/advance-article/doi/10.1093/nar/gkz1002/5606617/>. Raw reads were mapped to the human genome and discarded upon 95% identity, masked and then filtered. Finally, we validated the human reads found by filtering out potential “human” contamination and aligned these against the NCBI nt database, resulting in only human top hits. After removal of human contamination, remaining raw reads were processed using LotuS (1.62)(16). Poisson binomial model based read filtering was applied(17).”

***Comment:** Line 144, "OTU counts were rarefied to the size of the smallest retained sample." Please specify this size. This approach has the disadvantage of including low-quality samples that generally have fewer raw reads. An alternative approach could be to set the rarefaction threshold above the number of reads obtained in negative controls. Alternatively, set a threshold where a good coverage of the microbial community richness can be obtained. I would recommend plotting the rarefaction curves at different sampling sizes for each sample to assess how well the sputum communities were covered.*

Response: We did not exclude any samples before rarefaction. The number of reads obtained after rarefaction to the depth of the smallest sample was sufficient for further analysis and similar in scope to that used in typical 16S studies.

***Comment:** Line 147, "We accounted for these differences during further analysis using a rarefaction toolkit for normalization." Rarefaction to an even sampling size does not remove the effect of differences in raw read counts. Raw read counts should be included in all the models to account for heterogeneous sequencing depth, especially if the HIV-/COPD- group has higher raw read counts.*

Response: We thank the reviewer for this suggestion. We implemented additional functionality in metadeconfoundR (updating the package accordingly to include this as a user option) to include the raw read count as an additional covariate into all GLMs used in the analysis. Comparing this approach to our previous approach showed minimal differences in the resulting inferences, though we agree this may have greater impact in another dataset.

Impact of disease status, medication, and other collected metadata variables on taxonomic composition of the sputum microbiome. Heatmap shows all genus-level taxa significantly [MWU (for categorical factors) and Spearman (for continuous features) *FDR<0.1, **FDR<0.01, ***FDR<0.001] different in abundance (binned rarefied 16S gene counts) depending on disease status (HIV/COPD) alongside participant characteristics. Heatmap cells show effect size (Cliff's Delta for binary factors, Spearman's Rho for continuous features). Parallel post-hoc testing for all possible confounders was applied (using nested linear model comparisons and including total read count to account for heterogeneous sequencing depth), for each cell showing no stars or circles if the association was not significant (NS) in the initial naïve test step. In the remaining naïvely significant associations, only those additionally passing the deconfounding post-hoc testing step as being strictly deconfounded (SD) or laxly deconfounded (LD), or having no other significant covariates (NC) are shown as black stars, while any confounded signal is shown as a grey circle.

Using this method, we indeed get a signal for COPD, however we decided to use the more conservative approach for this manuscript.

Comment: Line 223, "COPD was also associated with a higher abundance of *Staphylococcus* and lower abundance of organisms belonging to the genera *Pseudopropionibacterium*, *Porphyromonas*, and *Parvimonas*." Here the paragraph does not discuss the lower abundance of *Pseudopropionibacterium*, *Porphyromonas*, and *Parvimonas* associated with COPD in the study's cohort, which is the opposite of what has been found in the previous studies mentioned in the same paragraph.

Response: We agree with the reviewer, we have elaborated on the findings regarding these genera in the discussion section of the revised manuscript.

Comment: Line 253, "A stringent quality control check at the time of sample collection was followed to reduce saliva and postnasal drip contamination." The methods section only mentions checking for consistency (mucoïd), were there other qualitative/quantitative features assessed (e.g. volume, color, presence of blood)?

Response: Upon deep coughing and expectoration, each sputum sample was assessed for mucoïd consistency, volume and color. Gram's stain procedure was performed for quality assessment. Sputum samples with less than 10 squamous epithelial cells per low-power field (x10) microscopy (indicative of a lower airway sample) passed quality control check(41).

Comment: Line 254, "We also included negative controls (sputum kit with sterile water and buffer) during sample collection." This is a strength of the work presented. However, it is not mentioned how the negative controls were used to identify contaminated samples or the presence of contaminants in the community profiles.

Response: We agree with the reviewer, and have now included this information in the methods section of the revised manuscript.

Comment : Line 315, "... and the induction procedure repeated." Is there a record of the number of attempts the induction procedure was performed? When repeated, was it performed just after the previous attempt? Would this introduce certain biases in the sampling of the sputum microbial community?

Responses: We successfully induced sputum from 200 participants on the first attempt within 15 minutes of nebulisation with 3% hypertonic saline. Three participants who failed induction on the first and second attempt were excluded.

Comment: Line 374, "The code is available upon request". The code should be available as supplementary material.

Response: It is now online and can we link to it in the revised manuscript.

Comment: Line 57, the authors stated, "we show that among PLWH, airway enrichment with *Staphylococcus* spp as well as depletion of *Pseudopropionibacterium* and *Porphyromonas* spp are associated with COPD." However, based on the results presented, *Staphylococcus* was associated with COPD status (including HIV- participants) but not with COPD-HIV status. Thus the statement needs to be rephrased.

Response: We thank the reviewer for this comment. The results section has been updated and re-written following the re-analysis steps suggested by the reviewers.

Comment: Line 132 "... 86% virologically suppressed with a median viral load of <20 copies/ml". This seems to conflict with what is shown in Table 2. Only 14% and 8% of HIV+ participants with COPD+ and COPD-, respectively, had a viral load of <20 copies/ml. Please clarify.

Response: We thank the reviewers for this observation. We have corrected this in Table 2 in the revised manuscript.

Comment: Line 146, "The number of raw reads retrieved from the HIV-COPD- group was significantly higher compared to all HIV+, all COPD+ and both COPD/HIV+ groups (S1)." Notice that the raw read count of the HIV-COPD- is not significantly higher than in the HIV+.

Response: We thank the reviewer for this observation. This has been corrected.

Comment: Line 167, "Microbial richness was significantly lower among COPD+/HIV+ group compared with other groups (Figure 4A)." Notice that mean microbial richness in COPD+/HIV+ does not seem to be lower than in COPD-/HIV- in Figure 4A. Please add the actual mean values to the legend in Figure 4A.

Response: We thank the reviewer and prepared a table showing the mean \pm sd for each evenness and diversity index showed in the manuscript.

	Shannon	Simpson	InvSimpson	Richness	chao1	Evenness	Pielou
HIV-/COPD+	3.64 (± 0.6)	0.93 (± 0.06)	19 (± 9.8)	297.12 (± 65.33)	415.98 (± 82.13)	0.64 (± 0.09)	0.64 (± 0.09)
HIV+/COPD+	3.56 (± 0.55)	0.93 (± 0.06)	18.95 (± 10.63)	263.12 (± 62.02)	359.87 (± 72.71)	0.64 (± 0.08)	0.64 (± 0.08)
HIV+/COPD-	3.74 (± 0.33)	0.94 (± 0.02)	20.08 (± 7.18)	277.98 (± 59.67)	385.59 (± 81.77)	0.67 (± 0.04)	0.67 (± 0.04)
HIV-/COPD-	3.53 (± 0.44)	0.93 (± 0.03)	18.01 (± 9.21)	258.91 (± 58.85)	365.23 (± 77.53)	0.64 (± 0.06)	0.64 (± 0.06)

Line 213, "Nevertheless, 16S raw reads and microbial richness was reduced among PLWH with COPD despite HIV control". See previous comment.

Line 241, "Furthermore, in our study, antimicrobial resistance genes reflect a potential multidrug resistome reservoir among COPD individuals". Considering that Staphylococcus is enriched in COPD participants, could Staphylococcus be the primary driver of the antimicrobial resistance genes associated with COPD?

Response: We thank the reviewer for this interesting suggestion. We performed an additional metadeconfoundR run on the functional prediction data, this time including Staphylococcales (the nearest representative of Staphylococcus on genus level) abundance as an additional covariate. Interestingly, the only drug resistance gene remaining after our new, more stringent contamination filtering, is indeed positively correlated with Staphylococcales. However, this was not associated with COPD or HIV status. Thus, mostly likely, it does not drive antimicrobial resistance in COPD cohort.

Impact of Disease Status, Medication, and other collected Metadata variables on functional profile of the sputum microbiota. Heatmap shows KEGG modules significantly [MWU (for categorical factors) and Spearman (for continuous features) *FDR<0.1, **FDR<0.01, ***FDR<0.001] different in abundance (KEGG modules) depending on Disease Status (HIV/COPD) alongside participant characteristics. Heatmap cells show effect size (Cliff's Delta for categorical factors, Spearman's Rho for continuous features). Multi-confounder testing (nested linear model testing, post hoc test) was applied showing no stars or circles if either not significant (NS) in naive test or failing the pure pseudoreplication test. And, in the remaining naive-significant associations, only those passing the deconfounding step as strictly deconfounded (SD), laxly deconfounded (LD) or no other covariates (NC) are black star, while any confounded signal is grey circle.

Line 324, "Bacterial 16S rRNA V3 region was amplified and ..." Notice that the supplementary material mentions the V3-V4 hypervariable region. Please clarify.

Response: We amplified 16S rRNA V3-V4 region. This has been clarified in the revised manuscript.

Comment: Line 445, the legend of Figure 2B: "Abundances of microbiome at genus level stratified by COPD and HIV." are the displayed genera representing the top 20 most abundant ones? *Streptococcus* is not displayed, but it should be expected to be among the most abundant genera.

Response: Correct, this has been rectified clarified in the revised manuscript.

Comment: Line 450, Figure 3A: for consistency, please display relative abundances either as percentages, fractions, or rarefied counts in all figures.

Response: Agreed, we have displayed them as counts.

Comment: Line 476, Legend of Figure 4: The panel's figures should be shown in the order they appear in the text. It seems the letters of the sub-figures were swapped.

Response: This has been corrected in the revised manuscript

Comment: Line 481, Please display the amount of variance explained by PCo1 and PCo2.

Response: This has been done.

Comment; Line 567, correct title. it should be ST1

Response: ST1 has been changed to Table 4

Comment: Line 570, correct title. it should be ST2

Response: ST2 has been removed since all results were summarized by Figure S2 after metadecomfoundR analysis.

August 7, 2022

Dr. Sofia Kirke Forslund
Max Delbrück Center for Molecular Medicine
Berlin, Germany
Germany

Re: Spectrum02139-21R1 (Sputum microbiome and Chronic Obstructive Pulmonary Disease in a rural Ugandan cohort of well-controlled HIV infection)

Dear Dr. Sofia Kirke Forslund:

Thank you for submitting your revised manuscript to Microbiology Spectrum. The manuscript has been reviewed by one of the former reviewers and the reviewer indicates that the quality of the manuscript has improved. However, in the revised version, two new primary concerns are evident. First, please directly address the reviewer's concern regarding changes to the study design during the revision process. Second, please also address if there have been changes in methodology related to sequence data processing (See comment 2). Two of the other remaining comments, although less critical, are also important to address. Specifically, with respect to Comment #3, the reviewer is correct that it is typical for background technical controls to generate sequence data unless the number of amplification cycles is very low. If the technical controls did not produce sequences, then this can be stated as is currently done in the manuscript; however, please reaffirm this is so and potentially explain why. Lastly, it is also important to directly address Comment #10. It is well established that methodology greatly affects microbiome profiles. Therefore, please explain why this is not an issue for this particular analysis comparing data from cohorts in which samples were processed in different ways.

When submitting the revised version of your paper, please provide (1) point-by-point responses to the issues raised by the reviewer as file type "Response to Reviewers," not in your cover letter, and (2) a PDF file that indicates the changes from the original submission (by highlighting or underlining the changes) as file type "Marked Up Manuscript - For Review Only". Please use this link to submit your revised manuscript - we recommend that you submit your paper within the next 60 days or reach out to me. Detailed instructions on submitting your revised paper are below.

Link Not Available

Sincerely,

Kevin R. Theis

Journals Department
Reviewer comments:

Reviewer #2 (Comments for the Author):

[Please see the attached file for formatted review.]

Thank you to the authors for answering most of my comments and incorporating changes based on those comments. The changes made to clarify the study design, sample collection, and statistical approach have improved the quality and clarity of the manuscript considerably. However, some of the changes raise the following major concerns:

1. In the previous version of the manuscript, the authors described a case-control design where a case group (HIV+ COPD+) is matched to three control groups (HIV- COPD-, HIV- COPD+, HIV+ COPD-). The matching was performed based on the frequency distribution of values for three variables in the case group: age (4 categories), sex (2 categories), and smoking status (4 categories). I asked the authors to clarify and extend in the manuscript the description of how the frequency-based approach for matching was achieved. It is surprising to see that in the current version, the authors removed any mention of the frequency-based matching. What happened? Did the authors mistakenly describe the study design in the first version? Although the study design that is described in the current version would be fine to support the current results (which have also changed), the drastic change raises suspicion about the integrity of the research.

2. The authors explained in their response to the reviewers, and I quote, that "the results section has been updated and re-written following the re-analysis steps suggested by the reviewers". One of the requests, from both reviewers, was not to do rarefaction and include library sizes as co-variables in the models. The authors showed evidence that their current results do not change considerably when including total read counts in their models, therefore they decided to present their results using rarefied data. However, the current results considerably differ from the previous version. If the current results are still based on rarefied data, and using the same set of samples, what were the modifications made to the methodology? I read the comments of reviewer #1 and the author's answers. I can only pinpoint an additional quality filtering step that removes reads matching human DNA. Is there any other change to the data processing?

3. The authors claimed in the first version of the manuscript "We also included negative controls (sputum kit with sterile water and buffer) during sample collection, DNA extraction, PCR amplification and sequencing."

We, both reviewers, pointed out that the data from negative controls is not presented nor how it was used to identify contaminated samples or the presence of contaminants in the community profiles. In the current version, the authors state that "We included negative controls (sputum kit with sterile water and buffer) during sample collection, DNA extraction, PCR amplification and sequencing. Negative controls were negative for V3-V4 amplicons at PCR and no sequences were generated after batch processing and sequencing with all other samples."

Based on my experience, and also from reports of previous studies [see Segal et al. *Nature Microbiology* 2016 (DOI: 10.1038/NMICROBIOL.2016.31), Segal et al. *Microbiome* 2013 (<https://doi.org/10.1038/nmicrobiol.2016.31>)], negative controls or background controls do end up producing sequences.

Other comments:

4. Line 230-231 ("Within the significantly associated gene families, only genes for the bacterial malate transport pathway were enriched in HIV").

The authors didn't discuss this finding.

5. Line 263 ("we detected a significant reduction in bacterial richness."). This statement is vague, it gives the impression that HIV-COPD comorbidity had reduced bacterial richness compared to the opposite (HIV and COPD negative). Based on the data, this is not the case.

6. Line 265 ("three community types, whose distribution was significantly impacted by HIV status."). Again, based on the data, this is an overstatement. The data showed that only community type 3 had a statistically significant higher frequency among HIV+.

7. Lines 268-281. The second paragraph in the Discussion section is confusing. It focuses on describing how sputum microbial composition varies across HIV-infected individuals and the factors influencing that variation. The paragraph does not discuss, clearly, why the current study only shows "subtle compositional differences" between HIV+/- groups as opposed to substantial differences found in other studies.

8. Lines 289-294 ("In this study, we clearly demonstrate the effects of HIV status on the distribution of the microbial community types we defined."). This is an overstatement since the data does not support it. The authors describe how the frequencies of those community types were higher or lower despite those differences not being statistically significant for all community types.

9. Lines 315-316 ("We could show that community type 3, dominated by the *Prevotella* genera, is predominant in HIV-positive study participants.").

Another overstatement. Although community type 3 is more frequent in HIV+, compared to HIV-, the other two community types represent 63% of HIV+ participants.

10. Lines 326-339. It is not clear why the authors decided to compare the microbial profiles of their

Ugandan cohort with a cohort from the UK. Geographical differences are reported but this finding is completely confounded by the use of a different DNA extraction method in the UK cohort. Including this comparison in the manuscript only adds confusion.

11. Revise the labels of Y-axes in Figure 4A. There are either duplicates or incorrect labels.

Staff Comments:

Preparing Revision Guidelines

Please return the manuscript within 60 days; if you cannot complete the modification within this time period, please contact me. If you do not wish to modify the manuscript and prefer to submit it to another journal, please notify me of your decision immediately so that the manuscript may be formally withdrawn from consideration by Microbiology Spectrum.

Thank you to the authors for answering most of my comments and incorporating changes based on those comments. The changes made to clarify the study design, sample collection, and statistical approach have improved the quality and clarity of the manuscript considerably. However, some of the changes raise the following major concerns:

1. In the previous version of the manuscript, the authors described a case-control design where a case group (HIV+COPD+) is matched to three control groups (HIV-/COPD-, HIV-/COPD+, HIV+/COPD-). The matching was performed based on the frequency distribution of values for three variables in the case group: age (4 categories), sex (2 categories), and smoking status (4categories). I asked the authors to clarify and extend in the manuscript the description of how the frequency-based approach for matching was achieved. It is surprising to see that in the current version, the authors removed any mention of the frequency-based matching. What happened? Did the authors mistakenly describe the study design in the first version? Although the study design that is described in the current version would be fine to support the current results (which have also changed), the drastic change raises suspicion about the integrity of the research.

2. The authors explained in their response to the reviewers, and I quote, that “the results section has been updated and re-written following the re-analysis steps suggested by the reviewers”. One of the requests, from both reviewers, was not to do rarefaction and include library sizes as co-variates in the models. The authors showed evidence that their current results do not change considerably when including total read counts in their models, therefore they decided to present their results using rarefied data. However, the current results considerably differ from the previous version. If the current results are still based on rarefied data, and using the same set of samples, what were the modifications made to the methodology? I read the comments of reviewer #1 and the author’s answers. I can only pinpoint an additional quality filtering step that removes reads matching human DNA. Is there any other change to the data processing?

3. The authors claimed in the first version of the manuscript “We also included negative controls (sputum kit with sterile water and buffer) during sample collection, DNA extraction, PCR amplification and sequencing.”

We, both reviewers, pointed out that the data from negative controls is not presented nor how it was used to identify contaminated samples or the presence of contaminants in the community profiles.

In the current version, the authors state that “We included negative controls (sputum kit with sterile water and buffer) during sample collection, DNA extraction, PCR amplification and sequencing. Negative controls were negative for V3-V4 amplicons at PCR and no sequences were generated after batch processing and sequencing with all other samples.”

Based on my experience, and also from reports of previous studies [see Segal et al. *Nature Microbiology* 2016 (DOI: 10.1038/NMICROBIOL.2016.31), Segal et al. *Microbiome* 2013(<https://doi.org/10.1038/nmicrobiol.2016.31>)], negative controls or background controls do end up producing sequences.

Other comments:

4. Line 230-231 (“Within the significantly associated gene families, only genes for the bacterial malate transport pathway were enriched in HIV”).

The authors didn’t discuss this finding.

5. Line 263 (“we detected a significant reduction in bacterial richness.”). This statement is vague, It gives the impression that HIV-COPD comorbidity had reduced bacterial richness compared to the opposite (HIV and COPD negative). Based on the data, this is not the case.
6. Line 265 (“three community types, whose distribution was significantly impacted by HIV status.”). Again, based on the data, this is an overstatement. The data showed that only community type 3 had a statistically significant higher frequency among HIV+.
7. Lines 268-281. The second paragraph in the Discussion section is confusing. It focuses on describing how sputum microbial composition varies across HIV-infected individuals and the factors influencing that variation. The paragraph does not discuss, clearly, why the current study only shows “subtle compositional differences” between HIV+/- groups as opposed to substantial differences found in other studies.
8. Lines 289-294 (“In this study, we clearly demonstrate the effects of HIV status on the distribution of the microbial community types we defined.”). This is an overstatement since the data does not support it. The authors describe how the frequencies of those community types were higher or lower despite those differences not being statistically significant for all community types.
9. Lines 315-316 (“We could show that community type 3, dominated by the Prevotella genera, is predominant in HIVpositive study participants.”).
Another overstatement. Although community type 3 is more frequent in HIV+, compared to HIV-, the other two community types represent 63% of HIV+ participants.
10. Lines 326-339. It is not clear why the authors decided to compare the microbial profiles of their Ugandan cohort with a cohort from the UK. Geographical differences are reported but this finding is completely confounded by the use of a different DNA extraction method in the UK cohort. Including this comparison in the manuscript only adds confusion.
11. Revise the labels of Y-axes in Figure 4A. There are either duplicates or incorrect labels.

Thank you to the authors for answering most of my comments and incorporating changes based on those comments. The changes made to clarify the study design, sample collection, and statistical approach have improved the quality and clarity of the manuscript considerably. However, some of the changes raise the following major concerns:

1. In the previous version of the manuscript, the authors described a case-control design where a case group (HIV+COPD+) is matched to three control groups (HIV-/COPD-, HIV-/COPD+, HIV+/COPD-). The matching was performed based on the frequency distribution of values for three variables in the case group: age (4 categories), sex (2 categories), and smoking status (4 categories). I asked the authors to clarify and extend in the manuscript the description of how the frequency-based approach for matching was achieved. It is surprising to see that in the current version, the authors removed any mention of the frequency-based matching. What happened? Did the authors mistakenly describe the study design in the first version? Although the study design that is described in the current version would be fine to support the current results (which have also changed), the drastic change raises suspicion about the integrity of the research.

2. The authors explained in their response to the reviewers, and I quote, that “the results section has been updated and re-written following the re-analysis steps suggested by the reviewers”. One of the requests, from both reviewers, was not to do rarefaction and include library sizes as co-variates in the models. The authors showed evidence that their current results do not change considerably when including total read counts in their models, therefore they decided to present their results using rarefied data. However, the current results considerably differ from the previous version. If the current results are still based on rarefied data, and using the same set of samples, what were the modifications made to the methodology? I read the comments of reviewer #1 and the author’s answers. I can only pinpoint an additional quality filtering step that removes reads matching human DNA. Is there any other change to the data processing?

3. The authors claimed in the first version of the manuscript “We also included negative controls (sputum kit with sterile water and buffer) during sample collection, DNA extraction, PCR amplification and sequencing.”

We, both reviewers, pointed out that the data from negative controls is not presented nor how it was used to identify contaminated samples or the presence of contaminants in the community profiles.

In the current version, the authors state that “We included negative controls (sputum kit with sterile water and buffer) during sample collection, DNA extraction, PCR amplification and sequencing. Negative controls were negative for V3-V4 amplicons at PCR and no sequences were generated after batch processing and sequencing with all other samples.”

Based on my experience, and also from reports of previous studies [see Segal et al. *Nature Microbiology* 2016 (DOI: 10.1038/NMICROBIOL.2016.31), Segal et al. *Microbiome* 2013(<https://doi.org/10.1038/nmicrobiol.2016.31>)], negative controls or background controls do end up producing sequences.

Other comments:

4. Line 230-231 (“Within the significantly associated gene families, only genes for the bacterial malate transport pathway were enriched in HIV”).

The authors didn’t discuss this finding.

5. Line 263 (“we detected a significant reduction in bacterial richness.”). This statement is vague, It gives the impression that HIV-COPD comorbidity had reduced bacterial richness compared to the opposite (HIV and COPD negative). Based on the data, this is not the case.
6. Line 265 (“three community types, whose distribution was significantly impacted by HIV status.”). Again, based on the data, this is an overstatement. The data showed that only community type 3 had a statistically significant higher frequency among HIV+.
7. Lines 268-281. The second paragraph in the Discussion section is confusing. It focuses on describing how sputum microbial composition varies across HIV-infected individuals and the factors influencing that variation. The paragraph does not discuss, clearly, why the current study only shows “subtle compositional differences” between HIV+/- groups as opposed to substantial differences found in other studies.
8. Lines 289-294 (“In this study, we clearly demonstrate the effects of HIV status on the distribution of the microbial community types we defined.”). This is an overstatement since the data does not support it. The authors describe how the frequencies of those community types were higher or lower despite those differences not being statistically significant for all community types.
9. Lines 315-316 (“We could show that community type 3, dominated by the Prevotella genera, is predominant in HIVpositive study participants.”).
Another overstatement. Although community type 3 is more frequent in HIV+, compared to HIV-, the other two community types represent 63% of HIV+ participants.
10. Lines 326-339. It is not clear why the authors decided to compare the microbial profiles of their Ugandan cohort with a cohort from the UK. Geographical differences are reported but this finding is completely confounded by the use of a different DNA extraction method in the UK cohort. Including this comparison in the manuscript only adds confusion.
11. Revise the labels of Y-axes in Figure 4A. There are either duplicates or incorrect labels.

Revision II : Spectrum02139-21R1 (Sputum microbiome and Chronic Obstructive Pulmonary Disease in a rural Ugandan cohort of well-controlled HIV infection)

Reviewers' comments

We thank the reviewers for their keen observation, interest, time and dedication to provide detailed feedback concerning our manuscript. We have taken ample time to critically consider and work on the reviewers' suggestions and address their concerns below in a point-by-point response below.

Major concern 1:

1. In the previous version of the manuscript, the authors described a case-control design where a case group (HIV+ COPD+) is matched to three control groups (HIV-/COPD-, HIV-/COPD+, HIV+/COPD-). The matching was performed based on the frequency distribution of values for three variables in the case group: age (4 categories), sex (2 categories), and smoking status (4 categories). I asked the authors to clarify and extend in the manuscript the description of how the frequency-based approach for matching was achieved. It is surprising to see that in the current version, the authors removed any mention of the frequency-based matching. What happened? Did the authors mistakenly describe the study design in the first version? Although the study design that is described in the current version would be fine to support the current results (which have also changed), the drastic change raises suspicion about the integrity of the research.

Response

The reviewer is correct. We mistakenly described the study design in the first version, for which we apologise! The intent at the planning stage was to match these variables, but as we clarify in more detail below, this was not feasible. Instead, the study design is not a true matched comparison but a cross-sectional comparison, where we assess for the influence of remaining (modest) confounders using post-hoc statistical tests as we have done for medication status in other recent work. That is to say, this aspect of the work has not changed; it was merely incorrectly described in the first version due to miscommunication between authors. As we revised the manuscript, this was corrected, but we failed to comprehensively document that we did so when responding to your initial concerns, for which we also apologise. The present phrasing, accordingly, is that which correctly describes the work which was done, and we hope this response here clarifies. As for the change in results, this follows not from the study design but from addressing another reviewer's concern, namely that of potential human contamination of the 16S results, which we now filter out.

In more detail, the "COPD+/HIV+" group comprised initial participant screening and enrollment. Among accessible HIV-infected individuals from the previously established HiLiNK cohort, only 50 participants had COPD (Figure 1). We successfully recruited all these participants. We then aimed to frequency-match controls to the COPD+/HIV+ group based on three characteristics (i.e. age, sex and smoking status). Unfortunately, entirely doing so was not possible within the scope of our available source cohorts, resulting in at least moderate bias between groups in these regards (Table 1). We thus must rely on post-hoc testing for the role of these covariates, as outlined elsewhere in the manuscript. We have added a discussion on the resulting limitations to the revised manuscript.

Major concern 2

2. The authors explained in their response to the reviewers, and I quote, that "the results section has been updated and re-written following the re-analysis steps suggested by the reviewers". One of the requests, from both reviewers, was not to do rarefaction and include library sizes as co-variates in the models. The authors showed evidence that their current results do not change considerably when including total read counts in their models, therefore they decided to present their results using rarefied data. However, the current results considerably differ from the previous version. If the current results are still based on rarefied data, and using the same set of samples, what were the modifications made to the methodology? I read the comments of reviewer #1 and the author's answers. I can only pinpoint an additional quality filtering step that removes reads matching human DNA. Is there any other change to the data processing?

Response: Indeed major changes in the result section were induced by discarding reads mapping to the human genome. We were very much surprised as well to see that this step had such a substantial impact; indeed, "standard" 16S workflows - at least when analysing gut microbiome data - usually do not consider performing this step. However, the impact of host contamination seems more significant in low-biomass samples such as sputum samples. We are following up on these phenomena in an independent study and plan to publish the results soon.

Additionally, as we rewrote the analysis framework scripts now allow the comparison between 1) rarefied and unrarefied reads with total number of reads as a covariate and 2) with and without reads aligned to human DNA; we discovered and corrected two minor mistakes in the original scripts, which further modified results between the two versions. These mistakes were i) some metadata variables were wrongly parsed upon loading into R in the original code, and ii) a list containing names of functional gene modules, which was assumed to be sorted, was not actually sorted when first used. As a result, the originally submitted results, which i) had some spurious disease association now fixed and ii) had the wrong functional modules reported and discussed, are now corrected. This correction induced further changes in the results apart from human contamination and rarefaction vs covariate inclusion.

In the present version, as outlined below, we have corrected these issues leading to the difference in results that the reviewer notes. Together with this response, we provide a "clean" comparison of results, unaffected by those mistakes, under the choice of 1) rarefied vs unrarefied reads w. read total covariate and 2) with and without filtering reads for contamination. Of these comparisons, it is clear that removing human DNA is necessary and warranted for this dataset, whereas the impact of rarefaction vs inclusion of total read count as a covariate is minimal. Please see the comments below for further detail on this revision.

Figure A-D show the overall analysis run with and without human contamination and rarefaction, separately and in combination, reporting sputum microbiome impact on main clinical variables and covariates under these settings for comparison. In line with the above summary, results with and without human read filtering are quite different, whereas rarefied and unrarefied results are not.

B) filtered for human reads non rarefied data

D) unfiltered rarefied data

We cannot thank the reviewer enough for their patience in this matter, and their dedication in helping us to spot what would otherwise have been an insidious bug!

Concern 3

3. The authors claimed in the first version of the manuscript "We also included negative controls (sputum kit with sterile water and buffer) during sample collection, DNA extraction, PCR amplification and sequencing." We, both reviewers, pointed out that the data from negative controls is not presented nor how it was used to identify contaminated samples or the presence of contaminants in the community profiles. In the current version, the authors state that "We included negative controls (sputum kit with sterile water and buffer) during sample collection, DNA extraction, PCR amplification and sequencing. Negative controls were negative for V3-V4 amplicons at PCR and no sequences were generated after batch processing

and sequencing with all other samples." Based on my experience, and also from reports of previous studies [see Segal et al. Nature Microbiology 2016 (DOI: 10.1038/NMICROBIOL.2016.31), Segal et al. Microbiome 2013 (<https://doi.org/10.1038/nmicrobiol.2016.31>)], negative controls or background controls do end up producing sequences.

Response: We thank the reviewer for this insightful observation! Most papers have indeed reported sequences generated from their negative controls, similar to Segal et al. 2016. In the present study, all samples were batch processed together with the negative control, with all samples assigned pseudonymous IDs to maintain uniform blinding. The sequencing workflow was such that samples which failed to reach PCR amplification thresholds were considered failures and omitted from the sequencing step. This was the case for one sample, which was revealed to be the negative control upon unblinding. We recognise that a better, more sophisticated utilisation of a negative control sample would be to take it further to sequencing and consider any hits there as "contaminant taxa" in other samples. While we cannot rerun it now, we will take this insight with us to future studies. Details on how the negative control was used and ensuing limitations are now elaborated in the revised manuscript.

Comment 4:

4. Line 230-231 ("Within the significantly associated gene families, only genes for the bacterial malate transport pathway was enriched in HIV"). The authors didn't discuss this finding.

Response: As noted above in response to major concern 2, there were two coding mistakes in the originally submitted manuscript version, which we corrected in the first revision without realising they were there. The second affected the listed functional pathway names of the significantly differentially abundant gene functional modules. Thus, while there is an HIV-associated functional profile visible in this dataset, it does not center malate transport but actually a different set of modules. We report and discuss the corrected profile in the present manuscript version, as outlined below.

New Results:

To determine the projected function profiles of the sputum microbiota using 16S rRNA data, we used PICRUST2 (version 2.2.3). PICRUST2 is a tool to infer the functional profiles of bacterial communities based on their taxonomic composition. Among the significantly associated KEGG and GMM modules, only the glutamate degradation module (MF0015) was negatively associated with HIV status and its associated antiviral therapy. COPD status was not associated with any changes in modules. However, we could detect a significant increase in modules associated with signalling machinery of two-component systems (TCSs), drug resistance, and smoking. Even more peculiarly, an individual's water source (dug well, borehole, or public tap) is significantly associated with abundant gene modules for propionate production and cellular transport systems.

New discussion:

Furthermore, we found a depletion of Staphylococales and Negativicutes, predominantly derived from the oral flora (26) under COPD/HIV comorbidity, again indicating an interaction between these conditions about host-microbiome homeostasis. Our finding of decreased glutamate degradation capacity in the HIV sputum microbiota may further elucidate aspects of pathology in context. Amino acid availability is central to the immune system's metabolism and function, especially during infection. As a condition becomes

chronic, these alterations become more complex as various other areas of metabolism become impaired, and amino acids may antagonise each other's effects. Glutaminolysis has been postulated as a mechanism by which the TCA cycle is replenished during viral infection (36). This decrease might indicate that the sputum microbiome in HIV patients reduces its ability to generate energy via TCA as an appropriate response to changes in the microenvironment, which might subsequently lead to dysbiosis, facilitating COPD pathogenesis.

Further direct functional assessment is needed to validate and explore this finding. Even with ART available, HIV patients are at high risk of suffering comorbidities, as shown by the high prevalence of non-infectious lung diseases in the HIV population. It is, therefore, important to better understand the complex changes in the sputum microbiota in patients with COPD/HIV comorbidity to find potential prevention and intervention targets. The presented study cohort is well-standardised and characterized. However, the cross-sectional design, which limits inference of causality, as well as the use of induced sputum samples, causing possible contamination from the oral cavity, leads to limitations. The use of short-read 16S amplicon 16S rRNA gene Illumina sequencing is limiting the resolution of taxonomic classification to genus-level taxonomy, and the inferred function profiles using taxonomic projection likewise are limited in terms of interpretability.

References

26. Li Y, Saxena D, Chen Z, Liu G, Abrams WR, Phelan JA, et al. HIV infection and microbial diversity in saliva. *J Clin Microbiol.* 2014 May;52(5):1400–11.

36. González Plaza JJ, Hulak N, Kausova G, Zhumadilov Z, Akilzhanova A. Role of metabolism during viral infections, and crosstalk with the innate immune system. *Intractable rare Dis Res.* 2016 May;5(2):90–6.

Comment 5

5. Line 263 ("we detected a significant reduction in bacterial richness."). This statement is vague. It gives the impression that HIV-COPD comorbidity had reduced bacterial richness compared to the opposite (HIV and COPD negative). Based on the data, this is not the case.

Response: We thank the reviewer for indicating this and changed the paragraph in the discussion accordingly.

New paragraph :

While the disease subcohorts showed only subtle differences in sputum microbiome composition in our present study, we observed a significantly higher microbiome richness for the COPD+/HIV- group than the COPD+/HIV+ and COPD-/HIV- groups. Additionally, the Chao index was significantly higher in the COPD+/HIV- group than in the other subgroups. Such loss of diversity was previously reported in patients who have HIV. Somewhat unexpectedly, however, sputum richness and Chao1 index were elevated in COPD patients. It seems that these two diseases affect the microbiome differently. If underlying comorbidity of COPD/HIV is present, synergistic effects might occur, forming an interesting approach for future hypotheses.

Comment 6:

6. Line 265 ("*three community types, whose distribution was significantly impacted by HIV status.*"). Again, based on the data, this is an overstatement. The data showed that only community type 3 had a statistically significant higher frequency among HIV+.

We thank the reviewer for bringing this to our attention. We have clarified the results and subsequently included a discussion in comment 8-9 as requested by the reviewer.

To determine whether distinct microbial community structures exist within our cohort, unsupervised modeling of genus abundance frequencies using Dirichlet multinomial mixtures (DMM) was applied to the 16S rRNA datasets. Using a Laplace approximation, DMM indicated that the dataset presents three distinct microbial community structures (community type 1-3). Community type 1, primarily dominated by *Streptococcus* followed by *Neisseria*, *Haemophilus* and *Prevotella*, characterised 72 samples. Community type 2 is dominated by a mix of bacterial genera, including *Neisseria*, *Streptococcus*, and *Haemophilus*, followed by *Veillonella*, *Fusobacteria*, *Porphyromonas*, and *Prevotella* classified in 67 samples. Community type 3, dominated by *Prevotella*, *Streptococcus*, and *Veillonella*, followed by *Bacteroidia* and *Alloprevotella*, characterised 61 samples (Figure 3A). Accordingly, these community types show overlapping sets of driver taxa in different proportions and accompanying rarer taxa. Univariate analysis of microbial richness and evenness diversity indices showed a significant reduction in Shannon, Simpson, Inverse Simpson and Pielou's indices and microbial evenness in community type 1 compared to the others (Figure 3C). In contrast, community type 2 showed a significant increase compared to the other two in Shannon, Simpson and Inverse Simpson's indices and overall microbial richness (Figure 3C). Stratification by disease status showed a slight skew in the community type distribution (Figure 3B), with community type 3 being the rarest and slightly less so in the two HIV-positive subgroups. However, significance was achieved only by comparing COPD-/HIV+ subjects with COPD-/HIV- ones (39% versus 17%, $q=0.07$, FDR-corrected).

Comment 7

7. Lines 268-281. The second paragraph in the Discussion section is confusing. It focuses on describing how sputum microbial composition varies across HIV-infected individuals and the factors influencing that variation. The paragraph does not discuss, clearly, why the current study only shows "subtle compositional differences" between HIV+/- groups as opposed to substantial differences found in other studies.

Response

We thank the reviewer for bringing this to our attention. As requested, we have clarified and edited our discussion, outlining how the present cohort differs from previous work in a way that seems to underlie this subtler signal.

As expected, analysis of sputum microbe differential abundance reflected collinearity in the cohort of HIV seropositivity with ART treatment and its duration. Thus, we cannot at present disentangle the impacts of these factors. However, previously reported genera like *Veillonella*, *Actinomyces*, *Atopobium*, and *Filifactor* were significantly enriched in HIV-positive individuals (23,27,31). These genera were previously associated with proinflammatory cytokine production (27,32), which may be an aspect of airway dysbiosis in HIV+ subjects, possibly further interacting with other risk factors of COPD.

References

23. Lawani MB, Morris A. The respiratory microbiome of HIV-infected individuals. *Expert Rev Anti Infect Ther*. 2016 Aug;14(8):719–29.
27. Bhadriraju S, Fadrosch DW, Shenoy MK, Lin DL, Lynch K V, McCauley K, et al. Distinct lung microbiota associate with HIV-associated chronic lung disease in children. *Sci Rep [Internet]*. 2020;10(1):16186. Available from: <https://doi.org/10.1038/s41598-020-73085-1>
31. Ueckermann V, Lebre P, Geldenhuys J, Hoosien E, Cowan D, van Rensburg LJ, et al. The lung microbiome in HIV-positive patients with active pulmonary tuberculosis. *Sci Rep [Internet]*. 2022;12(1):8975. Available from: <https://doi.org/10.1038/s41598-022-12970-3>
32. Zevin AS, McKinnon L, Burgener A, Klatt NR. Microbial translocation and microbiome dysbiosis in HIV-associated immune activation. *Curr Opin HIV AIDS*. 2016 Mar;11(2):182–90.

Comments 8 and 9:

8. Lines 289-294 ("In this study, we clearly demonstrate the effects of HIV status on the distribution of the microbial community types we defined."). This is an overstatement since the data does not support it. The authors describe how the frequencies of those community types were higher or lower despite those differences not being statistically significant for all community types. 9. Lines 315-316 ("We could show that community type 3, dominated by the *Prevotella* genera, is predominant in HIV positive study participants."). Another overstatement. Although community type 3 is more frequent in HIV+, compared to HIV-, the other two community types represent 63% of HIV+ participants.

Response

We thank the reviewer for bringing this to our attention. We have clarified the discussion as requested by the reviewer

In our analysis, we used an unsupervised cluster approach to define sputum microbiome communities describing variability in our cohort. The communities broadly separate into three clusters. Each sample consistently contains *Streptococcus*, *Veillonella*, *Fusobacterium*, and *Prevotella*, though in varying proportions and with other associated taxa accompanying them. We find these structures in healthy and diseased individuals; overall, the disease status does not substantially determine the community type. Comparing disease sub-cohorts, there is a trend towards community state 3, which generally is the least common, to be slightly less rare among participants living with HIV. However, for this dataset, the significance of this trend is reached only by comparing HIV-discordant COPD-negative participants. Accordingly, while this sputum microbiome composition may represent a more dysbiotic state associated perhaps with immunosuppression, we cannot as yet conclude it, only raise it as a possibility for further testing.

Community type three is dominated by *Prevotella*, *Streptococcus* and *Veillonellaceae*. This association between *Prevotella*, *Veillonella* and HIV has been reported previously (24). We did not demonstrate a significant skew in community type along COPD morbidity,

but community type 1, dominated by *Streptococci*, *Neisseria*, and *Haemophilus*, was slightly more prominent in COPD+ participants. This increase in *Streptococci*, *Haemophilus* and *Neisseria* was previously reported in COPD-positive individuals (25). Here, COPD mortality risk could be predicted using microbial-specific signatures such as the presence of *Staphylococcus*, absence of *Veillonella*, and lower alpha diversity (26). COPD+/HIV+ patients showed a more even distribution between the three community types and no further bias towards any community type. How these dynamics change in HIV-associated COPD remains to be elucidated.

While the disease subcohorts showed only subtle differences in sputum microbiome composition in our present study, we observed a significantly higher microbiome richness for the COPD+/HIV- group compared to COPD+/HIV+ and COPD-/HIV- groups. Additionally, the Chao index was significantly higher in the COPD+/HIV- group than in the other subgroups. Such loss of diversity was previously reported in patients who have HIV. Somewhat unexpectedly, however, sputum richness and Chao1 index were elevated in COPD patients. It seems that these two diseases affect the microbiome differently. If underlying comorbidity of COPD/HIV is present, synergistic effects might occur, forming an interesting approach for future hypotheses.

Reference:

Segal LN, Dickson RP. *The Lung Microbiome in HIV. Getting to the HAART of the Host-Microbe Interface. Am J Respir Crit Care Med.* 2016 Jul;194(2):136–7.

25. Huang YJ, Boushey HA. *The Sputum Microbiome in Chronic Obstructive Pulmonary Disease Exacerbations. Ann Am Thorac Soc.* 2015 Nov;12 Suppl 2(Suppl 2):S176-80.

26. Li Y, Saxena D, Chen Z, Liu G, Abrams WR, Phelan JA, et al. *HIV infection and microbial diversity in saliva. J Clin Microbiol.* 2014 May;52(5):1400–11.

Comment 10

10. Lines 326-339. *It is not clear why the authors decided to compare the microbial profiles of their Ugandan cohort with a cohort from the UK. Geographical differences are reported but this finding is completely confounded by the use of a different DNA extraction method in the UK cohort. Including this comparison in the manuscript only adds confusion.*

Response

We thank the reviewer for bringing this to our attention. In the manuscript, we do acknowledge and describe in detail methodological differences between Ugandan versus UK samples, outlining resulting limitations. However, if the editor and reviewers think it is rather confusing, we are happy to omit these comparisons.

Comment 11

11. *Revise the labels of Y-axes in Figure 4A. There are either duplicates or incorrect labels.*

Response: We thank the reviewer for this keen observation. We have revised the labels of the Y-axes.

January 23, 2023

Dr. Sofia Kirke Forslund
Max Delbrück Center for Molecular Medicine
Berlin, Germany
Germany

Re: Spectrum02139-21R2 (Sputum microbiome and Chronic Obstructive Pulmonary Disease in a rural Ugandan cohort of well-controlled HIV infection)

Dear Dr. Sofia Kirke Forslund:

Thank you for submitting your revised manuscript to Spectrum. Your manuscript has been accepted, and I am forwarding it to the ASM Journals Department for publication. You will be notified when your proofs are ready to be viewed.

Sincerely,

Kevin R. Theis
Editor, Microbiology Spectrum
